# Multi-Agent Meta-Reinforcement Learning: Sharper Convergence Rates with Task Similarity

**Weichao Mao**
University of Illinois Urbana-Champaign
weichao2@illinois.edu

**Haoran Qiu**
University of Illinois Urbana-Champaign
haoranq4@illinois.edu

**Chen Wang**
IBM Research
chen.wang1@ibm.com

**Hubertus Franke**
IBM Research
frankeh@us.ibm.com

**Zbigniew Kalbarczyk**
University of Illinois Urbana-Champaign
kalbarcz@illinois.edu

**Ravishankar K. Iyer**
University of Illinois Urbana-Champaign
rkiyer@illinois.edu

**Tamer Başar**
University of Illinois Urbana-Champaign
basar1@illinois.edu

## Abstract

Multi-agent reinforcement learning (MARL) has primarily focused on solving a single task in isolation, while in practice the environment is often evolving, leaving many related tasks to be solved. In this paper, we investigate the benefits of meta-learning in solving multiple MARL tasks collectively. We establish the first line of theoretical results for meta-learning in a wide range of fundamental MARL settings, including learning Nash equilibria in two-player zero-sum Markov games and Markov potential games, as well as learning coarse correlated equilibria in general-sum Markov games. Under natural notions of task similarity, we show that meta-learning achieves provable sharper convergence to various game-theoretical solution concepts than learning each task separately. As an important intermediate step, we develop multiple MARL algorithms with initialization-dependent convergence guarantees. Such algorithms integrate optimistic policy mirror descents with stage-based value updates, and their refined convergence guarantees (nearly) recover the best known results even when a good initialization is unknown. To our best knowledge, such results are also new and might be of independent interest. We further provide numerical simulations to corroborate our theoretical findings.

## 1 Introduction

Many real-world sequential decision-making problems involve multiple agents interacting in a shared environment, a scenario commonly captured by game theory and addressed using multi-agent reinforcement learning (MARL). Existing research in MARL has primarily focused on solving a single task (i.e., a game) independently. In practice, however, one often needs to collectively solve a set of similar tasks due to the dynamically evolving environment. For example, in sponsored search auctions [48], the advertising spaces and search results are dynamic, and each bidder with an active bid will participate in a sequence of related auctions. In multi-robot cooperation [31, 26], the learning agents are often first pre-trained in simplified environments and are then asked to quickly adapt to more complicated ones. In cloud computing [53, 70], a learning-based autoscaling policy needs to achieve fast model adaptation to deal with varied application workloads or constantly evolving cloud infrastructures. All of these intriguing applications call for the development of intelligent multi-agent systems that can continuously build on previous experiences to enhance the learning of new tasks.

37th Conference on Neural Information Processing Systems (NeurIPS 2023).

Meta-learning, or learning-to-learn [64, 56, 65, 58], is a rapidly developing approach that is particularly suitable for learning in a set of related tasks. In essence, meta-learning studies the use of data from existing tasks to learn representations or model parameters that enable quick adaptation to new tasks. By exploiting the knowledge obtained from prior tasks, the meta-learner can ideally solve an unseen task using much fewer training samples than learning from scratch, especially when the tasks share some inherent similarities. Despite many empirical successes [71, 31, 26], the theoretical results of meta-learning in multi-agent scenarios are still relatively lacking. It remains elusive whether meta-learning can provably expedite the convergence of MARL, and if so, what the proper task similarity assumptions to impose are. In fact, it is even unclear whether a meta-learner converges at all in a highly non-stationary system with loosely-coupled learning agents and diverse task setups.

In this paper, we make an initial attempt toward characterizing some of the central theoretical properties of meta-learning in a wide range of fundamental MARL settings. We focus on the classic model-agnostic meta-learning (MAML) [20] type of algorithms that aim to learn a good initialization for quick adaptation to new tasks. To study the convergence rate of MAML, an important prerequisite is to understand how the convergence of MARL algorithms depends on the quality of policy initialization. However, the convergence guarantees of most existing MARL algorithms are initialization-independent: They fail to track how the sub-optimality of the initial policy propagates during the learning process, and only provide pessimistic guarantees with respect to worst-case initialization. As a crucial intermediate step to meta-MARL, we need to establish refined *initialization-dependent* convergence guarantees for MARL. Our main contributions are thus summarized as follows.

**Contributions.** 1) For learning Nash equilibria (NE) in two-player zero-sum Markov games, we first propose an MARL algorithm blessed with a refined convergence analysis that explicitly characterizes the dependence on policy initialization (Section 3.1). Our algorithm runs optimistic online mirror descent for policy optimization and performs stage-based value function updates. Even when initialized with random policies, our algorithm still matches the best-known convergence rates in the literature except for an extra logarithmic term. Our algorithm and analysis appear to be new and might be of independent interest. 2) Based on such refined analysis, we show that meta-learning provably achieves faster convergence to NE when learning a sequence of "similar" zero-sum games collectively, where our similarity metric naturally depends on the closeness of the games' NE policies (Section 3.2). 3) For learning NE in Markov potential games (MPGs), we show that a simple refinement of an existing algorithm suffices to provide initialization-dependent guarantees. We establish sharper convergence rates of meta-learning when the MPGs have similar potential functions (Section 4.1). In addition, with a properly chosen policy update rule, we prove the non-asymptotic convergence of the exact MAML algorithm in MPGs (Section 4.2), despite the convoluted learning dynamics of multiple loosely-coupled agents. 4) For learning coarse correlated equilibria (CCE) in general-sum Markov games (Section 5), we analogously start by designing an initialization-dependent MARL algorithm, and then establish the sharper convergence rate of meta-learning under natural similarity metrics. 5) We provide numerical results to corroborate our theoretical findings (Section 6).

**Related Work.** Gradient-based meta-learning is a simple and effective approach that can be easily applied to any learning problem trained with gradient descent. The seminal MAML method [20] tries to learn a good model parameter initialization that leads to quick model adaptation. Theoretical properties of MAML have been investigated in a series of works [54, 17, 66, 18, 30]. In particular, [17, 30] have established the convergence of MAML to first-order stationarity for non-convex objectives. [18] has designed an unbiased gradient estimator for MAML in reinforcement learning tasks. Various first-order approximations [20, 49, 17] of MAML have been proposed to avoid the heavy computation of the Hessian. Meta-learning has also been studied in online convex optimization [21, 4, 14, 35], where regret bounds have been established under different metrics of task similarity. Another line of research [29, 51, 41] views meta-learning through the lens of task inference, where an RL policy is conditioned on a belief over tasks and perform Bayesian updates through interactions to adapt to different tasks.

MARL has been widely studied under the formulation of stochastic games (i.e., Markov games) [57]. Due to the fundamental difficulty of computing NE in generic games [10], most MARL research has focused on learning NE in games with special structures (such as zero-sum Markov games [67, 2, 69, 3, 12, 68, 7, 72, 73] and Markov potential games [43, 38, 76, 25, 22, 15, 78]) or learning weaker solution concepts such as (coarse) correlated equilibria [40, 44, 59, 33, 47, 16, 13]. The most relevant works are [77, 72], which have studied the convergence of optimistic no-regret

learning and smooth value updates in MARL with full-information feedback. For learning NE in MPGs, [38, 76, 15] have studied independent policy gradient methods and established their sample complexity results. These works have focused on learning a single game in isolation but have not considered exploiting the connections between multiple games to expedite the learning process.

Most related to ours, [27] has studied meta-learning in normal-form games. Under different notions of game similarities, [27] has shown faster convergences of meta-learning in zero-sum, general-sum, and Stackelberg games. [75] has investigated no-regret learning in time-varying zero-sum normal-form games. Compared to [27, 75], we consider meta-learning in the more generic and challenging Markov game setup with state transitions. Other related works include meta-learning for regret minimization in a distribution of games [61] and meta-safe RL for quick adaptation in constrained Markov decision processes (CMDPs) under task similarity [34]. Finally, meta-learning has also been empirically applied to many important MARL scenarios, including multi-intersection traffic signal control [71], multi-agent communication with natural language [26], and multi-agent collaboration with first-person pixel observations in open-ended tasks [63].

## 2 Preliminaries

**Markov game.** An $N$-player episodic Markov game is defined by a tuple $\mathbb{G} = (\mathcal{N}, H, \mathcal{S}, \{\mathcal{A}_i\}_{i=1}^N, \{r_i\}_{i=1}^N, P)$, where (1) $\mathcal{N} = \{1, 2, \ldots, N\}$ is the set of agents; (2) $H \in \mathbb{N}_+$ is the number of time steps in each episode; (3) $\mathcal{S}$ is the finite state space; (4) $\mathcal{A}_i$ is the finite action space for agent $i \in \mathcal{N}$; (5) $r_i : [H] \times \mathcal{S} \times \mathcal{A}_{\text{all}} \to [0, 1]$ is the reward function for agent $i$, where $\mathcal{A}_{\text{all}} = \times_{i=1}^N \mathcal{A}_i$ is the joint action space; and (6) $P : [H] \times \mathcal{S} \times \mathcal{A}_{\text{all}} \to \Delta(\mathcal{S})$ is the transition kernel. The agents interact in an unknown environment for $T$ episodes. Without loss of generality, we make a standard assumption [33, 59] that each episode starts from a fixed initial state $s_1$. Our results can be easily generalized to the setting where the initial state is sampled from a fixed distribution. At each time step $h \in [H]$, the agents observe the state $s_h \in \mathcal{S}$, and take actions $a_{h,i} \in \mathcal{A}_i, i \in \mathcal{N}$ simultaneously. Agent $i$ then receives its reward $r_{h,i}(s_h, \boldsymbol{a}_h)$, where $\boldsymbol{a}_h = (a_{h,1}, \ldots, a_{h,N})$, and the environment transitions to the next state $s_{h+1} \sim P_h(\cdot|s_h, \boldsymbol{a}_h)$. Let $S = |\mathcal{S}|$, $A_i = |\mathcal{A}_i|, \forall i \in \mathcal{N}$, and $A_{\max} = \max_{i \in \mathcal{N}} A_i$.

**Policy and Nash equilibrium.** A (Markov) policy $\pi_i \in \Pi_i : [H] \times \mathcal{S} \to \Delta(\mathcal{A}_i)$ for agent $i \in \mathcal{N}$ is a mapping from the time index and state space to a distribution over its own action space. Each agent seeks to find a policy that maximizes its own cumulative reward. A joint, product policy $\pi = (\pi_1, \ldots, \pi_N) \in \Pi$ induces a probability measure over the sequence of states and joint actions. We use the subscript $-i$ to denote the set of agents excluding agent $i$, i.e., $\mathcal{N} \backslash \{i\}$. We can rewrite $\pi = (\pi_i, \pi_{-i})$ using this convention. For a joint policy $\pi$, and for any $h \in [H]$, $s \in \mathcal{S}$, and $\boldsymbol{a} \in \mathcal{A}_{\text{all}}$, we define the value function and Q-function for agent $i$ as

$$V_{h,i}^{\pi}(s) := \mathbb{E}_{\pi}\Big[\sum_{h'=h}^H r_{h',i}(s_{h'}, \boldsymbol{a}_{h'})|s_h = s\Big], \; Q_{h,i}^{\pi}(s, \boldsymbol{a}) := \mathbb{E}_{\pi}\Big[\sum_{h'=h}^H r_{h',i}(s_{h'}, \boldsymbol{a}_{h'})|s_h = s, \boldsymbol{a}_h = \boldsymbol{a}\Big].$$

For agent $i$, a policy $\pi_i^{\dagger}$ is a *best response* to $\pi_{-i}$ if $V_{1,i}^{\pi_i^{\dagger}, \pi_{-i}}(s_1) = \sup_{\pi_i} V_{1,i}^{\pi_i, \pi_{-i}}(s_1)$. A joint (product) policy $\pi = (\pi_i, \pi_{-i}) \in \Pi$ is a *Nash equilibrium* (NE) if $\pi_i$ is a best response to $\pi_{-i}$ for all $i \in \mathcal{N}$. Similarly, for any $\varepsilon > 0$, a joint policy $\pi = (\pi_i, \pi_{-i})$ is an $\varepsilon$-approximate NE if $V_{1,i}^{\pi_i, \pi_{-i}}(s_1) \geq V_{1,i}^{\pi_i^{\dagger}, \pi_{-i}}(s_1) - \varepsilon, \forall i \in \mathcal{N}$.

**Correlated policy and coarse correlated equilibrium.** We define $\pi = \{\pi_h : \mathbb{R} \times (\mathcal{S} \times \mathcal{A})^{h-1} \times \mathcal{S} \to \Delta(\mathcal{A})\}_{h \in [H]}$ as a (non-Markov) *correlated policy*, where for each $h \in [H]$, $\pi_h$ maps from a coordination device $z \in \mathbb{R}$ and a history of length $h-1$ to a distribution over the joint action space. Let $\pi_i$ and $\pi_{-i}$ be the proper marginal distributions of $\pi$ whose outputs are restricted to $\Delta(\mathcal{A}_i)$ and $\Delta(\mathcal{A}_{-i})$, respectively. The value functions for non-Markov correlated policies at step $h = 1$ are defined in a similar way as for product policies. Given the PPAD-hardness of calculating NE in general [11], people often study a relaxed solution concept named coarse correlated equilibrium (CCE), which allows possible correlations in the policies: In particular, for any $\varepsilon > 0$, a *correlated policy* $\pi = (\pi_i, \pi_{-i})$ is an $\varepsilon$-approximate CCE if $V_{1,i}^{\pi_i, \pi_{-i}}(s_1) \geq V_{1,i}^{\pi_i^{\dagger}, \pi_{-i}}(s_1) - \varepsilon, \forall i \in \mathcal{N}$.

**Two-player zero-sum Markov game.** An important special case of Markov games is (two-player) zero-sum Markov games, where there are two players ($N = 2$) with exactly opposite rewards

$(r_1 = -r_2)$. In a zero-sum game, we simply use $r, V$, and $Q$ to denote the reward and (Q-)value functions for the max-player, i.e., agent 1. Correspondingly, the min-player has $-r, -V$, and $-Q$. For notational convenience, we denote the action space for the max-player (resp. min-player) by $\mathcal{A}$ (resp. $\mathcal{B}$), and let $A = |\mathcal{A}|, B = |\mathcal{B}|$. We also write their policies $(\pi_1, \pi_2)$ as $(\mu, \nu)$ for short. In zero-sum games, it is known that although the NE policy $(\mu^\star, \nu^\star)$ may not be unique, all the NE have the same values. We use $V_h^\star$ and $Q_h^\star$ to denote the NE value function and the NE Q-function. For any fixed $(h, s) \in [H] \times \mathcal{S}$ and an arbitrary function $Q : \mathcal{S} \times \mathcal{A} \times \mathcal{B} \to \mathbb{R}$, we may consider $Q(s, \cdot, \cdot)$ as an $A \times B$ matrix. Then, for any policy pair $(\mu_h, \nu_h)$ at step $h \in [H]$, we can write in shorthand:

$$\left[\mu_h^\top Q \nu_h\right](s) := \mathbb{E}_{a \sim \mu_h(\cdot|s), b \sim \nu_h(\cdot|s)}[Q(s, a, b)] = \langle \mu_h, Q\nu_h \rangle(s),$$

$$\left[\mu_h^\top Q\right](s, \cdot) := \mathbb{E}_{a \sim \mu_h(\cdot|s)}[Q(s, a, \cdot)], \text{ and } [Q\nu_h](s, \cdot) := \mathbb{E}_{b \sim \nu_h(\cdot|s)}[Q(s, \cdot, b)].$$

Given the transition function $P$ and an arbitrary function $V : \mathcal{S} \to \mathbb{R}$, we define

$$[P_h V](s, a, b) := \mathbb{E}_{s' \sim P_h(\cdot|s,a,b)}[V(s')].$$

The Bellman equations can hence be rewritten more succinctly as

$$V_h^{\mu,\nu}(s) = \left[\mu_h^\top Q_h^{\mu,\nu} \nu_h\right](s), \text{ and } Q_h^{\mu,\nu}(s, a, b) = r_h(s, a, b) + \left[P_h V_{h+1}^{\mu,\nu}\right](s, a, b).$$

**Markov potential game.** Another important class of games is Markov potential games [43, 39, 76]. MPGs cover Markov teams [36], a fully cooperative setting where all agents share the same rewards. A Markov game is an MPG if there exists a global potential function $\Phi : \Pi \times \mathcal{S} \to [0, \Phi_{\max}]$ that can capture the variations of the agents' individual values: Specifically, $\forall i \in \mathcal{N}$ and $s \in \mathcal{S}$,

$$\Phi_s(\pi_i, \pi_{-i}) - \Phi_s(\pi_i', \pi_{-i}) = V_{1,i}^{\pi_i, \pi_{-i}}(s) - V_{1,i}^{\pi_i', \pi_{-i}}(s), \forall \pi_i, \pi_i' \in \Pi_i, \pi_{-i} \in \Pi_{-i}.$$

Throughout the paper, we consider the classic full-information feedback setting [23, 8, 62, 68, 7], where the players are assumed to have exact information of the consequences of each of their candidate actions. In the case of zero-sum games, this implies that for any $(h, s)$, the max-player and min-player can query $[Q_h \nu_h](s, \cdot)$ and $[\mu_h^\top Q_h](s, \cdot)$, respectively. Our meta-learning results can be easily extended to the stochastic bandit feedback setting using standard techniques as in [3, 44, 33, 15].

**Meta-learning.** Let $\mathcal{G} = \{\mathbb{G}^k\}$ be a set of different Markov games. Each game is defined by $\mathbb{G}^k = (\mathcal{N}, H, \mathcal{S}, \{\mathcal{A}_i\}_{i=1}^N, \{r_i^k\}_{i=1}^N, P^k)$, where we assume without loss of generality that the games share the same agent set and state & action spaces, but can have different transition and reward functions. Most of our results are established in the online learning setting where we encounter a sequence of $K$ games $(\mathbb{G}^1, \ldots, \mathbb{G}^K)$ one by one. To achieve faster convergence, the learning agents should use the knowledge obtained from previous games to expedite the learning process in future games.

The underlying principle of MAML [20] is to learn a good initialization such that running a few training steps from this initialization can lead to well-performing model parameters on any new task. An MAML-type algorithm in the context of RL typically involves two nested stages. The inner stage (or "base algorithm") $\psi$ performs $T$ iterations of policy updates to optimize for an individual task $\mathbb{G}^k$:

$$\pi^{k,t} \leftarrow \psi(\pi^{k,t-1}; \mathbb{G}^k), \forall t \in [T]. \tag{1}$$

When task $\mathbb{G}^k$ is completed, the outer stage (or "meta-algorithm") $\Psi$ learns to form a good initialization $\pi^{k+1,0}$ for a new task $\mathbb{G}^{k+1}$ using all the knowledge obtained from all previous tasks:

$$\pi^{k+1,0} \leftarrow \Psi(\{\pi^{k',t}\}_{k' \in [k], t \in [T]}; \mathbb{G}^1, \ldots, \mathbb{G}^k). \tag{2}$$

In this paper, we seek to properly instantiate both the base algorithm $\psi$ and the meta-algorithm $\Psi$ for a variety of MARL problems. We aim to show that a proper design of the meta-learning procedure $(\psi, \Psi)$ can largely reduce the number of iterations $T$ required to find NE or CCE in a new game.

## 3 Meta-Learning for Two-Player Zero-Sum Markov Games

In this section, we study meta-learning for Nash equilibria in zero-sum Markov games, where players are fully competitive. Since MAML-type algorithms seek to learn a good initialization for quick adaptation, it is crucial to explicitly characterize how the convergence behavior of an MARL algorithm depends on the initial policy. To our best knowledge, such results are not directly achievable using existing algorithms. For this reason, in Section 3.1, we start by proposing a new base algorithm (1) for zero-sum Markov games that has a refined initialization-dependent convergence guarantee. Based on that, we present our meta-algorithm (2) in Section 3.2 and establish its sharper convergence rates.

## 3.1 Initialization-Dependent Convergence in an Individual Zero-Sum Markov Game

Algorithm 1 presents our optimistic online mirror descent algorithm with stage-based value updates for learning NE in a zero-sum Markov game. To establish initialization-dependent convergence, Algorithm 1 performs optimistic online mirror descent (OMD) [55, 62] for policy updates (Lines 5 and 6), in contrast to the popular optimistic follow the regularized leader (FTRL) method in recent MARL policy optimization [77, 72]. We choose the negative entropy as our regularizer $R$, in which case the Bregman divergence $D_R(\cdot, \cdot)$ reduces to the Kullback–Leibler divergence and optimistic OMD becomes an optimistic variant of the classic multiplicative weights update (MWU) algorithm.

---

**Algorithm 1:** Optimistic Online Mirror Descent for Zero-Sum Markov Games

---

1 **Input:** Initial policies $\tilde{\mu} : [\bar{\tau}] \times [H] \times \mathcal{S} \to \Delta(\mathcal{A})$ and $\tilde{\nu} : [\bar{\tau}] \times [H] \times \mathcal{S} \to \Delta(\mathcal{B})$;

2 Set stage index $\tau \leftarrow 1$, $t_\tau^{\text{start}} \leftarrow 1$, and $L_\tau \leftarrow H$;

3 **Initialize:** $\mu_h^0 = \hat{\mu}_h^0 \leftarrow \tilde{\mu}_h^1$, $\nu_h^0 = \hat{\nu}_h^0 \leftarrow \tilde{\nu}_h^1$, and $Q_h^\tau \leftarrow \mathbf{0}$, $\forall h \in [H]$;

4 **for** *iteration* $t \leftarrow 1$ *to* $T$ **do**

5   **Auxiliary policy update:** for each step $h \in [H]$ and state $s \in \mathcal{S}$:

$$\hat{\mu}_h^t(\cdot|s) \leftarrow \underset{\hat{\mu} \in \Delta(\mathcal{A})}{\arg\max} \, \eta \left\langle \hat{\mu}, [Q_h^\tau \nu_h^{t-1}](s, \cdot) \right\rangle - D_R(\hat{\mu}, \hat{\mu}_h^{t-1}(\cdot|s));$$

$$\hat{\nu}_h^t(\cdot|s) \leftarrow \underset{\hat{\nu} \in \Delta(\mathcal{B})}{\arg\max} \, \eta \left\langle \hat{\nu}, [(\mu_h^{t-1})^\top Q_h^\tau](s, \cdot) \right\rangle - D_R(\hat{\nu}, \hat{\nu}_h^{t-1}(\cdot|s));$$

6   **Policy update:** for each step $h \in [H]$ and state $s \in \mathcal{S}$:

$$\mu_h^t(\cdot|s) \leftarrow \underset{\mu \in \Delta(\mathcal{A})}{\arg\max} \, \eta \left\langle \mu, [Q_h^\tau \nu_h^{t-1}](s, \cdot) \right\rangle - D_R(\mu, \hat{\mu}_h^t(\cdot|s));$$

$$\nu_h^t(\cdot|s) \leftarrow \underset{\nu \in \Delta(\mathcal{B})}{\arg\max} \, \eta \left\langle \nu, [(\mu_h^{t-1})^\top Q_h^\tau](s, \cdot) \right\rangle - D_R(\nu, \hat{\nu}_h^t(\cdot|s));$$

7   **if** $t - t_\tau^{start} + 1 \geq L_\tau$ **then**

8     $t_\tau^{\text{end}} \leftarrow t, t_{\tau+1}^{\text{start}} \leftarrow t+1, L_{\tau+1} \leftarrow \lfloor (1 + 1/H)L_\tau \rfloor$;

9     **Value update:** for each $h \in [H], s \in \mathcal{S}, a \in \mathcal{A}, b \in \mathcal{B}$:

$$Q_h^{\tau+1}(s, a, b) \leftarrow \frac{1}{L_\tau} \sum_{t'=t_\tau^{\text{start}}}^{t_\tau^{\text{end}}} \left( r_h + P_h[(\mu_{h+1}^{t'})^\top Q_{h+1}^\tau(\nu_{h+1}^{t'})] \right)(s, a, b);$$

10     $\tau \leftarrow \tau + 1; \mu_h^t = \hat{\mu}_h^t \leftarrow \tilde{\mu}_h^\tau, \nu_h^t = \hat{\nu}_h^t \leftarrow \tilde{\nu}_h^\tau, \forall h \in [H]$;

11 **Output policy:** $\bar{\mu}_h(\cdot|s) = \frac{1}{T}\sum_{t=1}^T \mu_h^t(\cdot|s)$ and $\bar{\nu}_h(\cdot|s) = \frac{1}{T}\sum_{t=1}^T \nu_h^t(\cdot|s), \forall s \in \mathcal{S}, h \in [H]$.

---

In order to establish convergence to (approximate) NE, we need to show that our optimistic OMD policy updates achieve "no regret" with respect to the value estimate sequence at each state, i.e., to upper bound (3). If we were to use the celebrated $\alpha_t = \frac{H+1}{H+t}$ learning rate [32] to update the value function estimates, we will inevitably need to show a no-weighted-regret guarantee for optimistic OMD, because such a time-varying learning rate assigns non-uniform weights to each history step. However, incorporating OMD with a dynamic learning rate is known to be challenging and can easily lead to linear regret [50]. While a stabilization technique [19] has been introduced to tackle this challenge, we take a different route by resorting to an alternative value update method, namely *stage-based* value updates [79]. Specifically, we divide the total $T$ iterations into multiple stages and only update our value estimates at the end of a stage (Line 9). We let the lengths of the stages grow exponentially at a rate of $(1 + 1/H)$ (Line 8) [79, 45]. The exponential growth ensures that the total $T$ iterations can be covered by a small number of stages, while the $(1 + 1/H)$ growth rate guarantees that the value estimation error does not blow up during the $H$ steps of recursion (Lemma 9). Compared with the incremental $\alpha_t = \frac{H+1}{H+t}$ update rule that modifies the value estimates at every step, stage-based updates are more stationary and allow us to assign uniform weights to each history step. This leads to a simpler no(-average)-regret problem [47] that can be easily addressed by (optimistic) OMD.

We introduce a few notations before presenting the convergence analysis of Algorithm 1. Let $\tau(t)$ denote the index of the stage that iteration $t$ belongs to. We denote by $\bar{\tau}$ the total number of stages, i.e., $\bar{\tau} := \tau(T)$. For any $(\tau, h, s) \in [\bar{\tau}] \times [H] \times \mathcal{S}$, define the per-state regrets for the max-player as

$$\text{reg}_{h,1}^\tau(s) := \max_{\mu_h^{\tau,\dagger}(\cdot|s) \in \Delta(\mathcal{A})} \frac{1}{L_\tau} \sum_{j=t_\tau^{\text{start}}}^{t_\tau^{\text{end}}} \left\langle \mu_h^{\tau,\dagger} - \mu_h^j, Q_h^\tau \nu_h^j \right\rangle(s). \tag{3}$$

The per-state regret $\text{reg}_{h,2}^\tau(s)$ for the min-player can be defined symmetrically (see (14) in Appendix B). We define the maximal regret (over the states and the two players) as $\text{reg}_h^\tau := \max_{s \in \mathcal{S}} \max_{i=1,2} \{\text{reg}_{h,i}^\tau(s)\}$. An upper bound for the per-state regrets is provided in Lemma 8 of Appendix B, which is useful in the analysis of Algorithm 1. We use the standard notion of

$$\text{NE-gap}(\mu, \nu) := V_1^{\dagger,\nu}(s_1) - V_1^{\mu,\dagger}(s_1)$$

to measure the optimality of a policy pair $(\mu, \nu)$. The initialization-dependent convergence rate of Algorithm 1 is as follows.

**Theorem 1.** *If Algorithm 1 is run on a two-player zero-sum Markov game for $T$ iterations with a learning rate $\eta \leq 1/(8H^2)$, the output policy pair $(\bar{\mu}, \bar{\nu})$ satisfies:*

$$\text{NE-gap}(\bar{\mu}, \bar{\nu}) \leq \frac{192H^3}{T} \sum_{h=1}^H \sum_{\tau=1}^{\bar{\tau}} \max_s \left( D_R(\mu_h^{\tau,\dagger}(\cdot|s), \tilde{\mu}_h^\tau(\cdot|s)) + D_R(\nu_h^{\tau,\dagger}(\cdot|s), \tilde{\nu}_h^\tau(\cdot|s)) \right).$$

*In addition, if the players' policies are initialized to be uniform policies, i.e., $\tilde{\mu}_h^\tau(\cdot|s) = \mathbf{1}/A$ and $\tilde{\nu}_h^\tau(\cdot|s) = \mathbf{1}/B, \forall s \in \mathcal{S}, \tau \in [\bar{\tau}], h \in [H]$, we further have*

$$\text{NE-gap}(\bar{\mu}, \bar{\nu}) \leq \frac{768H^5 \log T \log(AB)}{T}. \tag{4}$$

Compared to existing results [77, 72], Theorem 1 directly associates the convergence rate with the quality of the initial policy $(\tilde{\mu}, \tilde{\nu})$. Even when a good policy initialization is unknown and the algorithm is initialized with uniformly random policies, our convergence rate in (4) still matches the best-known result in the literature [72] except for an extra factor of $O(\log T)$. When suppressing the logarithmic terms, Theorem 1 immediately implies that for any $\varepsilon > 0$, Algorithm 1 takes no more than $T = \widetilde{O}(H^5/\varepsilon)$ steps to learn an $\varepsilon$-approximate NE in an individual zero-sum Markov game.

### 3.2 Sharper Convergence with Meta-Learning

Having settled the initialization-dependent convergence in a zero-sum game, we proceed to show how meta-learning can learn a set of related games collectively and more rapidly. We consider an online setting with a sequence of $K$ games $\mathcal{G} = (\mathbb{G}^1, \ldots, \mathbb{G}^K)$. For the max-player, let $\tilde{\mu}^k$ and $\bar{\mu}^k$, respectively, denote the initial policy and output policy of Algorithm 1 on game $\mathbb{G}^k$. By putting together $\mu_h^{\tau,\dagger}(\cdot|s)$ over all $(\tau, h, s) \in [\bar{\tau}] \times [H] \times \mathcal{S}$, we let $\mu^{k,\dagger} : [\bar{\tau}] \times [H] \times \mathcal{S} \to \Delta(\mathcal{A})$ denote the best fixed policies in hindsight on $\mathbb{G}^k$. Define $\tilde{\nu}^k, \bar{\nu}^k$ and $\nu^{k,\dagger}$ analogously for the min-player. Let $\mu^\star = \frac{1}{K} \sum_{k=1}^K \mu^{k,\dagger}$ and $\nu^\star = \frac{1}{K} \sum_{k=1}^K \nu^{k,\dagger}$ be the empirical averages of the best response policies. To ensure that the knowledge gained from previous games is useful for learning future tasks, we need to impose some similarity assumptions on the games $\mathcal{G}$. We consider the following similarity metric:

$$\Delta_{\mu,\nu} := \sum_{k=1}^K \left( \text{KL} \left( \mu^{k,\dagger} \| \mu^\star \right) + \text{KL} \left( \nu^{k,\dagger} \| \nu^\star \right) \right).$$

Intuitively, since $\{\nu^{k,t}\}_{t \in [T]}$ converges to an equilibrium policy for $\mathbb{G}^k$ when $T$ is large, the best fixed responses $\mu^{k,\dagger}$ can be considered as an approximation of the max-player's NE policy on $\mathbb{G}^k$. In this sense, $\Delta_{\mu,\nu}$ essentially measures the distances between the NE policies of different games. It considers a set of games $\mathcal{G}$ to be "similar" if their NE policies lie in a close neighborhood of each other. We remark that there might be multiple NE policies (with the same value) in a zero-sum game, and $\Delta_{\mu,\nu}$ only takes into account the NE policy pairs that Algorithm 1 actually delivers.

Our meta-learning procedure proceeds as follows: Within each game $\mathbb{G}^k$, we run Algorithm 1 as our base algorithm (1) to find a NE of $\mathbb{G}^k$. In a new game $\mathbb{G}^{k+1}$, the initial policy of Algorithm 1 is given

by the following meta-updates in the outer loop (2), which essentially averages the best response policies of the previous tasks under $\alpha$-greedy parameterization:

$$\tilde{\mu}^{k+1} = \frac{1}{k}\sum_{k'=1}^{k}[\mu^{k',\dagger}]_\alpha, \quad \text{and} \quad \tilde{\nu}^{k+1} = \frac{1}{k}\sum_{k'=1}^{k}[\nu^{k',\dagger}]_\alpha. \tag{5}$$

In particular, for any vector $\mathbf{x} \in \mathbb{R}^d$, we define its $\alpha$-greedy parameterization $[\mathbf{x}]_\alpha := (1-\alpha)\mathbf{x} + \frac{\alpha}{d}\mathbf{1}$ to be a weighted average with a uniform vector $\mathbf{1}/d \in \mathbb{R}^d$ of a proper dimension, where $\alpha \in (0, 1/2)$. Since $\mu^{k,\dagger}$ denotes a set of vectors, we apply the operator $[\cdot]_\alpha$ element-wise to each of the vectors. The reason for using $\alpha$-greedy is mainly technical: $\mathrm{KL}\,(\cdot\|\cdot)$ is not Lipschitz continuous near the boundary of the probability simplex, and $\alpha$-greedy parameterization helps to stay $\alpha$-distance away from the boundary. We are now ready to present our sharper convergence rates for meta-learning.

**Theorem 2.** *In a sequence of $K$ two-player zero-sum Markov games, if Algorithm 1 is run for $T$ iterations as the base algorithm and (5) with $\alpha = 1/\sqrt{K}$ as the meta-updates, we have*

$$\frac{1}{K}\sum_{k=1}^{K}\mathrm{NE\text{-}gap}(\bar{\mu}^k, \bar{\nu}^k) \leq \frac{192H^5}{T}\left(\frac{\Delta_{\mu,\nu}}{KH^2} + \frac{10(A+B)\log K}{\sqrt{K}H^2} + \frac{16\log T\log(ABK)}{\sqrt{K}}\right). \tag{6}$$

*Consequently, for any $\varepsilon > 0$, $T = \tilde{O}(\frac{H^3}{\varepsilon}(\frac{\Delta_{\mu,\nu}}{K} + \frac{A+B+H^2}{\sqrt{K}}))$ steps on average suffice to find an $\varepsilon$-approximate Nash equilibrium in each game.*

When the number of games $K$ is large, the last two terms on the RHS of (6) become negligible. Hence, compared to the best-known results $\widetilde{O}(H^5/T)$ of learning each game individually, Theorem 2 implies a significantly sharper convergence rate when the games are similar, i.e., when $\Delta_{\mu,\nu} \ll KH^2$.

## 4 Meta-Learning for Markov Potential Games

In this section, we study meta-learning for NE in Markov potential games. We show that a straight-forward refinement to the analysis of an existing algorithm [15] provides initialization-dependent bounds. Building on it, in Section 4.1, we first investigate the sharper convergence of meta-learning in a sequence of similar MPGs. Further, since there exists an optimization objective universally agreed on by all the players in an MPG (i.e., the potential function), we can formulate the meta-learning problem in the same way as MAML [20]. In Section 4.2, by choosing a proper base algorithm, we establish the non-asymptotic convergence of MAML in the highly non-stationary multi-agent scenario, without even imposing any smoothness assumptions as in existing works [17, 18, 30].

### 4.1 Sharper Rates in Similar Games

To be consistent with existing results in the literature, in this section, we consider an infinite-horizon $\gamma$-discounted reward setting for MPGs [43, 39, 76, 15]. A detailed description of the setup is provided in Appendix C for completeness. Equivalent results for the finite-horizon episodic setting (as we defined in Section 2) can be derived in a straightforward way. We choose an existing state-of-the-art algorithm, namely independent projected Q-descent [15], as our base algorithm (1). Specifically, in an MPG $\mathbb{G}^k$, each agent independently runs policy gradient ascents to update its own policy for $T$ iterations:

$$\pi_i^{k,t}(\cdot|s) \leftarrow \mathrm{Proj}_{\Delta(\mathcal{A}_i)}\left(\pi_i^{k,t-1}(\cdot|s) + \alpha\bar{Q}_i^{\pi^{k,t-1}}(s,\cdot)\right), \forall t \in [T], \tag{7}$$

where $\bar{Q}_i^\pi$ is the "averaged" Q-function formally defined in Appendix C. Let $\Phi(\cdot\,; \mathbb{G}^k)$ denote the potential function of $\mathbb{G}^k$. Through a simple refinement of the analysis in [15], we can establish the following initialization-dependence bound for our base algorithm (7).

**Proposition 1.** *(Theorem 1 in [15]) Suppose that all players in a Markov potential game $\mathbb{G}^k$ run independent projected Q-descent (7) for $T$ iterations with $\alpha \leq \frac{(1-\gamma)^4}{8\kappa^3 N A_{\max}}$. Then, we have*

$$\frac{1}{T}\sum_{t=0}^{T-1}\mathrm{NE\text{-}gap}(\pi^{k,t}) \leq \sqrt{\frac{\kappa(\mathbb{G}^k)(\Phi(\pi^{k,T}; \mathbb{G}^k) - \Phi(\pi^{k,0}, \mathbb{G}^k))}{\alpha T(1-\gamma)^2}},$$

*where $\kappa(\mathbb{G}^k)$ is the standard distribution mismatch coefficient for $\mathbb{G}^k$ formally defined in Appendix C.*

Proposition 1 immediately implies that if we learn each MPG individually, it takes $T = O\left(\frac{NA_{\max}\kappa^4\Phi_{\max}}{(1-\gamma)^6\varepsilon^2}\right)$ steps to find an $\varepsilon$-approximate NE. To show the effectiveness of meta-learning, we consider the following similarity metric for a sequence of $K$ games, which measures the maximal point-wise deviations of the potential functions:

$$\Delta_\Phi := \sum_{k=1}^{K-1} \max_\pi \left(\Phi(\pi;\mathbb{G}^k) - \Phi(\pi;\mathbb{G}^{k+1})\right). \tag{8}$$

As for the meta-updates, we simply instantiate (2) as $\pi_i^{k,0} \leftarrow \pi_i^{k-1,T}$, which lets each agent play the converged policy in the previous game. The intuition is that after running $T$ steps on $\mathbb{G}^{k-1}$, the agents will converge to an approximate NE policy of $\mathbb{G}^{k-1}$. Since (8) requires the potential functions to be close, the converged policy $\pi^{k-1,T}$ should serve as a good starting point to search for NE in $\mathbb{G}^k$. We formally characterize such an intuition in the following theorem, which shows the sharper convergence of meta-learning in a large set of similar MPGs (i.e., when $K$ is large and $\Delta_\Phi$ is small):

**Theorem 3.** *In a sequence of $K$ Markov potential games, if (7) is run for $T$ iterations as the base algorithm and $\pi_i^{k,0} \leftarrow \pi_i^{k-1,T}$ as the meta-updates, then, for any $\varepsilon > 0$, $T = O\left(\frac{NA_{\max}\kappa^4(\Phi_{\max}+\Delta_\Phi)}{K(1-\gamma)^6\varepsilon^2}\right)$ steps on average suffice to find an $\varepsilon$-approximate Nash equilibrium in each game.*

## 4.2 Convergence to MAML Objective

In this subsection, we study meta-learning for MPGs under exactly the same formulation as in the seminal work of MAML [20]. Let $\mathcal{G} = \{\mathbb{G}^j\}$ be a set of different MPGs, where the games are now drawn from a fixed distribution $p$ that we can sample from. We consider parametric policy classes where agent $i$'s policy is parameterized by $\theta_i = \{\theta_i(a_i|s) \in \mathbb{R}\}_{s\in\mathcal{S},a_i\in\mathcal{A}_i}$. We focus on softmax parameterization where

$$\pi_{\theta_i}(a_i|s) = \frac{\exp(\theta_i(a_i|s))}{\sum_{a'_i\in\mathcal{A}_i}\exp(\theta_i(a'_i|s))}.$$

Let $\zeta(\,\cdot\,;\mathbb{G})$ denote the operator of performing one step of policy gradient update on game $\mathbb{G}$, i.e., $\zeta(\theta;\mathbb{G}) := \theta + \alpha\nabla\Phi(\theta;\mathbb{G})$, where $\alpha > 0$ is the learning rate. The $T$-step MAML objective [20, 18, 30] can be formulated as

$$\max_{\theta\in\Theta} F_T(\theta) := \mathbb{E}_{\mathbb{G}\sim p(\mathcal{G})}\left[\Phi\left(\zeta(\ldots(\zeta(\theta;\mathbb{G}))\ldots);\mathbb{G}\right)\right], \tag{9}$$

where $\theta = (\theta_1,\ldots,\theta_N) \in \Theta$, and the operator $\zeta(\,\cdot\,;\mathbb{G})$ is applied $T$ times. Intuitively, MAML tries to find a good parameter initialization from which running $T$ steps of gradient ascents on any new task $\mathbb{G}$ leads to well-performing policy parameters.

Similar to Section 2, the MAML procedure consists of two nested stages. For the inner stage (1), we let each agent independently run $T$ steps of policy gradient ascents to update its policy parameter $\theta_i^{(t)}$ on each encountered MPG. It is known (Theorem 5 of [78]) that $T = O(1/\varepsilon^2)$ steps will find an $\varepsilon$-approximate NE for each individual MPG. For the outer stage (2), MAML directly performs gradient ascents with respect to the meta-objective (9). The gradient of $F_T$ can be written in closed-form as

$$\nabla F_T(\theta) = \mathbb{E}_{\mathbb{G}\sim p(\mathcal{G})}\left[\left(\prod_{t=0}^{T-1}\left(I + \alpha\nabla^2\Phi(\theta^{(t)};\mathbb{G})\right)\right)\nabla\Phi(\theta^{(T)};\mathbb{G})\right]. \tag{10}$$

A detailed discussion of MAML and its instantiation in our problem are provided in Appendix D. Most importantly, Appendix D shows that both the policy gradient $\nabla\Phi(\theta)$ and the policy Hessian $\nabla^2\Phi(\theta)$ can be written in closed-form, which allows us to construct unbiased estimators of (10) from samples. Despite the fact that the learning agents update their policies independently in an intertwined multi-agent system, our next result shows that the MAML updates converge to a stationary point of the meta-objective (9) in a non-asymptotic manner. A key step of the proof is to prove (rather than assume, as in existing works [17, 30]) that the meta-objective is Lipschitz smooth in the policy parameter $\theta$. The smoothness constant can also be written in a closed form (Lemma 14).

**Theorem 4.** *Suppose that the agents run independent policy gradient ascents with softmax parameterization on each encountered MPG as the inner stage, and perform gradient ascents w.r.t the MAML objective as the outer stage. For any $\varepsilon > 0$, $K = \frac{4NL_F}{(1-\gamma)\varepsilon^2}$ iterations of MAML updates can find a policy $\theta^\star$ such that $\|\nabla F_T(\theta^\star)\| \leq \varepsilon$, where $L_F$ is given in Lemma 14 of Appendix D.*

# 5 Meta-Learning for General-Sum Markov Games

In this section, we consider learning coarse correlated equilibria in general-sum Markov games with no assumption on reward structures. Similar to Section 3, we start by developing an initialization-dependent algorithm, followed by investigating the sharper convergence of meta-learning.

Our base algorithm for learning CCE also uses optimistic OMD with stage-based value updates. Detailed descriptions are deferred to Algorithm 2 in Appendix E due to space limitations. Algorithm 2 follows a similar structure as Algorithm 1, but the output policy $\bar{\pi}$ of Algorithm 2 is no longer a state-wise average policy and is instead a correlated policy [3, 44, 33]. For any correlated policy $\pi$, we use the notion

$$\text{CCE-gap}(\pi) := \max_{i \in \mathcal{N}} V_{1,i}^{\dagger, \pi^{-i}}(s_1) - V_{1,i}^{\pi}(s_1)$$

to measure its distance to a CCE. Let $\bar{\tau}$ denote the total number of stages of Algorithm 2. Similar to zero-sum games (3), for any $(\tau, h, s) \in [\bar{\tau}] \times [H] \times \mathcal{S}$, we define the per-state regret for each player $i \in \mathcal{N}$ as

$$\text{reg}_{h,i}^{\tau}(s) := \max_{\pi_{h,i}^{\tau,\dagger}(\cdot|s) \in \Delta(\mathcal{A}_i)} \frac{1}{L_{\tau}} \sum_{j=t_{\tau}^{\text{start}}}^{t_{\tau}^{\text{end}}} \left\langle \pi_{h,i}^{\tau,\dagger} - \pi_{h,i}^{j}, Q_{h,i}^{\tau} \pi_{h,-i}^{j} \right\rangle (s),$$

where $Q_{h,i}^{\tau}$ is player $i$'s Q-function estimate at stage $\tau$. We define the maximal regret (over all states and players) as $\text{reg}_h^{\tau} := \max_{s \in \mathcal{S}} \max_{i \in \mathcal{N}} \{\text{reg}_{h,i}^{\tau}(s)\}$. The initialization-dependent convergence rate of Algorithm 2 is established in the following theorem.

**Theorem 5.** *If Algorithm 2 is run on a general-sum Markov game for $T$ iterations with a learning rate $\eta > 0$, the output policy $\bar{\pi}$ satisfies:*

$$\text{CCE-gap}(\bar{\pi}) \leq \frac{3}{\eta T} \sum_{\tau=1}^{\bar{\tau}} \sum_{h=1}^{H} \max_{i \in \mathcal{N}, s \in \mathcal{S}} D_R(\pi_{h,i}^{\tau,\dagger}(\cdot|s), \tilde{\pi}_{h,i}^{\tau}(\cdot|s)) + 36N^2 \eta^2 H^4.$$

*In addition, if the players' policies are initialized to be uniform policies $\tilde{\pi}_{h,i}^{\tau}(\cdot|s) = \mathbf{1}/A_i, \forall i \in \mathcal{N}$ and $\eta$ is chosen as $\eta = H^{-2/3} T^{-1/3} (N-1)^{-2/3}$, then we have*

$$\text{CCE-gap}(\bar{\pi}) \leq \frac{12 N^{\frac{2}{3}} H^{\frac{8}{3}} \log T \log A_{\max}}{T^{\frac{2}{3}}}. \tag{11}$$

Compared to existing results, Theorem 5 directly associates the convergence rate with the quality of the initial policy $\tilde{\pi}$. With uniform initialization, the convergence rate in (11) has a slightly worse dependence on $T$ than the best known result $\widetilde{O}(\sqrt{N} H^{11/4}/T^{3/4})$ [77]. Such deterioration is due to the potential lack of a smoothness condition for optimistic OMD that directly connects the stability of policies to the stability of utility functions (Lemma 18), unlike in optimistic FTRL. Although we believe that our rate in (11) can almost certainly be improved via a refined stability analysis, we leave the tightening of it to our future work as it would be a departure from the main focus of this work.

Let $\tilde{\pi}^k$ and $\bar{\pi}^k$, respectively, denote the initial policy and output policy of Algorithm 2 on game $\mathbb{G}^k$. For player $i \in \mathcal{N}$, by putting together $\pi_{h,i}^{\tau,\dagger}(\cdot|s)$ over all $(\tau, h, s)$, we use $\pi_i^{k,\dagger} : [\bar{\tau}] \times [H] \times \mathcal{S} \to \Delta(\mathcal{A}_i)$ to denote the best fixed policies in hindsight on $\mathbb{G}^k$. We consider a game similarity metric defined as

$$\Delta_{\pi} := \sum_{k=1}^{K} \sum_{i=1}^{N} \text{KL}(\pi_i^{k,\dagger} \| \pi_i^{\star}), \text{ where } \pi_i^{\star} = \frac{1}{K} \sum_{k=1}^{K} \pi_i^{k,\dagger}.$$

The following theorem presents the convergence rate of meta-learning, which again is sharper than learning each game individually when the games are similar, i.e., when $\Delta_{\pi}$ is sufficiently small.

**Theorem 6.** *In a sequence of $K$ general-sum Markov games, if Algorithm 2 is run for $T$ iterations as the base algorithm and the meta-updates $\tilde{\pi}_i^k = \frac{1}{k-1} \sum_{k'=1}^{k-1} [\pi_i^{k',\dagger}]_{\alpha}, \forall i \in \mathcal{N}$ are used with $\alpha = 1/\sqrt{K}$ for policy initializations, then, for any $\varepsilon > 0$, $T = \tilde{O}\big(\frac{HN}{\varepsilon^{3/2}} \big(\frac{\Delta_{\pi}^{3/2}}{K^{5/4}} + \frac{A_{\max}^{3/2} + H^3}{K^{1/2}}\big)\big)$ steps on average suffice to find an $\varepsilon$-approximate CCE in each game.*

# 6 Simulations

We numerically evaluate our meta-learning algorithms from Sections 3 and 4 on a sequence of $K$ games. In this section, we evaluate on a sequence of $K = 10$ zero-sum Markov games and Markov potential games with two states, two players, and two candidate actions for each player. In Appendix F, we further demonstrate the scalability of our methods by providing numerical results on larger-scale tasks, including a simplified version of the Poker endgame considered in [27] and a 1D linear-quadratic tracking problem [37] with 4 cooperative players.

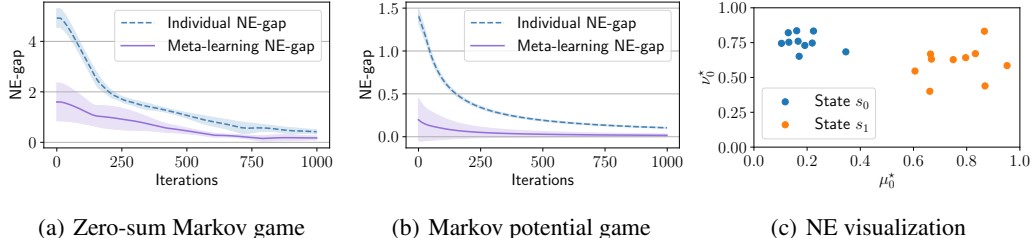

| (a) Zero-sum Markov game | (b) Markov potential game | (c) NE visualization |

Figure 1: NE-gap of policies output by individual learning and meta-learning in (a) zero-sum Markov games, and (b) Markov potential games. Shaded areas denote the standard deviations. (c) visualizes the NE policies of the $K$ games in the normalized space $[0, 1] \times [0, 1]$ to illustrate their closeness.

We generate the $K = 10$ games by first specifying a "base game" and then adding random perturbations to its reward function to get $K$ slightly different games. Each of the $K$ games is run for $T = 1000$ iterations. To better visualize the similarity level of these games, in Figure 1(c), we plot the NE policies of the perturbed zero-sum matrix games at each of the two states for the $K = 10$ games. We remark that due to the existence of state transitions, the NE policies with respect to the stage Q-functions can be more diversified than Figure 1(c). Detailed descriptions of the simulation setup are deferred to Appendix F.

We evaluate the convergences of the algorithms in terms of NE-gap. Figures 1(a) and 1(b) compare the average NE-gap over the $K$ games between individual learning and meta-learning for zero-sum Markov games and Markov potential games, respectively. We see that meta-learning can utilize knowledge from previous tasks to attain better policy initialization in a new task and converges to an approximate NE policy using much fewer iterations.

# 7 Concluding Remarks

In this paper, we have introduced meta-learning to solve multiple MARL tasks collectively. Under natural similarity metrics, we have shown that meta-learning achieves provably sharper convergence for learning NE in zero-sum and potential games and for learning CCE in general-sum games. Along the way, we have proposed new MARL algorithms with fine-grained initialization-dependent convergence guarantees. Our work appears to be the first to investigate the theoretical properties of meta-learning in MARL and provide reliable justifications for its usage. As for the limitations, our convergence rate for learning CCE (Theorem 5) is slightly less competitive than the best-known results when our policies are initialized conservatively, which might be improved via a refined policy stability analysis. Other future directions include further generalization of our results to alternative game similarity metrics and broader types of games (e.g., stochastic Stackelberg games).

## Acknowledgments and Disclosure of Funding

This work was partially supported by the National Science Foundation (NSF) under grant CCF 20-29049; by the IBM-ILLINOIS Center for Cognitive Computing Systems Research (C3SR), a research collaboration that is part of the IBM AI Horizon Network; by the IBM-ILLINOIS Discovery Accelerator Institute (IIDAI); and by the Air Force Office of Scientific Research (AFOSR) under grant FA9550-19-1-0353. Any opinions, findings or recommendations expressed in this material are those of the authors and do not necessarily reflect the views of the NSF or IBM or AFOSR.

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
