# Supplementary Materials for "Multi-Agent Meta-Reinforcement Learning"

## A  Technical Lemmas

**Lemma 1.** *Let $x, y \in \mathbb{R}^d$ be two probability distributions lying in the $d$-dimensional simplex for $d \geq 2$. For $\alpha \in (0, 1/2)$, let $[x]_\alpha = (1-\alpha)x + \frac{\alpha}{d}\mathbf{1}$ denote a weighted average between $x$ and a uniform vector $\mathbf{1}/d \in \mathbb{R}^d$ of a proper dimension. Denote by $\mathrm{KL}(x\|y)$ the Kullback–Leibler divergence between $x$ and $y$. If $y_i \geq \alpha/d, \forall i \in [d]$, then we have*

$$\mathrm{KL}(x\|y) \leq \mathrm{KL}(\tilde{x}\|y) + 4\alpha \ln \frac{d}{\alpha}.$$

*Proof.* From the three-points identity of the Bregman divergence (Lemma 3.1 of [9]),

$$\mathrm{KL}(x\|y) - \mathrm{KL}(\tilde{x}\|y) = \mathrm{KL}(x\|\tilde{x}) + \langle \ln\tilde{x} - \ln y, x - \tilde{x} \rangle \tag{12}$$

The first term in (12) can be bounded by

$$\mathrm{KL}(x\|\tilde{x}) = \sum_{i=1}^d x_i \ln \frac{x_i}{\tilde{x}_i} = \sum_{i=1}^d x_i \ln \frac{x_i}{(1-\alpha)x_i + \frac{\alpha}{d}} \leq \sum_{i=1}^d x_i \ln \frac{1}{1-\alpha} \leq \ln \frac{1}{1-\alpha}.$$

By the Hölder's inequality, the second term in (12) is bounded as

$$\langle \ln\tilde{x} - \ln y, x - \tilde{x} \rangle \leq \|\ln\tilde{x} - \ln y\|_\infty \|x - \tilde{x}\|_1. \tag{13}$$

We handle the two terms in (13) separately. First,

$$\|\ln\tilde{x} - \ln y\|_\infty = \sup_{i \in [d]} \left| \ln \frac{\tilde{x}_i}{y_i} \right| \leq \sup_{i \in [d]} \max \left\{ \ln \frac{\tilde{x}_i}{y_i}, \ln \frac{y_i}{\tilde{x}_i} \right\} \leq \ln \frac{1 - \alpha + \frac{\alpha}{d}}{\alpha/d} \leq \ln \frac{d}{\alpha},$$

where the second to last step uses the facts that $\alpha/d \leq \tilde{x}_i \leq 1$ and $\alpha/d \leq y_i \leq 1, \forall i \in [d]$. The last step is simply due to the fact that $d \geq 1$. To bound the second term in (13), notice that

$$\|x - \tilde{x}\|_1 = \|x - (1-\alpha)x - \alpha\mathbf{1}/d\|_1 = \alpha \|x - \mathbf{1}/d\|_1 \leq 2\alpha.$$

Putting everything together, (12) can be bounded by

$$\mathrm{KL}(x\|\tilde{x}) + \langle \ln\tilde{x} - \ln y, x - \tilde{x} \rangle \leq \ln \frac{1}{1-\alpha} + 2\alpha \frac{d}{\alpha} \leq \alpha^2 + \alpha + 2\alpha \ln \frac{d}{\alpha} \leq 4\alpha \ln \frac{d}{\alpha},$$

where the second to last step is derived using the Taylor expansion, and the last step holds due to the assumptions that $\alpha \in (0, 1/2)$ and $d \geq 2$. This completes the proof of the lemma. $\square$

**Lemma 2.** *(Proposition B.1 of [35]) Let $R : \Theta \to \mathbb{R}$ be 1-strongly convex with respect to $\|\cdot\|$ and consider any $\theta_1, \ldots, \theta_K \in \Theta$. Then, when run on the loss sequence $\alpha_1 D_R(\theta_1, \dot{)}, \ldots, \alpha_K D_R(\theta_K, \dot{)}$ for any positive scalars $\alpha_1, \ldots, \alpha_K \in \mathbb{R}_+$, the follow-the-leader (FTL) algorithm obtains regret*

$$\mathrm{reg}_K \leq 2CD \sum_{k=1}^K \frac{\alpha_k^2 G_k}{\alpha_k + 2\sum_{k'=1}^{k-1} \alpha_{k'}},$$

*for $C$ such that $\|\theta\| \leq C\|\theta\|_2, \forall\theta \in \Theta$, $D = \max_{\theta, \theta' \in \Theta} \|\theta - \theta'\|_2$ the L2 diameter of $\Theta$, and $G_k$ the Lipschitz constant of $D_R(\theta_k, \cdot)$ over $\Theta$ with respect to $\|\cdot\|$.*

**Lemma 3.** *(Lemma 2 of [18]) For any $i \in \{1, \ldots, n\}$, let $f_i : \mathbb{R}^d \to W_i$ be a continuous function with $W_i \in \{\mathbb{R}, \mathbb{R}^d, \mathbb{R}^{1 \times d}, \mathbb{R}^{d \times d}\}$ such that $g(\theta) = f_n(\theta) \ldots f_1(\theta)$ is well-defined. Suppose $f_i$ is $B_i$-bounded and $L_i$-Lipschitz, i.e., $\|f_i(\theta)\| \leq B_i$ and $\|f_i(\theta) - f_i(\theta')\| \leq L_i \|\theta - \theta'\|, \forall\theta, \theta' \in \mathbb{R}^d$ for some non-negative constants $B_i$ and $L_i$. Then, $g(\theta)$ is Lipschitz with constant $L_g = \sum_{i=1}^n (L_i \prod_{j \neq i} B_j)$, i.e., $\|g(\theta) - g(\theta')\| \leq L_g \|\theta - \theta'\|, \forall\theta, \theta' \in \mathbb{R}^d$.*

**Lemma 4.** *(Lemma 3 of [18]) For any $i \in \{1, \ldots, n\}$, let $f_i : \mathbb{R}^d \to \mathbb{R}^m$ be a continuously differentiable function that is $B_f$-bounded and $L_f$-Lipschitz continuous. Let $p(\cdot; \theta)$ be a distribution on $\{f_i\}_{i=1}^n$ where the probability of drawing $f_i$ is $p(i; \theta)$. Suppose there exists a non-negative constant $B_p$ such that $\|\nabla_\theta \log p(i; \theta)\| \leq B_p$ for any $i$ and $\theta$. Then, the function $g(\theta) = \mathbb{E}_{p(i;\theta)}[f(i; \theta)]$ is Lipschitz continuous with constant $B_f B_p + L_f$.*

**Lemma 5.** *Consider a block diagonal matrix $C$ that is a square matrix such that the main-diagonal consists of $N$ block matrices $A_1 \in \mathbb{R}^{d_1 \times d_1}, \ldots, A_N \in \mathbb{R}^{d_N \times d_N}$ and all off-diagonal blocks are zero matrices. Then, it holds that $\|C\| \leq \max_{1 \leq i \leq N} \|A_i\|$.*

*Proof.* We prove the lemma via induction on $N$. For the induction basis $N = 2$, we need to show

$$\|C\| = \left\| \begin{bmatrix} A_1 & \mathbf{0} \\ \mathbf{0} & A_2 \end{bmatrix} \right\| \leq \max\{\|A_1\|, \|A_2\|\}.$$

To see this, let $x \in \mathbb{R}^{d_1}$ and $y \in \mathbb{R}^{d_2}$ be such that $\left\| \begin{bmatrix} x \\ y \end{bmatrix} \right\|^2 = \|x\|^2 + \|y\|^2 = 1$. Then, by the definition of the matrix norm,

$$\left\| C \begin{bmatrix} x \\ y \end{bmatrix} \right\|^2 = \|A_1 x\|^2 + \|A_2 y\|^2 \leq \|A_1\|^2 \|x\|^2 + \|A_2\|^2 \|y\|^2 \leq \max\{\|A_1\|^2, \|A_2\|^2\},$$

where the last step uses the fact that $\|x\|^2 + \|y\|^2 = 1$. This completes the proof of the induction basis $N = 2$. Now, suppose that the lemma holds for $N = k-1$. We next show that it also holds for $N = k$.

Let $C = \begin{bmatrix} A_1 & \mathbf{0} & \ldots & \mathbf{0} \\ \mathbf{0} & A_2 & \ldots & \mathbf{0} \\ \vdots & \vdots & \ddots & \vdots \\ \mathbf{0} & \mathbf{0} & \ldots & A_k \end{bmatrix}$. Note that we can rewrite the matrix as $C = \begin{bmatrix} C_{k-1} & \mathbf{0} \\ \mathbf{0} & A_k \end{bmatrix}$, where

$C_{k-1} = \begin{bmatrix} A_1 & \ldots & \mathbf{0} \\ \vdots & \ddots & \vdots \\ \mathbf{0} & \ldots & A_{k-1} \end{bmatrix}$ is a block diagonal matrix consisting of $k-1$ matrices. Invoking the

induction hypothesis for $N = k-1$, we know that $\|C_{k-1}\| \leq \max_{1 \leq i \leq k-1} \|A_i\|$. Finally, using the induction hypothesis for $N = 2$, we conclude that $\|C\| \leq \max\{\|C_{k-1}\|, \|A_k\|\} \leq \max_{1 \leq i \leq k} \|A_i\|$. This completes the induction proof. $\square$

**Lemma 6.** *Consider a block matrix $A(\theta)$ with $N \times N$ blocks parameterized by $\theta \in \mathbb{R}^d$:*

$$A(\theta) = \begin{bmatrix} A_{1,1}(\theta) & \ldots & A_{1,N}(\theta) \\ \vdots & \ddots & \vdots \\ A_{N,1}(\theta) & \ldots & A_{N,N}(\theta) \end{bmatrix},$$

*where $A_{i,j}(\theta) \in \mathbb{R}^{d_i \times d_j}, \forall 1 \leq i, j \leq N$ and $d = \sum_{i=1}^N d_i$. Suppose that the norm of each matrix block is Lipschitz continuous with respect to $\theta$, i.e., $\|A_{i,j}(\theta) - A_{i,j}(\theta')\| \leq L_{i,j} \|\theta - \theta'\|, \forall \theta, \theta' \in \mathbb{R}^d, 1 \leq i, j \leq N$. Let $L = \max\{L_{i,j} : 1 \leq i, j \leq N\}$. Then, the norm of $A(\theta)$ is also Lipschitz, i.e.,*

$$\|A(\theta) - A(\theta')\| \leq NL \|\theta - \theta'\|, \forall \theta, \theta' \in \mathbb{R}^d.$$

*Proof.* Let $x \in \mathbb{R}^d$ be a vector such that $x = \begin{bmatrix} x_1^\top & x_2^\top & \cdots & x_N^\top \end{bmatrix}^\top$ and $\|x\|^2 = \sum_{i=1}^N \|x_i\|^2 = 1$, where $x_i \in \mathbb{R}^{d_i}, \forall 1 \le i \le N$. We have

$$
\|(A(\theta) - A(\theta'))x\|^2 = \left\| \begin{bmatrix} \sum_{j=1}^N (A_{1,j}(\theta) - A_{1,j}(\theta')) x_j \\ \vdots \\ \sum_{j=1}^N (A_{N,j}(\theta) - A_{N,j}(\theta')) x_j \end{bmatrix} \right\|^2
$$

$$
= \sum_{i=1}^N \left\| \sum_{j=1}^N (A_{i,j}(\theta) - A_{i,j}(\theta')) x_j \right\|^2
$$

$$
\le N \sum_{i=1}^N \sum_{j=1}^N \|(A_{i,j}(\theta) - A_{i,j}(\theta')) x_j\|^2
$$

$$
\le N \sum_{i=1}^N \sum_{j=1}^N \|A_{i,j}(\theta) - A_{i,j}(\theta')\|^2 \|x_j\|^2,
$$

where the first inequality follows from the Cauchy-Schwarz inequality, and the last step is due to the definition of the matrix norm. Applying the Lipschitz continuity of each matrix block $\|A_{i,j}(\theta) - A_{i,j}(\theta')\| \le L_{i,j} \|\theta - \theta'\|$ yields

$$
\|(A(\theta) - A(\theta'))x\|^2 \le N \sum_{i=1}^N \sum_{j=1}^N \|A_{i,j}(\theta) - A_{i,j}(\theta')\|^2 \|x_j\|^2
$$

$$
\le N \sum_{i=1}^N \sum_{j=1}^N L_{i,j}^2 \|\theta - \theta'\|^2 \|x_j\|^2
$$

$$
\le N^2 L^2 \|\theta - \theta'\|^2,
$$

where the last step uses the facts that $L_{i,j} \le L, \forall 1 \le i, j \le N$ and $\sum_{j=1}^N \|x_j\|^2 = 1$. Since the above condition holds for any vector $x$ with $\|x\| = 1$, we know from the definition of the matrix norm that

$$
\|A(\theta) - A(\theta')\| \le NL \|\theta - \theta'\|, \forall \theta, \theta' \in \mathbb{R}^d.
$$

This concludes the proof for the Lipschitz continuity of $A(\theta)$. $\square$

# B  Proofs for Section 3

## B.1  Proof of Theorem 1

We introduce one more notation before presenting the proof. For each iteration $t \in [T]$ and step $h \in [H]$, define the Q-function estimation error as

$$
\delta_h^t := \|Q_h^{\tau(t)} - Q_h^\star\|_\infty.
$$

Note that since Algorithm 1 performs stage-based value updates, the value estimation error $\delta_h^t$ does not change within a stage $\tau(t)$; that is, $\delta_h^t$ takes the same value for all $t \in [t_\tau^{\text{start}}, t_\tau^{\text{end}}]$. For this reason, we will sometimes abuse the notation and simply use $\delta_h^\tau$ to denote the estimation error for a stage $\tau$. In the rest of this paper, we will write $\delta_h^\tau$ and $\delta_h^t$ interchangeably since one of them will be more convenient than the other in certain contexts.

Further, recall that for any $(\tau, h, s) \in [\bar{\tau}] \times [H] \times \mathcal{S}$, the per-state regrets for the two players are defined as

$$
\text{reg}_{h,1}^\tau(s) := \max_{\mu_h^{\tau,\dagger} \in \Delta(\mathcal{A})} \frac{1}{L_\tau} \sum_{j=t_\tau^{\text{start}}}^{t_\tau^{\text{end}}} \left\langle \mu_h^{\tau,\dagger} - \mu_h^j, Q_h^\tau \nu_h^j \right\rangle(s),
$$

$$
\text{reg}_{h,2}^\tau(s) := \max_{\nu_h^{\tau,\dagger} \in \Delta(\mathcal{B})} \frac{1}{L_\tau} \sum_{j=t_\tau^{\text{start}}}^{t_\tau^{\text{end}}} \left\langle \nu_h^j - \nu_h^{\tau,\dagger}, (Q_h^\tau)^\top \mu_h^j \right\rangle(s). \tag{14}
$$

Note that the best response policies $\mu_h^{\tau,\dagger}(\cdot|s)$ and $\nu_h^{\tau,\dagger}(\cdot|s)$ should be state-dependent, but we will oftentimes omit the dependence on $s$ for notational convenience. This leads us to the initialization-dependent convergence rate of Algorithm 1, which we re-state and prove as follows.

**Theorem 1.** If we run Algorithm 1 on a two-player zero-sum Markov game for $T$ iterations with a learning rate $\eta \leq 1/(8H^2)$, the output policy pair $(\bar{\mu}, \bar{\nu})$ satisfies:

$$\text{NE-gap}(\bar{\mu}, \bar{\nu}) \leq \frac{192H^3}{T} \sum_{h=1}^{H} \sum_{\tau=1}^{\bar{\tau}} \max_s \left( D_R(\mu_h^{\tau,\dagger}(\cdot|s), \tilde{\mu}_h^{\tau}(\cdot|s)) + D_R(\nu_h^{\tau,\dagger}(\cdot|s), \tilde{\nu}_h^{\tau}(\cdot|s)) \right).$$

In addition, if we initialize the players' policies to be uniform policies, i.e., $\tilde{\mu}_h^{\tau}(\cdot|s) = \mathbf{1}/A$ and $\tilde{\nu}_h^{\tau}(\cdot|s) = \mathbf{1}/B, \forall s \in \mathcal{S}, \tau \in [\bar{\tau}], h \in [H]$, we further have

$$\text{NE-gap}(\bar{\mu}, \bar{\nu}) \leq \frac{768H^5 \log T \log(AB)}{T}.$$

*Proof.* The proof of the theorem follows from a series of lemmas, which we state and prove in the next few subsections. In particular, we first show in Lemma 7 that upper bounding the NE-gap breaks down to controlling the per-state regrets $\text{reg}_{h,1}^{\tau}(s) + \text{reg}_{h,2}^{\tau}(s)$ and the value estimation errors $\delta_h^{\tau}$, in a similar fashion as in the analysis of [72]. For this purpose, Lemma 8 provides an upper bound on the per-state regrets, while Lemma 9 and Lemma 10 together bound the value estimation error via a recursive argument. The rest of the proof follows by putting all the aforementioned results together.

Specifically, for $\eta \leq 1/(8H^2)$, by plugging in the results of Lemma 8 and Lemma 9 to Lemma 7, we obtain that

$$\text{NE-gap}(\bar{\mu}, \bar{\nu}) \leq \frac{2}{T} \sum_{h=1}^{H} \sum_{\tau=1}^{\bar{\tau}} L_\tau \max_{s \in \mathcal{S}} \left( \text{reg}_{h,1}^{\tau}(s) + \text{reg}_{h,2}^{\tau}(s) \right) + \frac{2}{T} \sum_{h=1}^{H} \sum_{\tau=1}^{\bar{\tau}} L_\tau \delta_h^{\tau}$$

$$\leq \frac{16H^2}{T} \sum_{h=1}^{H} \sum_{\tau=1}^{\bar{\tau}} \max_{s \in \mathcal{S}} \left( D_R(\mu_h^{\tau,\dagger}, \tilde{\mu}_h^{\tau}(\cdot|s)) + D_R(\nu_h^{\tau,\dagger}, \tilde{\nu}_h^{\tau}(\cdot|s)) \right)$$

$$+ \frac{192H^2}{T} \sum_{h=1}^{H} \sum_{\tau=1}^{\bar{\tau}} \sum_{h'=h+1}^{H} \max_{s \in \mathcal{S}} \left( D_R(\mu_{h'}^{\tau-h'+h,\dagger}, \tilde{\mu}_{h'}^{\tau-h'+h}(\cdot|s)) + D_R(\nu_{h'}^{\tau-h'+h,\dagger}, \tilde{\nu}_{h'}^{\tau-h'+h}(\cdot|s)) \right)$$

$$\leq \frac{192H^2}{T} \sum_{h=1}^{H} \sum_{\tau=1}^{\bar{\tau}} \sum_{h'=h}^{H} \max_{s \in \mathcal{S}} \left( D_R(\mu_{h'}^{\tau-h'+h,\dagger}, \tilde{\mu}_{h'}^{\tau-h'+h}(\cdot|s)) + D_R(\nu_{h'}^{\tau-h'+h,\dagger}, \tilde{\nu}_{h'}^{\tau-h'+h}(\cdot|s)) \right)$$

$$\leq \frac{192H^3}{T} \sum_{h=1}^{H} \sum_{\tau=1}^{\bar{\tau}} \max_s \left( D_R(\mu_h^{\tau,\dagger}, \tilde{\mu}_h^{\tau}(\cdot|s)) + D_R(\nu_h^{\tau,\dagger}, \tilde{\nu}_h^{\tau}(\cdot|s)) \right), \tag{15}$$

where the last step is by switching the order of counting. This proves the first claim in the Theorem.

We now proceed to establish the second statement. Recall that we chose the negative entropy as the regularizer $R$. In this case, the Bregman divergence $D_R(\cdot, \cdot)$ reduces to the Kullback–Leibler divergence. Since $\mu_h^{\tau,\dagger}$ lies in the simplex, when we initialize $\tilde{\mu}_h^{\tau}(\cdot|s) = \mathbf{1}/A$ to be a uniform distribution, we naturally have $D_R(\mu_h^{\tau,\dagger}, \tilde{\mu}_h^{\tau}(\cdot|s)) \leq \log A, \forall s \in \mathcal{S}, h \in [H]$. A similar result holds for $D_R(\nu_h^{\tau,\dagger}, \tilde{\nu}_h^{\tau}(\cdot|s))$. We can hence obtain that

$$\max_s \left( D_R(\mu_h^{\tau,\dagger}, \tilde{\mu}_h^{\tau}(\cdot|s)) + D_R(\nu_h^{\tau,\dagger}, \tilde{\nu}_h^{\tau}(\cdot|s)) \right) \leq \log(AB). \tag{16}$$

To prove the statement, it remains to upper bound the total number of stages $\bar{\tau}$. Recall that we have defined the lengths of the stages to increase exponentially with $L_{\tau+1} = \lfloor (1 + 1/H)L_\tau \rfloor$. Since the $\bar{\tau}$ stages sum up to $T$ iterations in total, by taking the sum of a geometric series, it suffices to find a value of $\bar{\tau}$ such that $(1 + 1/H)^{\bar{\tau}} \geq T/H$. Using the Taylor series expansion, one can show that $(1 + \frac{1}{H})^H \geq e - \frac{e}{2H}$. Hence, it reduces to finding a minimum $\bar{\tau}$ such that

$$\left( e - \frac{e}{2H} \right)^{\bar{\tau}/H} \geq \frac{T}{H}. \tag{17}$$

One can easily see that any $\bar{\tau} \geq \frac{H \log T}{\log(e/2)}$ satisfies the condition. Together with (15) and (16), we obtain that

$$\text{NE-gap}(\bar{\mu}, \bar{\nu}) \leq \frac{768 H^5 \log T}{T} \log(AB).$$

This completes the proof of the theorem. $\qquad \square$

## B.2 Supporting Lemmas for Section 3

Before presenting the supporting lemmas of the section, we remark that we will reload the notations $\mu_h^t$ and $\nu_h^t$ with some slight abuse of notations. Specifically, when $t$ is the last iteration of a stage, $\mu_h^t$ can be used to denote not only the policy at iteration $t$, but also the initial policy of the next stage (see Line 10 of Algorithm 1). In the following proofs, it should be clear from the context which specific policy $\mu_h^t$ refers to. A similar rule applies to $\nu_h^t$.

**Lemma 7.** *Let $(\bar{\mu}, \bar{\nu})$ be the output policies of Algorithm 1. Then,*

$$\text{NE-gap}(\bar{\mu}, \bar{\nu}) \leq \frac{2}{T} \sum_{h=1}^{H} \sum_{\tau=1}^{\bar{\tau}} L_\tau \max_{s \in \mathcal{S}} \left( \text{reg}_{h,1}^\tau(s) + \text{reg}_{h,2}^\tau(s) \right) + \frac{2}{T} \sum_{h=1}^{H} \sum_{\tau=1}^{\bar{\tau}} L_\tau \delta_h^\tau.$$

*Proof.* From Lemma C.1 in [77], we know that

$\text{NE-gap}(\bar{\mu}, \bar{\nu})$

$= V_1^{\dagger, \bar{\nu}}(s_1) - V_1^\star(s_1) + V_1^\star(s_1) - V_1^{\bar{\mu}, \dagger}(s_1)$

$\leq 2 \sum_{h=1}^{H} \max_s \left\{ \max_{\mu_h^\dagger, \nu_h^\dagger} \left[ \langle \mu_h^\dagger, Q_h^\star \bar{\nu}_h \rangle - \langle \nu_h^\dagger, (Q_h^\star)^\top \bar{\mu}_h \rangle \right] (s) \right\}$

$= 2 \sum_{h=1}^{H} \max_s \left\{ \max_{\mu_h^\dagger, \nu_h^\dagger} \frac{1}{T} \sum_{t=1}^{T} \left[ \langle \mu_h^\dagger, Q_h^\star \nu_h^t \rangle - \langle \nu_h^\dagger, (Q_h^\star)^\top \mu_h^t \rangle \right] (s) \right\}$

$\leq 2 \sum_{h=1}^{H} \max_s \left\{ \max_{\mu_h^\dagger, \nu_h^\dagger} \frac{1}{T} \sum_{t=1}^{T} \left[ \langle \mu_h^\dagger, Q_h^{\tau(t)} \nu_h^t \rangle - \langle \nu_h^\dagger, (Q_h^{\tau(t)})^\top \mu_h^t \rangle \right] (s) \right\} + \frac{2}{T} \sum_{h=1}^{H} \sum_{t=1}^{T} \delta_h^t, \qquad (18)$

where the last step is by adding and subtracting the estimated values $Q_h^{\tau(t)}$, and invoking the definition that $\delta_h^t = \left\| Q_h^{\tau(t)} - Q_h^\star \right\|_\infty$. To further bound the first term in (18), notice that

$$\max_s \left\{ \max_{\mu_h^\dagger, \nu_h^\dagger} \frac{1}{T} \sum_{t=1}^{T} \left[ \langle \mu_h^\dagger, Q_h^{\tau(t)} \nu_h^t \rangle - \langle \nu_h^\dagger, (Q_h^{\tau(t)})^\top \mu_h^t \rangle \right] (s) \right\}$$

$$\leq \frac{1}{T} \sum_{\tau=1}^{\bar{\tau}} \max_s \left\{ \max_{\mu_h^{\tau, \dagger}, \nu_h^{\tau, \dagger}} \sum_{j=t_\tau^{\text{start}}}^{t_\tau^{\text{end}}} \left[ \langle \mu_h^{\tau, \dagger}, Q_h^\tau \nu_h^j \rangle - \langle \nu_h^{\tau, \dagger}, (Q_h^\tau)^\top \mu_h^j \rangle \right] (s) \right\}$$

$$\leq \frac{1}{T} \sum_{\tau=1}^{\bar{\tau}} L_\tau \max_s \left( \text{reg}_{h,1}^\tau(s) + \text{reg}_{h,2}^\tau(s) \right). \qquad (19)$$

The first step holds because the LHS uses a fixed pair of best responses $(\mu_h^\dagger, \nu_h^\dagger)$ for the entire $T$ iterations, while the RHS uses a separate best response pair $(\mu_h^{\tau, \dagger}, \nu_h^{\tau, \dagger})$ for each individual stage $\tau$ and then puts them together. The RHS clearly upper bounds the LHS as the RHS maximizes over each stage separately. The last step in (19) holds due to the definitions of $\text{reg}_{h,1}^\tau(s)$ and $\text{reg}_{h,2}^\tau(s)$ that

$$\text{reg}_{h,1}^\tau(s) + \text{reg}_{h,2}^\tau(s) = \max_{\mu_h^{\tau, \dagger}, \nu_h^{\tau, \dagger}} \frac{1}{L_\tau} \sum_{j=t_\tau^{\text{start}}}^{t_\tau^{\text{end}}} \left[ \langle \mu_h^{\tau, \dagger}, Q_h^\tau \nu_h^j \rangle - \langle \nu_h^{\tau, \dagger}, (Q_h^\tau)^\top \mu_h^j \rangle \right] (s).$$

To control the second term in (18), we use the fact that with stage-based value updates, the value estimation error $\delta_h^t$ does not change within a stage. Therefore,

$$\frac{2}{T} \sum_{h=1}^{H} \sum_{t=1}^{T} \delta_h^t = \frac{2}{T} \sum_{h=1}^{H} \sum_{\tau=1}^{\bar{\tau}} \sum_{j=t_\tau^{\text{start}}}^{t_\tau^{\text{end}}} \delta_h^j = \frac{2}{T} \sum_{h=1}^{H} \sum_{\tau=1}^{\bar{\tau}} L_\tau \delta_h^\tau. \qquad (20)$$

Finally, substituting (19) and (20) back to (18) completes the proof. □

**Lemma 8.** *For every stage $\tau \in \mathbb{N}_+$, every step $h \in [H]$ and every state $s \in \mathcal{S}$, the per-state average regret is bounded by:*

$$\text{reg}_{h,1}^{\tau}(s) \leq \frac{1}{\eta L_{\tau}} D_R(\mu_h^{\tau,\dagger}, \tilde{\mu}_h^{\tau}(\cdot|s)) + \frac{2\eta H^2}{L_{\tau}} \sum_{j=t_{\tau}^{start}}^{t_{\tau}^{end}} \left\| \nu_h^j(\cdot \mid s) - \nu_h^{j-1}(\cdot \mid s) \right\|_1^2$$

$$- \frac{1}{8\eta L_{\tau}} \sum_{j=t_{\tau}^{start}}^{t_{\tau}^{end}} \left\| \mu_h^j(\cdot \mid s) - \mu_h^{j-1}(\cdot \mid s) \right\|_1^2, \tag{21}$$

$$\text{reg}_{h,2}^{\tau}(s) \leq \frac{1}{\eta L_{\tau}} D_R(\nu_h^{\tau,\dagger}, \tilde{\nu}_h^{\tau}(\cdot|s)) + \frac{2\eta H^2}{L_{\tau}} \sum_{j=t_{\tau}^{start}}^{t_{\tau}^{end}} \left\| \mu_h^j(\cdot \mid s) - \mu_h^{j-1}(\cdot \mid s) \right\|_1^2$$

$$- \frac{1}{8\eta L_{\tau}} \sum_{j=t_{\tau}^{start}}^{t_{\tau}^{end}} \left\| \nu_h^j(\cdot \mid s) - \nu_h^{j-1}(\cdot \mid s) \right\|_1^2. \tag{22}$$

*In particular, for $\eta \leq 1/(8H^2)$, we further have*

$$\text{reg}_{h,1}^{\tau}(s) + \text{reg}_{h,2}^{\tau}(s) \leq \frac{1}{\eta L_{\tau}} \left( D_R(\mu_h^{\tau,\dagger}, \tilde{\mu}_h^{\tau}(\cdot|s)) + D_R(\nu_h^{\tau,\dagger}, \tilde{\nu}_h^{\tau}(\cdot|s)) \right)$$

$$- \sum_{j=t_{\tau}^{start}}^{t_{\tau}^{end}} \frac{4\eta H^3}{L_{\tau}} \left( \|\nu_h^j(\cdot|s) - \nu_h^{j-1}(\cdot|s)\|_1^2 + \|\mu_h^j(\cdot|s) - \mu_h^{j-1}(\cdot|s)\|_1^2 \right). \tag{23}$$

*Proof.* We prove the regret bound for the max-player, i.e., $\text{reg}_{h,1}^{\tau}(s)$. The bound for the min-player holds analogously. Notice that the policy update steps in Algorithm 1 are exactly the same as the optimistic online mirror descent algorithm [55, 62], with the loss vector $g^t = [Q_h^{\tau} \nu_h^t](s, \cdot)$ and the recency bias $M^t = [Q_h^{\tau} \nu_h^{t-1}](s, \cdot)$. Since our stage-based value updates assign equal weights to each iteration, we end up with a classic no-(average-)regret learning problem instead of a no-(weighed-)regret learning problem as in [72, 77]. This allows us to directly apply the standard optimistic OMD results (e.g., Lemma 1 in [55] and Proposition 5 in [62]) to obtain

$$\text{reg}_{h,1}^{\tau}(s) = \max_{\mu_h^{\tau,\dagger} \in \Delta(\mathcal{A})} \frac{1}{L_{\tau}} \sum_{j=t_{\tau}^{start}}^{t_{\tau}^{end}} \left\langle \mu_h^{\tau,\dagger} - \mu_h^j, Q_h^{\tau} \nu_h^j \right\rangle (s)$$

$$\leq \frac{1}{\eta L_{\tau}} D_R(\mu_h^{\tau,\dagger}, \tilde{\mu}_h^{\tau}(\cdot|s)) + \frac{\eta}{L_{\tau}} \sum_{j=t_{\tau}^{start}}^{t_{\tau}^{end}} \left\| [Q_h^{\tau} \nu_h^j - Q_h^{\tau} \nu_h^{j-1}](s, \cdot) \right\|_{\infty}^2 \tag{24}$$

$$- \frac{1}{8\eta L_{\tau}} \sum_{j=t_{\tau}^{start}}^{t_{\tau}^{end}} \left\| \mu_h^j(\cdot \mid s) - \mu_h^{j-1}(\cdot \mid s) \right\|_1^2. \tag{25}$$

To further upper bound the term in (24), notice that

$$\left\| \left[ Q_h^{\tau} \nu_h^j - Q_h^{\tau} \nu_h^{j-1} \right] (s, \cdot) \right\|_{\infty}^2 \leq 2H^2 \left\| \nu_h^j(\cdot \mid s) - \nu_h^{j-1}(\cdot \mid s) \right\|_1^2,$$

where we used the Hölder's inequality and the fact that $\|Q_h^{\tau}(s, \cdot)\|_{\infty} \leq H$. Substituting the above result back to (25) yields

$$\text{reg}_{h,1}^{\tau}(s) \leq \frac{1}{\eta L_{\tau}} D_R(\mu_h^{\tau,\dagger}, \tilde{\mu}_h^{\tau}(\cdot|s)) + \frac{\eta}{L_{\tau}} \sum_{j=t_{\tau}^{start}}^{t_{\tau}^{end}} 2H^2 \left\| \nu_h^j(\cdot \mid s) - \nu_h^{j-1}(\cdot \mid s) \right\|_1^2$$

$$- \frac{1}{8\eta L_{\tau}} \sum_{j=t_{\tau}^{start}}^{t_{\tau}^{end}} \left\| \mu_h^j(\cdot \mid s) - \mu_h^{j-1}(\cdot \mid s) \right\|_1^2.$$

This completes the proof of (21). The regret bound in (22) can be shown via symmetry.

Combining (21) and (22) leads to

$$
\begin{aligned}
&\operatorname{reg}_{h,1}^{\tau}(s) + \operatorname{reg}_{h,2}^{\tau}(s) \\
&\leq \frac{1}{\eta L_\tau} \left( D_R(\mu_h^{\tau,\dagger}, \tilde{\mu}_h^\tau(\cdot|s)) + D_R(\nu_h^{\tau,\dagger}, \tilde{\nu}_h^\tau(\cdot|s)) \right) \\
&\quad + \sum_{j=t_\tau^{\text{start}}}^{t_\tau^{\text{end}}} \left( \frac{2H^2\eta}{L_\tau} - \frac{1}{8\eta L_\tau} \right) \left( \|\nu_h^j(\cdot|s) - \nu_h^{j-1}(\cdot|s)\|_1^2 + \|\mu_h^j(\cdot|s) - \mu_h^{j-1}(\cdot|s)\|_1^2 \right).
\end{aligned}
$$

When $\eta \leq 1/(8H^2)$, we further have

$$
\begin{aligned}
\operatorname{reg}_{h,1}^{\tau}(s) + \operatorname{reg}_{h,2}^{\tau}(s) \leq &\frac{1}{\eta L_\tau} \left( D_R(\mu_h^{\tau,\dagger}, \tilde{\mu}_h^\tau(\cdot|s)) + D_R(\nu_h^{\tau,\dagger}, \tilde{\nu}_h^\tau(\cdot|s)) \right) \\
&- \sum_{j=t_\tau^{\text{start}}}^{t_\tau^{\text{end}}} \frac{4\eta H^3}{L_\tau} \left( \|\nu_h^j(\cdot|s) - \nu_h^{j-1}(\cdot|s)\|_1^2 + \|\mu_h^j(\cdot|s) - \mu_h^{j-1}(\cdot|s)\|_1^2 \right).
\end{aligned}
$$

This completes the proof of the lemma. □

**Lemma 9.** *With $\eta \leq 1/(8H^2)$, for any iteration $t \in [T]$ and any step $h \in [H]$, we have that*

$$
\delta_h^t \leq \frac{12}{\eta L_{\tau(t)}} \sum_{h'=h+1}^H \max_s \left( D_R(\mu_{h'}^{\tau(t)-h'+h,\dagger}, \tilde{\mu}_{h'}^{\tau(t)-h'+h}(\cdot|s)) + D_R(\nu_{h'}^{\tau(t)-h'+h,\dagger}, \tilde{\nu}_{h'}^{\tau(t)-h'+h}(\cdot|s)) \right).
$$

*Proof.* In the following, when we consider a fixed iteration $t \in [T]$, we drop the notational dependence on $t$ and simply use $\tau$ (instead of $\tau(t)$) to denote the stage that iteration $t$ belongs to. For any $h \in [H-1]$, we can use Lemma 10 (similar to Lemma C.2 of [77]) to establish the following recursion for the value estimation error:

$$
\delta_h^t \leq \delta_{h+1}^{\tau-1} + \operatorname{reg}_{h+1}^{\tau-1}, \tag{26}
$$

where recall that $\operatorname{reg}_h^\tau = \max_{s \in \mathcal{S}} \{\operatorname{reg}_{h,1}^\tau(s), \operatorname{reg}_{h,2}^\tau(s)\}$. Using Lemma 8, we can upper bound the individual regrets $\operatorname{reg}_{h,1}^\tau(s)$ and $\operatorname{reg}_{h,2}^\tau(s)$ by

$$
\operatorname{reg}_{h,1}^\tau(s) \leq \frac{1}{\eta L_\tau} D_R(\mu_h^{\tau,\dagger}, \tilde{\mu}_h^\tau(\cdot|s)) + \frac{2\eta H^2}{L_\tau} \sum_{j=t_\tau^{\text{start}}}^{t_\tau^{\text{end}}} \left\| \nu_h^j(\cdot \mid s) - \nu_h^{j-1}(\cdot \mid s) \right\|_1^2, \tag{27}
$$

$$
\operatorname{reg}_{h,2}^\tau(s) \leq \frac{1}{\eta L_\tau} D_R(\nu_h^{\tau,\dagger}, \tilde{\nu}_h^\tau(\cdot|s)) + \frac{2\eta H^2}{L_\tau} \sum_{j=t_\tau^{\text{start}}}^{t_\tau^{\text{end}}} \left\| \mu_h^j(\cdot \mid s) - \mu_h^{j-1}(\cdot \mid s) \right\|_1^2. \tag{28}
$$

where we have dropped the negative terms in (21) and (22). Following a similar approximate nonnegativity argument as in Lemma 5 of [72] (reproduced in Lemma 11 for our stage-based approach), we obtain that

$$
\operatorname{reg}_{h,1}^\tau(s) + \operatorname{reg}_{h,2}^\tau(s) \geq -2\delta_h^\tau.
$$

Together with (23) in Lemma 8, we obtain that

$$
\begin{aligned}
&\frac{2\eta H^2}{L_\tau} \sum_{j=t_\tau^{\text{start}}}^{t_\tau^{\text{end}}} \left( \|\nu_h^j(\cdot|s) - \nu_h^{j-1}(\cdot|s)\|_1^2 + \|\mu_h^j(\cdot|s) - \mu_h^{j-1}(\cdot|s)\|_1^2 \right) \\
&\leq \frac{\delta_h^\tau}{H} + \frac{1}{2H\eta L_\tau} \left( D_R(\mu_h^{\tau,\dagger}, \tilde{\mu}_h^\tau(\cdot|s)) + D_R(\nu_h^{\tau,\dagger}, \tilde{\nu}_h^\tau(\cdot|s)) \right)
\end{aligned}
$$

Since the above inequality holds for any state $s \in \mathcal{S}$, substituting it back to (27) and (28) yields

$$
\operatorname{reg}_h^\tau \leq \max_s \frac{3}{2\eta L_\tau} \left( D_R(\mu_h^{\tau,\dagger}, \tilde{\mu}_h^\tau(\cdot|s)) + D_R(\nu_h^{\tau,\dagger}, \tilde{\nu}_h^\tau(\cdot|s)) \right) + \frac{\delta_h^\tau}{H}. \tag{29}
$$

We can further substitute the regret bound above back to the recursion 26 to get that

$$\delta_h^\tau \le \frac{3}{2\eta L_{\tau-1}} \max_s \left( D_R(\mu_{h+1}^{\tau-1,\dagger}, \tilde{\mu}_{h+1}^{\tau-1}(\cdot|s)) + D_R(\nu_{h+1}^{\tau-1,\dagger}, \tilde{\nu}_{h+1}^{\tau-1}(\cdot|s)) \right) + (1 + \frac{1}{H})\delta_{h+1}^{\tau-1}, \quad (30)$$

where we used the fact that the value estimation error $\delta_h^t$ does not change within a stage $\tau$ since we perform stage-based value updates. Using a backward inductive argument (starting from the induction basis that $\delta_H^\tau = 0, \forall \tau$), the above recursion in (30) leads us to the following result:

$$\delta_h^\tau \le \sum_{h'=h+1}^{H} \frac{3}{2\eta L_{\tau-h'+h}} \left(1 + \frac{1}{H}\right)^{h'-h-1} \max_s \left( D_R(\mu_{h'}^{\tau-h'+h,\dagger}, \tilde{\mu}_{h'}^{\tau-h'+h}(\cdot|s)) + D_R(\nu_{h'}^{\tau-h'+h,\dagger}, \tilde{\nu}_{h'}^{\tau-h'+h}(\cdot|s)) \right)$$

$$\le \frac{3}{2\eta L_\tau} \sum_{h'=h+1}^{H} \left(1 + \frac{1}{H}\right)^{2(h'-h)-1} \max_s \left( D_R(\mu_{h'}^{\tau-h'+h,\dagger}, \tilde{\mu}_{h'}^{\tau-h'+h}(\cdot|s)) + D_R(\nu_{h'}^{\tau-h'+h,\dagger}, \tilde{\nu}_{h'}^{\tau-h'+h}(\cdot|s)) \right)$$

$$\le \frac{3}{2\eta L_\tau} \sum_{h'=h+1}^{H} \left(1 + \frac{1}{H}\right)^{2H} \max_s \left( D_R(\mu_{h'}^{\tau-h'+h,\dagger}, \tilde{\mu}_{h'}^{\tau-h'+h}(\cdot|s)) + D_R(\nu_{h'}^{\tau-h'+h,\dagger}, \tilde{\nu}_{h'}^{\tau-h'+h}(\cdot|s)) \right)$$

$$\le \frac{12}{\eta L_\tau} \sum_{h'=h+1}^{H} \max_s \left( D_R(\mu_{h'}^{\tau-h'+h,\dagger}, \tilde{\mu}_{h'}^{\tau-h'+h}(\cdot|s)) + D_R(\nu_{h'}^{\tau-h'+h,\dagger}, \tilde{\nu}_{h'}^{\tau-h'+h}(\cdot|s)) \right), \quad (31)$$

where the second step uses our choice of the stage lengths that $L_{\tau+1} = \lfloor (1 + 1/H)L_\tau \rfloor$, which further implies that

$$\frac{1}{L_{\tau-h'+h}} \le \frac{1}{L_\tau} \left(1 + \frac{1}{H}\right)^{h'-h}.$$

The last step in (31) is due to the fact that $(1 + 1/H)^H \le e \approx 2.71828$. This completes the proof of the lemma. $\square$

**Lemma 10.** *(Value estimation error recursion) For any iteration $t \in [T]$ and any step $h \in [H]$, we have the following recursion for the value estimation error $\delta_h^t$:*

$$\delta_h^t \le \delta_{h+1}^{\tau(t)-1} + \mathrm{reg}_{h+1}^{\tau(t)-1}.$$

*Proof.* The proof essentially follows a similar procedure as that of Lemma C.2 in [77]. Let $\tau = \tau(t)$. For any $(h, s, a, b) \in [H] \times \mathcal{S} \times \mathcal{A} \times \mathcal{B}$, we know from the definition of $Q_h^\star$ that

$$Q_h^\star(s, a, b) = r_h(s, a, b) + \max_{\mu_{h+1}\in\Delta(\mathcal{A})} \min_{\nu_{h+1}\in\Delta(\mathcal{B})} P_h \left[ \mu_{h+1}^\top Q_{h+1}^\star \nu_{h+1} \right](s, a, b)$$

$$\le r_h(s, a, b) + \max_{\mu_{h+1}} P_h \left[ \mu_{h+1}^\top Q_{h+1}^\star \left( \frac{1}{L_{\tau-1}} \sum_{j=t_{\tau-1}^{\mathrm{start}}}^{t_{\tau-1}^{\mathrm{end}}} \nu_{h+1}^j \right) \right](s, a, b)$$

$$\le r_h(s, a, b) + \max_{\mu_{h+1}\in} \frac{1}{L_{\tau-1}} \sum_{j=t_{\tau-1}^{\mathrm{start}}}^{t_{\tau-1}^{\mathrm{end}}} P_h \left[ \mu_{h+1}^\top Q_{h+1}^\star \nu_{h+1}^j \right](s, a, b)$$

$$\le r_h(s, a, b) + \max_{\mu_{h+1}\in} \frac{1}{L_{\tau-1}} \sum_{j=t_{\tau-1}^{\mathrm{start}}}^{t_{\tau-1}^{\mathrm{end}}} \left( P_h \left[ \mu_{h+1}^\top Q_{h+1}^{\tau-1} \nu_{h+1}^j \right](s, a, b) + \|Q_{h+1}^\star - Q_{h+1}^{\tau-1}\|_\infty \right),$$

where the second step holds because $\frac{1}{L_{\tau-1}} \sum_{j=t_{\tau-1}^{\mathrm{start}}}^{t_{\tau-1}^{\mathrm{end}}} \nu_{h+1}^j(\cdot|s) \in \Delta(\mathcal{B})$. Using the definitions of $\mathrm{reg}_{h+1}^{\tau-1}$ and $\delta_{h+1}^{\tau-1}$, the above inequality further leads to

$$Q_h^\star(s, a, b) \le r_h(s, a, b) + \frac{1}{L_{\tau-1}} \sum_{j=t_{\tau-1}^{\mathrm{start}}}^{t_{\tau-1}^{\mathrm{end}}} P_h \left[ (\mu_{h+1}^j)^\top Q_{h+1}^{\tau-1} \nu_{h+1}^j \right](s, a, b) + \delta_{h+1}^{\tau-1} + \mathrm{reg}_{h+1}^{\tau-1}$$

$$\le Q_h^\tau(s, a, b) + \delta_{h+1}^{\tau-1} + \mathrm{reg}_{h+1}^{\tau-1}$$

where the last step is due to the value update rule in Algorithm 1. This implies that

$$Q_h^\star(s,a,b) - Q_h^\tau(s,a,b) \le \delta_{h+1}^{\tau-1} + \mathrm{reg}_{h+1}^{\tau-1}.$$

Using a similar argument, we can show a symmetric result for the min-player:

$$Q_h^\tau(s,a,b) - Q_h^\star(s,a,b) \le \delta_{h+1}^{\tau-1} + \mathrm{reg}_{h+1}^{\tau-1}.$$

Combining both directions yields the desired result. $\qquad\square$

**Lemma 11.** *(Approximate non-negativity) For any $\tau \in [\bar\tau]$ and $h \in [H]$, we have that*

$$\mathrm{reg}_{h,1}^\tau(s) + \mathrm{reg}_{h,2}^\tau(s) \ge -2\delta_h^\tau.$$

*Proof.* This lemma can be considered as a stage-based variant of Lemma 5 in [72]. From the definitions of $\mathrm{reg}_{h,1}^\tau(s)$ and $\mathrm{reg}_{h,2}^\tau(s)$, we have that

$$\mathrm{reg}_{h,1}^\tau(s) + \mathrm{reg}_{h,2}^\tau(s)$$

$$= \max_{\mu_h^{\tau,\dagger},\nu_h^{\tau,\dagger}} \frac{1}{L_\tau} \sum_{j=t_\tau^{\mathrm{start}}}^{t_\tau^{\mathrm{end}}} \left( \left\langle \mu_h^{\tau,\dagger}, Q_h^\tau \nu_h^j \right\rangle - \left\langle \nu_h^{\tau,\dagger}, (Q_h^\tau)^\top \mu_h^j \right\rangle \right)(s)$$

$$= \max_{\mu_h^{\tau,\dagger},\nu_h^{\tau,\dagger}} \frac{1}{L_\tau} \left[ \sum_{j=t_\tau^{\mathrm{start}}}^{t_\tau^{\mathrm{end}}} \left( \left\langle \mu_h^{\tau,\dagger}, Q_h^\star \nu_h^j \right\rangle - \left\langle \nu_h^{\tau,\dagger}, (Q_h^\star)^\top \mu_h^j \right\rangle \right)(s) \right.$$

$$\left. + \sum_{j=t_\tau^{\mathrm{start}}}^{t_\tau^{\mathrm{end}}} \left( \left\langle \mu_h^{\tau,\dagger}, (Q_h^\tau - Q_h^\star)\nu_h^j \right\rangle - \left\langle \nu_h^{\tau,\dagger}, (Q_h^\tau - Q_h^\star)^\top \mu_h^j \right\rangle \right)(s) \right]$$

$$\ge \max_{\mu_h^{\tau,\dagger},\nu_h^{\tau,\dagger}} \frac{1}{L_\tau} \left[ \sum_{j=t_\tau^{\mathrm{start}}}^{t_\tau^{\mathrm{end}}} \left( \left\langle \mu_h^{\tau,\dagger}, Q_h^\star \nu_h^j \right\rangle - \left\langle \nu_h^{\tau,\dagger}, (Q_h^\star)^\top \mu_h^j \right\rangle \right)(s) \right] - 2\delta_h^\tau, \qquad (32)$$

where the second step is by adding and subtracting the same term, and the last step uses the definition that $\delta_h^\tau = \|Q_h^\tau - Q_h^\star\|_\infty$. Since both $\frac{1}{L_\tau}\sum_{j=t_\tau^{\mathrm{start}}}^{t_\tau^{\mathrm{end}}} \mu_h^j(\cdot|s)$ and $\frac{1}{L_\tau}\sum_{j=t_\tau^{\mathrm{start}}}^{t_\tau^{\mathrm{end}}} \nu_h^j(\cdot|s)$ are valid probability distributions over the action spaces, the first term in (32) is always non-negative:

$$\max_{\mu_h^{\tau,\dagger},\nu_h^{\tau,\dagger}} \frac{1}{L_\tau} \left[ \sum_{j=t_\tau^{\mathrm{start}}}^{t_\tau^{\mathrm{end}}} \left( \left\langle \mu_h^{\tau,\dagger}, Q_h^\star \nu_h^j \right\rangle - \left\langle \nu_h^{\tau,\dagger}, (Q_h^\star)^\top \mu_h^j \right\rangle \right)(s) \right]$$

$$= \max_{\mu_h^{\tau,\dagger},\nu_h^{\tau,\dagger}} \left[ \left\langle \mu_h^{\tau,\dagger}, Q_h^\star \left( \frac{1}{L_\tau} \sum_{j=t_\tau^{\mathrm{start}}}^{t_\tau^{\mathrm{end}}} \nu_h^j \right) \right\rangle (s) - \left\langle \nu_h^{\tau,\dagger}, (Q_h^\star)^\top \left( \frac{1}{L_\tau} \sum_{j=t_\tau^{\mathrm{start}}}^{t_\tau^{\mathrm{end}}} \mu_h^j \right) \right\rangle (s) \right]$$

$$\ge \left\langle \left( \frac{1}{L_\tau} \sum_{j=t_\tau^{\mathrm{start}}}^{t_\tau^{\mathrm{end}}} \mu_h^j \right), Q_h^\star \left( \frac{1}{L_\tau} \sum_{j=t_\tau^{\mathrm{start}}}^{t_\tau^{\mathrm{end}}} \nu_h^j \right) \right\rangle (s) - \left\langle \left( \frac{1}{L_\tau} \sum_{j=t_\tau^{\mathrm{start}}}^{t_\tau^{\mathrm{end}}} \nu_h^j \right), (Q_h^\star)^\top \left( \frac{1}{L_\tau} \sum_{j=t_\tau^{\mathrm{start}}}^{t_\tau^{\mathrm{end}}} \mu_h^j \right) \right\rangle (s)$$

$$= 0.$$

Plugging the above inequality back into (32) completes the proof. $\qquad\square$

### B.3 Proof of Theorem 2

*Proof.* First, recall the definitions of $(\tilde\mu^k, \tilde\nu^k)$, $(\bar\mu^k, \bar\nu^k)$ and $(\mu^{k,\dagger}, \nu^{k,\dagger})$. Since we use a negative entropy regularizer $R$, the Bregman divergence $D_R(\cdot,\cdot)$ reduces to the Kullback–Leibler divergence. Using these notations, our convergence results of learning in an individual zero-sum game $\mathbb{G}^k$ (Theorem 1) can be written more succinctly as

$$\mathrm{NE\text{-}gap}(\bar\mu^k, \bar\nu^k) \le \frac{192H^3}{T} \left( \mathrm{KL}\left(\mu^{k,\dagger}\|\tilde\mu^k\right) + \mathrm{KL}\left(\nu^{k,\dagger}\|\tilde\nu^k\right) \right),$$

where for ease of notations, we write

$$\text{KL}\left(\mu^{k,\dagger}\|\tilde{\mu}^k\right) := \sum_{h=1}^{H}\sum_{\tau=1}^{\bar{\tau}}\max_s \text{KL}\left(\mu_h^{k,\tau,\dagger}(\cdot|s)\|\tilde{\mu}_h^k(\cdot|s)\right).$$

Here, $\mu_h^{k,\tau,\dagger}(\cdot|s)$ represents the value of $\mu_h^{\tau,\dagger}(\cdot|s)$ in game $\mathbb{G}^k$. The notation $D_R(\nu^{k,\dagger}, \tilde{\nu}^k)$ can be decomposed in a similar manner. By running Algorithm 1 on a sequence of $K$ games, we have that

$$\frac{1}{K}\sum_{k=1}^{K}\text{NE-gap}(\bar{\mu}^k, \bar{\nu}^k) \leq \frac{192H^3}{KT}\sum_{k=1}^{K}\left(\text{KL}\left(\mu^{k,\dagger}\|\tilde{\mu}^k\right) + \text{KL}\left(\nu^{k,\dagger}\|\tilde{\nu}^k\right)\right). \tag{33}$$

In the following, we will focus on the term for the maximizing player in (33). The results for the minimizing player's term can be obtained via symmetry.

Recall the notation that $[\mathbf{x}]_\alpha = (1-\alpha)\mathbf{x} + \frac{\alpha}{d}\mathbf{1}$ for $\mathbf{x} \in \mathbb{R}^d$. By applying this notation entry-wise to each probability distribution in $\mu^{k,\dagger}$ and invoking Lemma 1, we obtain that

$$\frac{1}{K}\sum_{k=1}^{K}\text{KL}\left(\mu^{k,\dagger}\|\tilde{\mu}^k\right) \leq \frac{1}{K}\sum_{k=1}^{K}\text{KL}\left([\mu^{k,\dagger}]_\alpha\|\tilde{\mu}^k\right) + 4H\bar{\tau}\alpha\ln\frac{A}{\alpha}. \tag{34}$$

Notice that the conditions of Lemma 1 are satisfied here because we select our initial policies to be $\tilde{\mu}^k = \frac{1}{k-1}\sum_{k'=1}^{k-1}[\mu^{k',\dagger}]_\alpha$, which assigns a probability of at least $\alpha\mathbf{1}/A$ to each action. Adding and subtracting the same term leads to

$$\sum_{k=1}^{K}\text{KL}\left([\mu^{k,\dagger}]_\alpha\|\tilde{\mu}^k\right) = \min_\mu\sum_{k=1}^{K}\text{KL}\left([\mu^{k,\dagger}]_\alpha\|\mu\right) + \min_\mu\sum_{k=1}^{K}\left(\text{KL}\left([\mu^{k,\dagger}]_\alpha\|\tilde{\mu}^k\right) - \text{KL}\left([\mu^{k,\dagger}]_\alpha\|\mu\right)\right)$$

$$\leq \min_\mu\sum_{k=1}^{K}\text{KL}\left([\mu^{k,\dagger}]_\alpha\|\mu\right) + \frac{8A(1+\ln K)}{\alpha}, \tag{35}$$

where the minimum $\mu$ is taken over all policies of the form of $\mu : [\bar{\tau}] \times [H] \times \mathcal{S} \to \Delta(\mathcal{A})$. We now turn to establish the second step in (35), which reduces to bounding the following regret where the loss functions are given by the Bregman divergences:

$$\text{reg} = \min_\mu\sum_{k=1}^{K}\left(\text{KL}\left([\mu^{k,\dagger}]_\alpha\|\tilde{\mu}^k\right) - \text{KL}\left([\mu^{k,\dagger}]_\alpha\|\mu\right)\right).$$

It is known that the unique minimum of $\sum_{k'=1}^{k}\text{KL}([\mu^{k',\dagger}]_\alpha\|\cdot)$ is attained at $\frac{1}{k}\sum_{k'=1}^{k}[\mu^{k',\dagger}]_\alpha$ (see Proposition 1 of [5] for a proof of this claim). Therefore, by letting $\tilde{\mu}^k = \frac{1}{k-1}\sum_{k'=1}^{k-1}[\mu^{k',\dagger}]_\alpha$, we are essentially running the follow-the-leader (FTL) algorithm (separately for each entry $(\tau, h, s) \in [\bar{\tau}] \times [H] \times \mathcal{S}$) on the sequence of losses defined by $\sum_{k=1}^{K}\text{KL}([\mu^{k,\dagger}]_\alpha\|\cdot)$. We can then invoke the logarithmic regret guarantee of FTL with respect to Bregman divergences, which was established in [35] and was reproduced as Lemma 2 in Appendix A for completeness. To show that Lemma 2 is applicable, we remark that the Kullback–Leibler divergence is not Lipschitz continuous near the boundary of the probability simplex, which breaks condition required by Lemma 2. However, by restricting to policies of the form $[\mu]_\alpha = (1-\alpha)\mu + \frac{\alpha}{A}\mathbf{1}$, which is at least $\frac{\alpha}{A}$-distance away from the simplex boundary, the Kullback–Leibler divergence is indeed Lipschitz continuous within this $\frac{\alpha}{A}$-restricted domain. One can show that the Lipschitz constant of each entry of $\text{KL}([\mu^{k,\dagger}]_\alpha\|\cdot)$ is $\frac{2A}{\alpha}$ within the $\frac{\alpha}{A}$-restricted domain. This allows us to apply Lemma 2 to obtain the result in (35).

Moving forward from (35), we again apply the property that the unique minimum of $\sum_{k=1}^{K}\text{KL}([\mu^{k,\dagger}]_\alpha\|\cdot)$ is attained at $\mu = \frac{1}{K}\sum_{k=1}^{K}[\mu^{k,\dagger}]_\alpha$, which leads to

$$\sum_{k=1}^{K}\text{KL}\left([\mu^{k,\dagger}]_\alpha\|\tilde{\mu}^k\right) \leq \min_\mu\sum_{k=1}^{K}\text{KL}\left([\mu^{k,\dagger}]_\alpha\|\mu\right) + \frac{8A(1+\ln K)}{\alpha}$$

$$= \sum_{k=1}^{K}\text{KL}\left([\mu^{k,\dagger}]_\alpha\|[\mu^\star]_\alpha\right) + \frac{8A(1+\ln K)}{\alpha}$$

$$\leq (1-\alpha)\sum_{k=1}^{K}\text{KL}\left(\mu^{k,\dagger}\|\mu^\star\right) + \frac{8A(1+\ln K)}{\alpha}, \tag{36}$$

where the second step uses the definition that $\mu^\star = \frac{1}{K}\sum_{k=1}^{K}\mu^{k,\dagger}$, and the last step is by the (joint) convexity of the Kullback–Leibler divergence. Substituting (36) to (34) yields

$$\frac{1}{K}\sum_{k=1}^{K}\mathrm{KL}\left(\mu^{k,\dagger}\|\tilde{\mu}^k\right) \leq \frac{1}{K}\sum_{k=1}^{K}\mathrm{KL}\left(\mu^{k,\dagger}\|\mu^\star\right) + \frac{8A(1+\ln K)}{K\alpha} + 4H\bar{\tau}\alpha\ln\frac{A}{\alpha}.$$

By a similar argument, we can show an analogous result for the minimizing player:

$$\frac{1}{K}\sum_{k=1}^{K}\mathrm{KL}\left(\nu^{k,\dagger}\|\tilde{\nu}^k\right) \leq \frac{1}{K}\sum_{k=1}^{K}\mathrm{KL}\left(\nu^{k,\dagger}\|\nu^\star\right) + \frac{8B(1+\ln K)}{K\alpha} + 4H\bar{\tau}\alpha\ln\frac{B}{\alpha}$$

Substituting the above results back into (33) and using the definition

$$\Delta_{\mu,\nu} = \sum_{k=1}^{K}\left(\mathrm{KL}\left(\mu^{k,\dagger}\|\mu^\star\right) + \mathrm{KL}\left(\nu^{k,\dagger}\|\nu^\star\right)\right),$$

we obtain that

$$\frac{1}{K}\sum_{k=1}^{K}\mathrm{NE\text{-}gap}(\bar{\mu}^k,\bar{\nu}^k) \leq \frac{192H^3}{KT}\left(\Delta_{\mu,\nu} + \frac{10(A+B)\ln K}{\alpha} + 4KH\bar{\tau}\alpha\ln\frac{AB}{\alpha^2}\right)$$

Further using the conditions that $\alpha = 1/\sqrt{K}$ and $\bar{\tau} \leq 4H\log T$ (see (17) for a proof) yields

$$\frac{1}{K}\sum_{k=1}^{K}\mathrm{NE\text{-}gap}(\bar{\mu}^k,\bar{\nu}^k) \leq \frac{192H^3}{T}\left(\frac{\Delta_{\mu,\nu}}{K} + \frac{10(A+B)\log K}{\sqrt{K}} + \frac{16H^2\log T\log(ABK)}{\sqrt{K}}\right).$$

This completes the proof of the theorem. $\qquad\square$

## C Infinite-Horizon Discounted Markov Potential Game

To be consistent with existing results in the literature, we consider an infinite-horizon $\gamma$-discounted reward setting for MPGs [43, 39, 76, 15]. An $N$-player, infinite-horizon, discounted stochastic (or Markov) game $\mathbb{G}$ is defined by a tuple $(\mathcal{N}, \mathcal{S}, \{\mathcal{A}_i\}_{i=1}^{N}, P, \{r_i\}_{i=1}^{N}, \gamma, \rho)$, where (1) $\mathcal{N} = \{1, 2, \ldots, N\}$ is the set of players (agents); (2) $\mathcal{S}$ is the finite state space; (3) $\mathcal{A}_i$ is the finite action space for agent $i \in \mathcal{N}$; (4) $P : \mathcal{S} \times \mathcal{A} \to \Delta(\mathcal{S})$ is the transition kernel, where $\mathcal{A} = \times_{i=1}^{N}\mathcal{A}_i$ is the joint action space, and $P(\cdot|s, a) \in \Delta(\mathcal{S})$ denotes the distribution over the next state for $a \in \mathcal{A}$; (5) $r_i : \mathcal{S} \times \mathcal{A} \to [-1, 1]$ is the reward function for agent $i$; (6) $\gamma \in [0, 1)$ denotes the discount factor; and (7) $\rho \in \Delta(\mathcal{S})$ is the initial state distribution. Both the reward function and the state transition function depend on the joint actions of all the agents. We use $a_i \in \mathcal{A}_i$ to denote the individual action of agent $i \in \mathcal{N}$. The subscript $-i$ to denotes the set of agents excluding agent $i$, i.e., $\mathcal{N}\backslash\{i\}$. We can rewrite $a = (a_i, a_{-i})$ using this convention. Let $S = |\mathcal{S}|$, $A_i = |\mathcal{A}_i|, \forall i \in \mathcal{N}$, and $A_{\max} = \max_{i \in \mathcal{N}} A_i$.

A (Markov) policy $\pi_i : \mathcal{S} \to \Delta(\mathcal{A}_i)$ for agent $i \in \mathcal{N}$ is a mapping from the state space to a distribution over the action space. We let agent $i$'s policy be parameterized by $\theta_i = \{\theta_i(a_i|s) \in \mathbb{R}\}_{s \in \mathcal{S}, a_i \in \mathcal{A}_i}$, and denote the policy by $\pi_{\theta_i}$ to emphasize such parameterization. Important examples include direct policy parameterization $\pi_{\theta_i}(a_i|s) = \theta_i(a_i|s)$ and softmax parameterization $\pi_{\theta_i}(a_i|s) = \exp(\theta_i(a_i|s))/\sum_{a_i' \in \mathcal{A}_i}\exp(\theta_i(a_i'|s)), \forall s \in \mathcal{S}, a_i \in \mathcal{A}_i$. Let $\Theta_i$ denote the parameterization-dependent[1] space where $\theta_i$ takes values from, and let $\Theta = \times_{i=1}^{N}\Theta_i$. A joint (product) policy $\pi_\theta = (\pi_{\theta_1}, \ldots, \pi_{\theta_N})$ induces a probability measure over the sequence of states and joint actions. When the policy parameterization scheme is fixed, we sometimes denote a policy $\pi_\theta$ (resp. $\pi_{\theta_i}$) simply by its parameter $\theta$ (resp. $\theta_i$). For a joint policy $\theta = (\theta_1, \ldots, \theta_N)$, and for any $s \in \mathcal{S}$ and $a \in \mathcal{A}$, we define the value function and the state-action value function (or Q-function) for agent $i$ as follows:

$$V_i^s(\theta; \mathbb{G}) := \mathbb{E}_{\theta,\mathbb{G}}\left[\sum_{t=0}^{\infty}\gamma^t r_i(s^t, a^t) \mid s^0 = s\right], \tag{37}$$

$$Q_i^{s,a}(\theta; \mathbb{G}) := \mathbb{E}_{\theta,\mathbb{G}}\left[\sum_{t=0}^{\infty}\gamma^t r_i(s^t, a^t) \mid s^0 = s, a^0 = a\right].$$

---

[1]For example, direct parameterization requires that $\theta_{s,a_i} \geq 0$ and $\sum_{a_i \in \mathcal{A}_i}\theta_{s,a_i} = 1, \forall s \in \mathcal{S}, a_i \in \mathcal{A}_i$, while softmax parameterization allows for $\Theta_i = \mathbb{R}^{|\mathcal{S}||\mathcal{A}_i|}$.

For each agent $i$, by averaging over the other agents' policies, we define the averaged Q-function $\bar{Q}_i^{s,a_i}$ of a joint policy $\theta = (\theta_i, \theta_{-i})$ for any $s \in \mathcal{S}, a_i \in \mathcal{A}_i$ as:

$$\bar{Q}_i^{s,a_i}(\theta; \mathbb{G}) := \sum_{a_{-i} \in \mathcal{A}_{-i}} \theta_{-i}(a_{-i}|s) Q_i^{s,(a_i,a_{-i})}(\theta; \mathbb{G}).$$

With a slight abuse of notation, we write $V_i^\rho(\theta; \mathbb{G}) := \mathbb{E}_{s \sim \rho}[V_i^s(\theta; \mathbb{G})]$ for a state distribution $\rho \in \Delta(\mathcal{S})$. We sometimes also suppress the notation of $\mathbb{G}$ when it is clear from context.

Each agent seeks to find a policy that maximizes its own cumulative reward. The notion of Nash equilibrium in such an infinite-horizon discounted reward setting is defined as follows.

**Definition 1.** *(Nash Equilibrium). For any $\varepsilon \geq 0$, a joint (product) policy $\theta^\star = (\theta_i^\star, \theta_{-i}^\star)$ is an $\varepsilon$-approximate (Markov perfect) Nash equilibrium of a game $\mathbb{G}$ if*

$$V_i^s(\theta_i^\star, \theta_{-i}^\star; \mathbb{G}) \geq V_i^s(\theta_i, \theta_{-i}^\star; \mathbb{G}) - \varepsilon, \forall i \in \mathcal{N}, \theta_i \in \Theta_i, s \in \mathcal{S}.$$

In the infinite-horizon setting, a Markov game $\mathbb{G}$ is a Markov potential game (MPG) if there exists a global potential function $\Phi : \Theta \times \mathcal{S} \to \mathbb{R}$, such that for any state $s \in \mathcal{S}$, any $i \in \mathcal{N}$, and any $\theta_i, \theta_i' \in \Theta_i, \theta_{-i} \in \Theta_{-i}$:

$$\Phi_s(\theta_i, \theta_{-i}; \mathbb{G}) - \Phi_s(\theta_i', \theta_{-i}; \mathbb{G}) = V_i^s(\theta_i, \theta_{-i}; \mathbb{G}) - V_i^s(\theta_i', \theta_{-i}; \mathbb{G}). \tag{38}$$

Intuitively, MPGs capture the variations of the agents' individual values by a single global potential function. MPGs cover Markov teams [36] as a special case, a cooperative setting where all agents share the same reward function $r = r_i, \forall i \in \mathcal{N}$. We also write $\Phi(\theta; \mathbb{G}) := \mathbb{E}_{s \sim \rho}[\Phi_s(\theta; \mathbb{G})]$ for the initial state distribution $\rho \in \Delta(\mathcal{S})$. By linearity of expectation, $\Phi(\theta_i, \theta_{-i}; \mathbb{G}) - \Phi(\theta_i', \theta_{-i}; \mathbb{G}) = V_i^\rho(\theta_i, \theta_{-i}; \mathbb{G}) - V_i^\rho(\theta_i', \theta_{-i}; \mathbb{G})$. One can easily show that there exists a constant $\Phi_{\max} \in [0, \frac{2N}{1-\gamma}]$, such that $|\Phi(\theta; \mathbb{G}) - \Phi(\theta'; \mathbb{G})| \leq \Phi_{\max}, \forall \theta, \theta' \in \Theta$. Finally, we define the discounted state visitation distribution of policy $\theta$ on game $\mathbb{G}$ as

$$d_\rho^\theta(s; \mathbb{G}) = (1 - \gamma)\mathbb{E}_{s^0 \sim \rho} \sum_{t=0}^\infty \gamma^t \mathbb{P}_{\theta, \mathbb{G}}(s^t = s|s_0).$$

Subsequently, the distribution mismatch coefficient of game $\mathbb{G}$ is defined as $\kappa(\mathbb{G}) = \sup_{\theta \in \Theta} \|d_\theta^\rho(\cdot; \mathbb{G})/\rho\|_\infty$. For a set $\mathcal{G}$ of games, we let $\kappa = \sup_{\mathbb{G} \in \mathcal{G}} \kappa(\mathbb{G})$.

# D   Supplementary Material for Section 4

## D.1   Proof of Theorem 3

*Proof.* Proposition 1 implies that if the agents run projected Q-descent on the Markov potential game $\mathbb{G}^k$ for $T$ iterations, we have

$$\sum_{t=0}^{T-1} \max_{i \in \mathcal{N}} \left( \max_{\theta_i' \in \Theta_i} V_i^\rho(\theta_i', \theta_{-i}^{k,t}; \mathbb{G}^k) - V_i^\rho(\theta_i^{k,t}, \theta_{-i}^{k,t}; \mathbb{G}^k) \right) \leq \sqrt{\frac{\kappa(\mathbb{G}^k)T(\Phi(\theta^{k,T}; \mathbb{G}^k) - \Phi(\theta^{k,0}, \mathbb{G}^k))}{\alpha(1-\gamma)^2}}. \tag{39}$$

From the Cauchy-Schwarz inequality, we have that

$$\frac{1}{K} \sum_{k=1}^K \sqrt{\Phi(\theta^{k,T}; \mathbb{G}^k) - \Phi(\theta^{k,0}; \mathbb{G}^k)} \leq \sqrt{\frac{1}{K} \sum_{k=1}^K (\Phi(\theta^{k,T}; \mathbb{G}^k) - \Phi(\theta^{k,0}; \mathbb{G}^k))}$$

$$\leq \sqrt{\frac{1}{K} \left( 2\Phi_{\max} + \sum_{k=1}^{K-1} (\Phi(\theta^{k,T}; \mathbb{G}^k) - \Phi(\theta^{k+1,0}; \mathbb{G}^{k+1})) \right)}$$

$$\leq \sqrt{\frac{1}{K} \left( 2\Phi_{\max} + \sum_{k=1}^{K-1} (\Phi(\theta^{k,T}; \mathbb{G}^k) - \Phi(\theta^{k,T}; \mathbb{G}^{k+1})) \right)}$$

$$\leq \sqrt{\frac{1}{K} (2\Phi_{\max} + \Delta_\Phi)}$$

where the third inequality uses the outer stage update rule that $\theta^{k+1,0} = \theta^{k,T}$, and the last inequality follows from the definition of the similarity metric $\Delta_\Phi$. Plugging the above result into (39), we have that

$$\frac{1}{K}\frac{1}{T}\sum_{k=1}^{K}\sum_{t=0}^{T-1}\max_{i\in\mathcal{N}}\left(\max_{\theta_i'\in\Theta_i}V_i^\rho(\theta_i',\theta_{-i}^{k,t};\mathbb{G}^k) - V_i^\rho(\theta_i^{k,t},\theta_{-i}^{k,t};\mathbb{G}^k)\right)$$

$$\leq\sqrt{\frac{\kappa(2\Phi_{\max}+\Delta_\Phi)}{\alpha(1-\gamma)^2 KT}} \leq \sqrt{\frac{8\kappa^4 NA_{\max}(2\Phi_{\max}+\Delta_\Phi)}{(1-\gamma)^6 KT}},$$

where in the second inequality we set the learning rate as $\alpha = \frac{(1-\gamma)^4}{8\kappa^3 NA_{\max}}$. Therefore, for an average game, $T = O\left(\frac{NA_{\max}\kappa^4(\Phi_{\max}+\Delta_\Phi)}{K(1-\gamma)^6\varepsilon^2}\right)$ steps in the inner stage suffice to find an $\varepsilon$-approximate Nash equilibrium. $\qquad\square$

## D.2 Model-Agnostic Meta-Learning in Markov Potential Games

In what follows, we study meta-learning in MPG under the same formulation as MAML [20, 17, 30]. Let $\mathcal{G} = \{\mathbb{G}^j\}$ be a set of different infinite-horizon discounted reward Markov potential games. The games are drawn from a fixed distribution $p$ that we can sample from. Each game is defined by a tuple $\mathbb{G}^j = (\mathcal{N}, \mathcal{S}, \{\mathcal{A}_i\}_{i=1}^N, P^j, \{r_i^j\}_{i=1}^N, \gamma, \rho^j)$, where we assume without loss of generality that the games share the same agent set, state & action spaces and discount factor, but can have different transition and reward functions and initial state distributions. MAML tries to learn a good initialization from which running one or a few steps of gradient descents/ascents with respect to a new task lead to well-performing model parameters. In the case of multi-agent meta-reinforcement learning with one gradient ascent step, the problem can be formulated as

$$\max_{\theta\in\Theta}F_1(\theta) := \mathbb{E}_{\mathbb{G}\sim p(\mathcal{G})}\left[\Phi\left(\theta + \alpha\nabla\Phi(\theta;\mathbb{G});\mathbb{G}\right)\right], \tag{40}$$

where $\alpha > 0$ is the step size of the policy gradient update. Such a formulation can also be extended to multiple steps of policy gradients. Let $\zeta(\cdot\,;\mathbb{G})$ denote the operator of performing one step of policy gradient update on game $\mathbb{G}$, i.e., $\zeta(\theta;\mathbb{G}) := \theta + \alpha\nabla\Phi(\theta;\mathbb{G})$. The $T$-step extension of the objective (40) can be written as

$$\max_{\theta\in\Theta}F_T(\theta) := \mathbb{E}_{\mathbb{G}\sim p(\mathcal{G})}\left[\Phi\left(\zeta(\ldots(\zeta(\theta;\mathbb{G}))\ldots);\mathbb{G}\right)\right], \tag{41}$$

where the operator $\zeta(\cdot\,;\mathbb{G})$ is applied $T$ times.

Optimizing the multi-step MAML objective typically involves two nested stages: The inner stage (or base algorithm) runs multiple steps of gradient ascents for each individual task, while the outer stage (or meta-algorithm) is an iterative process that updates the meta-parameter $\theta$ over different tasks. Specifically, suppose the outer stage runs for $K$ iterations. Let $\theta^k$ denote the value of $\theta$ at the beginning of the $k$-th iteration of the outer stage. In each iteration, we sample games from the set $\mathcal{G}$ according to the distribution $p$. For each individual game $\mathbb{G} \in \mathcal{G}$ encountered during iteration $k$, the inner stage runs $T$ steps of gradient ascent (or its variants) on it:

$$\theta^{k,t+1}(\mathbb{G}) \leftarrow \psi(\theta^{k,t}(\mathbb{G});\mathbb{G}), \text{ for } 0 \leq t \leq T-1, \tag{42}$$

where $\theta^{k,0}(\mathbb{G}) = \theta^k, \forall\mathbb{G} \in \mathcal{G}$. We often suppress the notation of $\mathbb{G}$ in $\theta^{k,t}(\mathbb{G})$ when there is no ambiguity. Finally, the outer stage updates the meta-parameter by

$$\theta^{k+1} \leftarrow \Psi(\theta^k, \mathcal{G}), \tag{43}$$

using a certain update rule $\Psi$. The meta-parameter $\theta^{k+1}$ is then used as the initialization $\theta^{k+1,0}$ for iteration $k+1$. For simplicity of presentation, we present our results in the same setting as in [66] where $\mathcal{G}$ consists of a finite set of $M$ games and $p$ is a uniform distribution. Our results can be easily extended to the settings where there is an infinite number of games and $p$ is a generic probability distribution, as has been done in existing works [17, 18, 30].

In the following, we develop a meta-learning procedure $(\psi, \Psi)$ that finds a stationary point of the meta-objective (41) while at the same time converging to an approximate Nash equilibrium for each individual game encountered, assuming a sufficient number of policy gradient steps are taken in each game. We focus on softmax parameterization where each agent's policy is given by

$\pi_{\theta_i}(a_i|s) = \exp(\theta_i(a_i|s))/\sum_{a'_i \in \mathcal{A}_i} \exp(\theta_i(a'_i|s)), \forall s \in \mathcal{S}, a_i \in \mathcal{A}_i$. In the inner stage, each agent independently runs gradient ascents with respect to its own value functions to update its parameters. Specifically, on each game $\mathbb{G} \in \mathcal{G}$ encountered during the $k$-th outer iteration, agent $i$ updates its policy parameter $\theta_i$ by

$$\theta_i^{k,t+1}(\mathbb{G}) \leftarrow \theta_i^{k,t}(\mathbb{G}) + \alpha \nabla_{\theta_i} V_i^\rho(\theta^{k,t}(\mathbb{G}); \mathbb{G}), \forall 0 \leq t \leq T-1. \tag{44}$$

We sometimes omit the dependence of $\theta_i^{k,t}(\mathbb{G})$ on $\mathbb{G}$ when the game is clear from the context. Using (the multi-agent extension of) the policy gradient theorem [60, 76], the gradient $\nabla_{\theta_i} V_i^\rho(\theta; \mathbb{G})$ can be calculated as

$$\frac{\partial V_i^\rho(\theta; \mathbb{G})}{\partial \theta_i(a_i|s)} = \frac{1}{1-\gamma} d_\rho^\theta(s; \mathbb{G}) \pi_{\theta_i}(a_i|s) \bar{A}_i^{s,a_i}(\theta; \mathbb{G}), \tag{45}$$

where $d_\rho^\theta(s; \mathbb{G}) = (1-\gamma)\mathbb{E}_{s^0 \sim \rho} \sum_{t=0}^\infty \gamma^t \mathbb{P}_{\theta,\mathbb{G}}(s^t = s|s_0)$ is the discounted state visitation distribution, and $\bar{A}_i^{s,a_i}(\theta; \mathbb{G})$ is the averaged advantage function. Unbiased estimators of the policy gradient can be constructed by using the sampler from [1]. For simplicity, we assume that the exact policy gradients are given. It follows from the definition of the potential function (38) that $\nabla_{\theta_i} V_i^\rho(\theta; \mathbb{G}) = \nabla_{\theta_i} \Phi(\theta; \mathbb{G})$, which indicates that independent policy gradient updates with individual value functions (44) is equivalent to running centralized gradient ascents with respect to the potential function (42). Hence, the base algorithm for each individual game can be executed in a decentralized way. Finally, we invoke Theorem 5 of [78] to show that under mild assumptions, our policy gradient updates with softmax parameterization (44) find an approximate Nash equilibrium of each individual game. Specifically, for any $\varepsilon > 0$, if we run the inner stage for sufficient number of steps $T = O(1/\varepsilon^2)$, our method will find an $\varepsilon$-approximate NE for each individual game.

Our outer stage follows the MAML algorithm by running gradient ascent with respect to the meta-objective $F_T$ from (40). The gradient of $F_T$ can be written as

$$\nabla F_T(\theta) = \mathbb{E}_{\mathbb{G} \sim p(\mathcal{G})} \left[ \left( \prod_{t=0}^{T-1} \left( I + \alpha \nabla^2 \Phi(\theta^{(t)}(\mathbb{G}); \mathbb{G}) \right) \right) \nabla \Phi(\theta^{(T)}(\mathbb{G}); \mathbb{G}) \right], \tag{46}$$

where $\theta^{(0)}(\mathbb{G}) = \theta$ and $\theta^{(t+1)}(\mathbb{G}) = \Psi(\theta^{(t)}(\mathbb{G}); \mathbb{G})$. Accordingly, we instantiate the outer stage update (43) as

$$\theta^{k+1} \leftarrow \theta^k + \frac{\eta}{|\mathcal{G}|} \sum_{\mathbb{G} \in \mathcal{G}} \left( \prod_{t=0}^{T-1} \left( I + \alpha \nabla^2 \Phi(\theta^{k,t}(\mathbb{G}); \mathbb{G}) \right) \right) \nabla \Phi(\theta^{k,T}(\mathbb{G}); \mathbb{G}), \tag{47}$$

where $\eta > 0$ is the learning rate of the outer stage. We assume for simplicity that the exact values of the policy gradient $\nabla \Phi(\theta^{k,T}(\mathbb{G}); \mathbb{G})$ and the policy Hessian $\nabla^2 \Phi(\theta^{k,t}(\mathbb{G}); \mathbb{G})$ are given. In practice, one can construct unbiased estimators of the policy gradient from samples, as the policy gradient and policy Hessian can be written explicitly in a closed form that is compatible with samplers (Lemma 15). We remark that the policy Hessian depends on the cross terms of the agents' policy parameters, which can only be calculated in a centralized way. Our inner stage, though, can still be executed in a decentralized manner. Our algorithm hence falls into in the regime of centralized (meta-)training with decentralized (meta-)execution [42], a popular strategy used for training MARL algorithms.

In order to establish the convergence of (47) to the stationary point of the meta-objective (40), we first show the smoothness of the meta-objective through the following sequence of lemmas.

**Lemma 12.** *Under softmax parameterization, for any policy parameter $\theta \in \Theta$, any state $s \in \mathcal{S}$ and any joint action $a \in \mathcal{A}$, we have (i) $\|\nabla_\theta \log \pi_\theta(a|s)\| \leq \sqrt{2N}$, and (ii) $\|\nabla_\theta^2 \log \pi(a|s)\| \leq 2$. Furthermore, for any policy parameters $\theta, \theta' \in \Theta$, we have (iii) $\|\nabla_\theta^2 \log \pi_\theta(a|s) - \nabla_\theta^2 \log \pi_{\theta'}(a|s)\| \leq 12 \|\theta - \theta'\|$.*

**Lemma 13.** *Under softmax parameterization, for any Markov potential game $\mathbb{G} \in \mathcal{G}$, any policy parameters $\theta, \theta' \in \Theta$, any state $s \in \mathcal{S}$ and any joint action $a \in \mathcal{A}$, the potential function $\Phi$ satisfies the following properties:*

*(i) Bounded policy gradient: $\|\nabla \Phi(\theta; \mathbb{G})\| \leq B_G := \frac{\sqrt{2N}}{(1-\gamma)^2}$;*

*(ii) Bounded policy Hessian: $\|\nabla^2 \Phi(\theta; \mathbb{G})\| \leq L_G := \frac{6N}{(1-\gamma)^3}$;*

*(iii) Lipschitz policy Hessian: $\|\nabla^2 \Phi(\theta; \mathbb{G}) - \nabla^2 \Phi(\theta'; \mathbb{G})\| \leq L_H \|\theta - \theta'\|$, where $L_H := \frac{56N^{3/2}}{(1-\gamma)^4}$.*

**Lemma 14.** *(Meta-objective smoothness). Consider running* (44) *with softmax parameterization and* $\alpha = \frac{(1-\gamma)^3}{2N\gamma A_{\max}}$ *as the inner stage and running* (47) *as the outer stage. Then, the meta-objective* (41) *is $L_F$-smooth for $L_F = (\alpha T B_G L_H + L_G)2^{2T}$.*

The smoothness constant $L_F$ has an exponential dependence on the number of inner stage update steps $T$, which seems unavoidable even in supervised meta-learning. Based on the smoothness property, we can show that our method finds a stationary point of the meta-objective (Theorem 4).

### D.3  Proof of Lemma 12

*Proof.* For agent $i \in \mathcal{N}$, for any state $s \in \mathcal{S}$ and action $a_i \in \mathcal{A}_i$, the softmax policy with parameter $\theta_i$ can be written as

$$\pi_{\theta_i}(a_i|s) = \frac{\exp(\mathbf{1}_{s,a_i}^\top \theta_i)}{\sum_{a_i' \in \mathcal{A}_i} \exp(\mathbf{1}_{s,a_i'}^\top \theta_i)},$$

where $\theta_i \in \mathbb{R}^{|\mathcal{S}||\mathcal{A}_i|}$, and $\mathbf{1}_{s,a_i}$ is an $|\mathcal{S}||\mathcal{A}_i|$-dimensional one-hot vector that has a 1 at index $(s, a_i)$ and 0s at all the other indices. It is known that (see, e.g., [1])

$$\frac{\partial \log \pi_{\theta_i}(a_i|s)}{\partial \theta_i(a_i'|s')} = \mathbb{1}[s = s'](\mathbb{1}[a = a'] - \pi_{\theta_i}(a'|s)),$$

where $\mathbb{1}[\cdot]$ is the indicator function. Hence, we have

$$\|\nabla_{\theta_i} \log \pi_{\theta_i}(a_i|s)\| \leq \sqrt{2}. \tag{48}$$

Since we consider product policies, for any joint action $a = (a_1, \ldots, a_N)$, we have $\pi_\theta(a|s) = \prod_{i=1}^N \pi_{\theta_i}(a_i|s)$. Therefore, it holds that

$$\|\nabla_\theta \log \pi_\theta(a|s)\|^2 \leq \sum_{i=1}^N \|\nabla_{\theta_i} \log \pi_{\theta_i}(a_i|s)\|^2 \leq 2N.$$

We can hence conclude that $\|\nabla_\theta \log \pi_\theta(a|s)\| \leq \sqrt{2N}$. This completes the proof of result (i). Next, to show result (ii), we first write the Hessian $\nabla_{\theta_i}^2 \log \pi_{\theta_i}(a_i|s)$ as (see, e.g., [18] for a proof)

$$\nabla_{\theta_i}^2 \log \pi_{\theta_i}(a_i|s) = -\mathbb{E}_{a_i' \sim \pi_{\theta_i}(a_i'|s)} \left[ \left(\mathbf{1}_{s,a_i'} - \mathbb{E}_{a_i'' \sim \pi_{\theta_i}(a_i''|s)}[\mathbf{1}_{s,a_i''}]\right)\left(\mathbf{1}_{s,a_i'} - \mathbb{E}_{a_i'' \sim \pi_{\theta_i}(a_i''|s)}[\mathbf{1}_{s,a_i''}]\right)^\top \right].$$

To find the upper bound and Lipschitz constant of $\nabla_{\theta_i}^2 \log \pi_{\theta_i}(a_i|s)$, we will rely on two technical lemmas from [18], reproduced as Lemmas 3 and 4 in Appendix A. Since $\|\nabla_{\theta_i} \log \pi_{\theta_i}(a_i|s)\| \leq 2$, from Lemma 4, we know that $\mathbb{E}_{a_i'' \sim \pi_{\theta_i}(a_i''|s)}[\mathbf{1}_{s,a_i''}]$ is Lipschitz continuous with constant 2. By the definition of $\mathbf{1}_{s,a_i}$, we have $\left\|\mathbb{E}_{a_i'' \sim \pi_{\theta_i}(a_i''|s)}[\mathbf{1}_{s,a_i''}]\right\| \leq 1$. Since for any matrix $A$, a sub-multiplicative matrix norm $\|\cdot\|$ satisfies $\|A\|_2^2 \leq \|A\|_1 \|A\|_\infty$, we can conclude that

$$\left\| \left(\mathbf{1}_{s,a_i'} - \mathbb{E}_{a_i'' \sim \pi_{\theta_i}(a_i''|s)}[\mathbf{1}_{s,a_i''}]\right)\left(\mathbf{1}_{s,a_i'} - \mathbb{E}_{a_i'' \sim \pi_{\theta_i}(a_i''|s)}[\mathbf{1}_{s,a_i''}]\right)^\top \right\| \leq 2. \tag{49}$$

Further, by Lemma 3, the term in (49) is Lipschitz continuous with constant 8. By applying Lemma 4 one more time, we know that

$$\left\|\nabla_{\theta_i}^2 \log \pi_{\theta_i}(a_i|s)\right\| \leq 2, \text{ and } \left\|\nabla_{\theta_i}^2 \log \pi_{\theta_i}(a_i|s) - \nabla_{\theta_i'}^2 \log \pi_{\theta_i'}(a_i|s)\right\| \leq 12 \|\theta_i - \theta_i'\|. \tag{50}$$

Since $\nabla_\theta^2 \log \pi_\theta(a|s)$ is a block diagonal matrix, we apply the result on the block diagonal matrix norm in Lemma 5 to show that

$$\left\|\nabla_\theta^2 \log \pi_\theta(a|s)\right\| \leq \max_{i \in \mathcal{N}} \left\|\nabla_{\theta_i}^2 \log \pi_{\theta_i}(a_i|s)\right\| \leq 2.$$

This completes the proof of result (ii). To show result (iii), we again apply Lemma 5 to conclude that

$$\left\|\nabla_\theta^2 \log \pi_\theta(a|s) - \nabla_\theta^2 \log \pi_{\theta'}(a|s)\right\| \leq \max_{i \in \mathcal{N}} \left\|\nabla_{\theta_i}^2 \log \pi_{\theta_i}(a_i|s) - \nabla_{\theta_i'}^2 \log \pi_{\theta_i'}(a_i|s)\right\| \leq 12 \|\theta - \theta'\|,$$

where the last step is by (50). This completes the proof of the lemma. $\qquad\square$

## D.4 Proof of Lemma 13

In the following, since there is no possibility of ambiguity, we drop the dependence on $\mathbb{G}$ and simply write $\nabla \Phi(\theta; \mathbb{G})$ and $V_i^\rho(\theta; \mathbb{G})$ as $\nabla \Phi(\theta)$ and $V_i^\rho(\theta)$, respectively.

To establish Lemma 13, we first derive an explicit formula for the policy Hessian $\nabla^2 \Phi(\theta)$. Notice that $\nabla^2 \Phi(\theta)$ can be written as a block matrix with $N \times N$ blocks:

$$
\nabla^2 \Phi(\theta) = \begin{bmatrix} \nabla^2_{1,1}\Phi(\theta) & \dots & \nabla^2_{1,N}\Phi(\theta) \\ \vdots & \ddots & \vdots \\ \nabla^2_{N,1}\Phi(\theta) & \dots & \nabla^2_{N,N}\Phi(\theta) \end{bmatrix}, \tag{51}
$$

where in each block $\nabla^2_{i,j}\Phi(\theta) \in \mathbb{R}^{|\mathcal{A}_i| \times |\mathcal{A}_j|}$ we first take the gradient of $\Phi$ with respect to agent $i$'s policy parameters $\theta_i$ and then take the gradient with respect to agent $j$'s parameters $\theta_j$, i.e., $\nabla^2_{i,j}\Phi(\theta) = \frac{\partial^2 \Phi}{\partial \theta_i \partial \theta_j}, \forall i, j \in \mathcal{N}$. The following lemma states that each $\nabla^2_{i,j}\Phi(\theta)$ block can be written in an explicit form. This lemma can be considered as a multi-agent extension of Theorem 3 in [24]. For clarity of presentation, we defer its proof to Appendix D.5.

**Lemma 15.** *Each matrix block $\nabla^2_{i,j}\Phi(\theta)$ in the policy Hessian matrix* (51) *takes the form*

$$
\nabla^2_{i,j}\Phi(\theta) = \mathcal{H}_1^{i,j}(\theta) + \mathcal{H}_2^{i,j}(\theta) + \mathcal{H}_{12}^{i,j}(\theta) + (\mathcal{H}_{12}^{i,j})^\top(\theta).
$$

*The matrices $\mathcal{H}_1^{i,j}(\theta), \mathcal{H}_2^{i,j}(\theta)$, and $\mathcal{H}_{12}^{i,j}(\theta)$ can be written as*

$$
\mathcal{H}_1^{i,j}(\theta) = \frac{1}{1-\gamma} \sum_{s \in \mathcal{S}} \sum_{a \in \mathcal{A}} d_\rho^\theta(s, a) Q_i^{s,a}(\theta) \nabla_{\theta_i} \log \pi_\theta(a|s) \nabla_{\theta_j}^\top \log \pi_\theta(a|s),
$$

$$
\mathcal{H}_2^{i,j}(\theta) = \frac{1}{1-\gamma} \sum_{s \in \mathcal{S}} \sum_{a \in \mathcal{A}} d_\rho^\theta(s, a) Q_i^{s,a}(\theta) \nabla_{\theta_i \theta_j}^2 \log \pi_\theta(a|s),
$$

$$
\mathcal{H}_{12}^{i,j}(\theta) = \frac{1}{1-\gamma} \sum_{s \in \mathcal{S}} \sum_{a \in \mathcal{A}} d_\rho^\theta(s, a) \nabla_{\theta_i} \log \pi_\theta(a|s) \nabla_{\theta_j}^\top Q_i^{s,a}(\theta),
$$

*where we define $d_\rho^\theta(s, a) := d_\rho^\theta(s) \cdot \pi_\theta(a|s)$ for $d_\rho^\theta(s) = (1-\gamma)\mathbb{E}_{s^0 \sim \rho} \sum_{t=0}^\infty \gamma^t \mathbb{P}_\theta(s^t = s|s_0)$.*

The next lemma states that each matrix block $\nabla^2_{i,j}\Phi(\theta)$ is Lipschitz continuous with respect to $\theta$. The proof is deferred to Appendix D.6.

**Lemma 16.** *Each matrix block $\nabla^2_{i,j}\Phi(\theta)$ in the policy Hessian matrix* (51) *is Lipschitz continuous:*

$$
\left\| \nabla^2_{i,j}\Phi(\theta) - \nabla^2_{i,j}\Phi(\theta') \right\| \le L_{ij} \|\theta - \theta'\|, \forall i, j \in \mathcal{N},
$$

*where the Lipschitz constant satisfies $L_{ij} \le \frac{56\sqrt{N}}{(1-\gamma)^4}$.*

Equipped with the results from Lemma 15 and Lemma 16, we are now ready to prove Lemma 13.

*Proof* (of Lemma 13).

Proof of (i): From the definition of the potential function (38), we know that $\nabla_{\theta_i}\Phi(\theta) = \nabla_{\theta_i} V_i^\rho(\theta)$, and hence $\nabla \Phi(\theta) = (\nabla_{\theta_1} V_1^\rho(\theta), \dots, \nabla_{\theta_N} V_N^\rho(\theta))$. For each agent $i$, the policy gradient theorem states that

$$
\nabla_{\theta_i} V_i^\rho(\theta) = \frac{1}{1-\gamma} \mathbb{E}_{s \sim d_\rho^\theta, a_i \sim \pi_{\theta_i}(\cdot|s)} \left[ \nabla_{\theta_i} \log \pi_{\theta_i}(a_i|s) \bar{Q}_i^{s,a_i}(\theta) \right].
$$

Since (48) from Lemma 13 suggests that $\|\nabla_{\theta_i} \log \pi_{\theta_i}(a_i|s)\| \le \sqrt{2}$, we obtain $\|\nabla_{\theta_i} V_i^\rho(\theta)\| \le \frac{\sqrt{2}}{(1-\gamma)^2}$. Hence, $\|\nabla \Phi(\theta)\| \le \frac{\sqrt{2N}}{(1-\gamma)^2}$.

Proof of (ii): See Lemma 29 of [78].

Proof of (iii): From the above reasoning, we know that $\nabla^2 \Phi(\theta)$ can be written as a block matrix $\nabla^2 \Phi(\theta) = [\nabla^2_{i,j}\Phi(\theta)]_{1 \le i,j \le N}$, and Lemma 16 implies that each such block is Lipschitz continuous

$$
\left\| \nabla^2_{i,j}\Phi(\theta) - \nabla^2_{i,j}\Phi(\theta') \right\| \le L_{ij} \|\theta - \theta'\|, \forall i, j \in \mathcal{N},
$$

with $L_{ij} \leq \frac{56\sqrt{N}}{(1-\gamma)^4}$. We can then use Lemma 6 to conclude that $\nabla^2 \Phi(\theta)$ is also Lipschitz

$$\left\|\nabla^2 \Phi(\theta) - \nabla^2 \Phi(\theta')\right\| \leq \frac{56N^{3/2}}{(1-\gamma)^4} \left\|\theta - \theta'\right\|.$$

This completes the proof of Lemma 13. $\qquad\qquad\square$

### D.5   Proof of Lemma 15

*Proof.* The proof follows steps similar to those used in the proof of Theorem 3 in [24]. We first introduce a few notations. Let $s^{0:t}$ denote the sequence of states $(s^0, \ldots, s^t)$, and let $a^{0:t} := (a^0, \ldots, a^t)$, where $a^t = (a_1^t, \ldots, a_N^t)$ is the joint action at time step $t$. Further, let

$$p_\theta(s^{0:t}, a^{0:t} \mid \rho) := \mathbb{P}_\theta(s^{0:t}, a^{0:t}|s^0 \sim \rho) = \rho(s^0) \prod_{\tau=0}^{t-1} \left(\pi_\theta(a^\tau|s^\tau)P(s^{\tau+1}|s^\tau, a^\tau)\right) \pi_\theta(a^t|s^t). \tag{52}$$

From the definition in (37), we have

$$V_i^\rho(\theta) = \mathbb{E}_\theta\left[\sum_{t=0}^\infty \gamma^t r_i(s^t, a^t) \mid s^0 \sim \rho\right] = \sum_{t=0}^\infty \sum_{a^{0:t}} \sum_{s^{0:t}} \gamma^t p_\theta(s^{0:t}, a^{0:t}|\rho) r_i(s^t, a^t).$$

Using the definition of the potential function (38), we know that

$$\nabla_{\theta_i} \Phi(\theta) = \nabla_{\theta_i} V_i^\rho(\theta) = \sum_{t=0}^\infty \sum_{a^{0:t}} \sum_{s^{0:t}} \gamma^t p_\theta(s^{0:t}, a^{0:t} \mid \rho) \nabla_{\theta_i} \log p_\theta(s^{0:t}, a^{0:t} \mid \rho) r_i(s^t, a^t),$$

where we used the fact that $\nabla p_\theta = p_\theta \nabla \log p_\theta$. The second-order partial derivative can hence be written as

$$\nabla_{i,j}^2 \Phi(\theta) = \underbrace{\sum_{t=0}^\infty \sum_{a^{0:t}} \sum_{s^{0:t}} \gamma^t p_\theta(s^{0:t}, a^{0:t} \mid \rho) \nabla_{\theta_i\theta_j}^2 \log p_\theta(s^{0:t}, a^{0:t}) r_i(s^t, a^t)}_{①}$$

$$+ \underbrace{\sum_{t=0}^\infty \sum_{a^{0:t}} \sum_{s^{0:t}} \gamma^t p_\theta(s^{0:t}, a^{0:t} \mid \rho) \nabla_{\theta_i} \log p_\theta(s^{0:t}, a^{0:t} \mid \rho) \nabla_{\theta_j}^\top \log p_\theta(s^{0:t}, a^{0:t} \mid \rho) r_i(s^t, a^t)}_{②}$$

From (52), we can see that $\nabla_{\theta_i\theta_j}^2 \log p_\theta(s^{0:t}, a^{0:t} \mid \rho) = \sum_{\tau=0}^t \nabla_{\theta_i\theta_j}^2 \log \pi_\theta(a^\tau|s^\tau)$. Hence, the first term in the above equation can be written as

$$① = \sum_{t=0}^\infty \sum_{a^{0:t}} \sum_{s^{0:t}} \gamma^t p_\theta(s^{0:t}, a^{0:t} \mid \rho) \sum_{\tau=0}^t \nabla_{\theta_i\theta_j}^2 \log \pi_\theta(a^\tau|s^\tau) r_i(s^t, a^t)$$

$$= \sum_{\tau=0}^\infty \gamma^\tau \sum_{s^\tau} \sum_{a^\tau} p_\theta(s^\tau, a^\tau \mid \rho) \nabla_{\theta_i\theta_j}^2 \log \pi_\theta(a^\tau|s^\tau) \sum_{t=\tau}^\infty \gamma^{t-\tau} \sum_{s^t} \sum_{a^t} \mathbb{P}_\theta(s^t, a^t|s^\tau, a^\tau) r^i(s^t, a^t)$$

$$= \sum_{\tau=0}^\infty \gamma^\tau \sum_{s^\tau} \sum_{a^\tau} p_\theta(s^\tau, a^\tau \mid \rho) \nabla_{\theta_i\theta_j}^2 \log \pi_\theta(a^\tau|s^\tau) Q_i^{s^\tau, a^\tau}(\theta)$$

$$= \frac{1}{1-\gamma} \sum_{s \in \mathcal{S}} \sum_{a \in \mathcal{A}} d_\rho^\theta(s, a) Q_i^{s,a}(\theta) \nabla_{\theta_i\theta_j}^2 \log \pi_\theta(a|s)$$

$$= \mathcal{H}_2^{i,j}(\theta).$$

The second term can be written as

$$\textcircled{2} = \sum_{t=0}^{\infty}\sum_{\tau=0}^{t}\sum_{a^{0:t}}\sum_{s^{0:t}}\gamma^t p_\theta(s^{0:t}, a^{0:t}|\rho)\nabla_{\theta_i}\log\pi_\theta(a^\tau|s^\tau)\nabla_{\theta_j}^\top\log\pi_\theta(a^\tau|s^\tau)r_i(s^t, a^t)$$

$$+\sum_{t=0}^{\infty}\sum_{\tau_2=0}^{t}\sum_{\tau_1=0}^{\tau_2-1}\sum_{a^{0:t}}\sum_{s^{0:t}}\gamma^t p_\theta(s^{0:t}, a^{0:t}|\rho)\nabla_{\theta_i}\log\pi_\theta(a^{\tau_1}|s^{\tau_1})\nabla_{\theta_j}^\top\log\pi_\theta(a^{\tau_2}|s^{\tau_2})r_i(s^t, a^t)$$

$$+\sum_{t=0}^{\infty}\sum_{\tau_1=0}^{t}\sum_{\tau_2=0}^{\tau_1-1}\sum_{a^{0:t}}\sum_{s^{0:t}}\gamma^t p_\theta(s^{0:t}, a^{0:t}|\rho)\nabla_{\theta_i}\log\pi_\theta(a^{\tau_1}|s^{\tau_1})\nabla_{\theta_j}^\top\log\pi_\theta(a^{\tau_2}|s^{\tau_2})r_i(s^t, a^t).$$

$$(53)$$

By switching the order of summations and following a similar procedure as in the derivation of $\textcircled{1}$, we can show that the first term on the RHS of (53) is equal to $\mathcal{H}_1^{i,j}(\theta)$. The second and third terms on the RHS of (53) can be shown to be $\mathcal{H}_{12}^{i,j}(\theta)$ and $(\mathcal{H}_{12}^{i,j})^\top(\theta)$, respectively. We skip the rest of the proof as it follows the same procedure as in the proof of Theorem 3 in [24]. $\qquad\square$

### D.6 Proof of Lemma 16

*Proof.* Recall from Lemma 15 that

$$\nabla_{i,j}^2\Phi(\theta) = \mathcal{H}_1^{i,j}(\theta) + \mathcal{H}_2^{i,j}(\theta) + \mathcal{H}_{12}^{i,j}(\theta) + (\mathcal{H}_{12}^{i,j})^\top(\theta).$$

For any $(s,a)$, we write

$$h_1^{i,j}(\theta) = Q_i^{s,a}(\theta)\nabla_{\theta_i}\log\pi_\theta(a|s)\nabla_{\theta_j}^\top\log\pi_\theta(a|s),$$

$$h_2^{i,j}(\theta) = Q_i^{s,a}(\theta)\nabla_{\theta_i\theta_j}^2\log\pi_\theta(a|s),$$

$$h_{12}^{i,j}(\theta) = \nabla_{\theta_i}\log\pi_\theta(a|s)\nabla_{\theta_j}^\top Q_i^{s,a}(\theta),$$

and hence $\nabla_{i,j}^2\Phi(\theta)$ can be rewritten as

$$\nabla_{i,j}^2\Phi(\theta) = \frac{1}{1-\gamma}\sum_{s\in\mathcal{S}}\sum_{a\in\mathcal{A}}d_\rho^\theta(s,a)\left(h_1^{i,j}(\theta) + h_2^{i,j}(\theta) + h_{12}^{i,j}(\theta) + (h_{12}^{i,j})^\top(\theta)\right).$$

In the following, we proceed by showing that each of the three terms $h_1^{i,j}(\theta), h_2^{i,j}(\theta)$, and $h_{12}^{i,j}(\theta)$ is bounded and Lipschitz.

(i) Analysis of $h_1^{i,j}(\theta)$: First, notice that $|Q_i^{s,a}(\theta)| \leq \frac{1}{1-\gamma}$. From the Bellman equation $Q_i^{s,a}(\theta) = r_i(s,a) + \gamma\mathbb{E}_{s'\sim P(\cdot|s,a)}[V_i^{s'}(\theta)]$, we have $\nabla Q_i^{s,a}(\theta) = \gamma\mathbb{E}_{s'\sim P(\cdot|s,a)}[\nabla V_i^{s'}(\theta)]$. The policy gradient theorem states that

$$\nabla_{\theta_i}V_i^\rho(\theta) = \frac{1}{1-\gamma}\mathbb{E}_{s\sim d_\rho^\theta, a_i\sim\pi_{\theta_i}(\cdot|s)}\left[\nabla_{\theta_i}\log\pi_{\theta_i}(a_i|s)\bar{Q}_i^{s,a_i}(\theta)\right].$$

Since (48) from Lemma 12 suggests $\|\nabla_{\theta_i}\log\pi_{\theta_i}(a_i|s)\| \leq \sqrt{2}$, we obtain $\|\nabla_{\theta_i}V_i^\rho(\theta)\| \leq \frac{\sqrt{2}}{(1-\gamma)^2}$. Hence, $\|\nabla Q_i^{s,a}(\theta)\| \leq \frac{\sqrt{2}\gamma}{(1-\gamma)^2}$, and $Q_i^{s,a}(\theta)$ is Lipschitz continuous with constant $\frac{\sqrt{2}\gamma}{(1-\gamma)^2}$. In addition, the proof of Lemma 12 implies that $\nabla_{\theta_i}\log\pi_\theta(a|s)$ is bounded by $\sqrt{2}$ and is 2-Lipschitz continuous. Further using Lemma 3, we can conclude that

$$\left\|h_1^{i,j}(\theta)\right\| \leq \frac{2}{1-\gamma} \text{ and } \left\|h_1^{i,j}(\theta) - h_1^{i,j}(\theta')\right\| \leq \frac{2\sqrt{2}(2-\gamma)}{(1-\gamma)^2}\|\theta - \theta'\|. \tag{54}$$

(ii) Analysis of $h_2^{i,j}(\theta)$: From step (i) of the proof, we know that $Q_i^{s,a}(\theta)$ is bounded by $\frac{1}{1-\gamma}$ and is $\frac{\sqrt{2}\gamma}{(1-\gamma)^2}$-Lipschitz continuous. Since $\pi_\theta$ is a product policy, for $i \neq j$, we simply have $\nabla_{\theta_i\theta_j}^2\log\pi_\theta(a|s) = 0$. For $i = j$, we know from (50) that $\|\nabla_{\theta_i\theta_j}^2\log\pi_\theta(a|s)\| \leq 2$, and $\|\nabla_{\theta_i\theta_j}^2\log\pi_\theta(a|s) - \nabla_{\theta_i\theta_j}^2\log\pi_{\theta'}(a|s)\| \leq 12\|\theta_i - \theta_i'\|$. Therefore, we obtain from Lemma 3 that

$$h_2^{i,j}(\theta) = 0, \text{ if } i \neq j; \text{ and } \left\|h_2^{i,j}(\theta)\right\| \leq \frac{2}{1-\gamma}, \left\|h_2^{i,j}(\theta) - h_2^{i,j}(\theta')\right\| \leq \frac{8(2-\gamma)}{(1-\gamma)^2}\|\theta - \theta'\|, \text{ if } i = j.$$

$$(55)$$

(iii) Analysis of $h_{12}^{i,j}(\theta)$: In the following, we first establish the Lipschitz continuity of $\nabla_{\theta_j} Q_i^{s,a}(\theta)$, which can be shown in a similar manner as in Lemma A.2 of [74] and is reproduced below for completeness. Let

$$p_\theta(s^{0:t}, a^{0:t} \mid s, a) := \mathbb{P}_\theta(s^{0:t}, a^{0:t} | s^0 = s, a^0 = a) = \prod_{\tau=0}^{t-1} \pi_\theta(a^{\tau+1}|s^{\tau+1}) P(s^{\tau+1}|s^\tau, a^\tau).$$

By the definition of the Q-function (37),

$$Q_i^{s,a}(\theta) = \sum_{t=0}^\infty \sum_{a^{0:t}} \sum_{s^{0:t}} \gamma^t p_\theta(s^{0:t}, a^{0:t} \mid s, a) r_i(s^t, a^t)$$

The gradient of $Q_i^{s,a}(\theta)$ can hence be written as

$$\nabla_{\theta_j} Q_i^{s,a}(\theta) = \sum_{t=0}^\infty \sum_{a^{0:t}} \sum_{s^{0:t}} \gamma^t p_\theta(s^{0:t}, a^{0:t} \mid s, a) \nabla_{\theta_j} \log p_\theta(s^{0:t}, a^{0:t} \mid s, a) r_i(s^t, a^t)$$

$$= \sum_{t=0}^\infty \sum_{a^{0:t}} \sum_{s^{0:t}} \gamma^t p_\theta(s^{0:t}, a^{0:t} \mid s, a) \sum_{\tau=1}^t \nabla_{\theta_j} \log \pi_\theta(a^\tau|s^\tau) r_i(s^t, a^t).$$

To show the Lipschitz continuity of $Q_i^{s,a}(\theta)$, we first write

$$\left| \nabla_{\theta_j} Q_i^{s,a}(\theta) - \nabla_{\theta_j} Q_i^{s,a}(\theta') \right|$$

$$\leq \sum_{t=0}^\infty \sum_{a^{0:t}} \sum_{s^{0:t}} \gamma^t \left| p_\theta(s^{0:t}, a^{0:t}|s, a) \sum_{\tau=1}^t \nabla_{\theta_j} \log \pi_\theta(a^\tau|s^\tau) - p_{\theta'}(s^{0:t}, a^{0:t}|s, a) \sum_{\tau=1}^t \nabla_{\theta_j} \log \pi_{\theta'}(a^\tau|s^\tau) \right|$$

$$\leq \sum_{t=0}^\infty \sum_{a^{0:t}} \sum_{s^{0:t}} \gamma^t \left| p_\theta(s^{0:t}, a^{0:t}|s, a) - p_{\theta'}(s^{0:t}, a^{0:t}|s, a) \right| \left\| \sum_{\tau=1}^t \nabla_{\theta_j} \log \pi_\theta(a^\tau|s^\tau) \right\| \tag{56}$$

$$+ \sum_{t=0}^\infty \sum_{a^{0:t}} \sum_{s^{0:t}} \gamma^t p_{\theta'}(s^{0:t}, a^{0:t}|s, a) \left\| \sum_{\tau=1}^t \left( \nabla_{\theta_j} \log \pi_\theta(a^\tau|s^\tau) - \nabla_{\theta_j} \log \pi_{\theta'}(a^\tau|s^\tau) \right) \right\|. \tag{57}$$

In the following, we upper bound each of the two terms above separately. To analyze (56), we first apply the mean-value theorem to the function $\prod_{\tau=1}^t \pi_\theta(a^\tau|s^\tau)$ of $\theta$ and obtain

$$\left| \prod_{\tau=1}^t \pi_\theta(a^\tau|s^\tau) - \prod_{\tau=1}^t \pi_{\theta'}(a^\tau|s^\tau) \right| = \left| (\theta - \theta')^\top \left[ \sum_{m=1}^t \nabla \pi_{\tilde\theta}(a^m|s^m) \prod_{\tau\neq m, \tau=1}^t \pi_{\tilde\theta}(a^\tau|s^\tau) \right] \right|$$

$$\leq \|\theta - \theta'\| \cdot \sum_{m=1}^t \left\| \nabla \log \pi_{\tilde\theta}(a^m|s^m) \right\| \cdot \prod_{\tau=1}^t \pi_{\tilde\theta}(a^\tau|s^\tau)$$

$$\leq \sqrt{2N} t \|\theta - \theta'\| \cdot \prod_{\tau=1}^t \pi_{\tilde\theta}(a^\tau|s^\tau),$$

where $\tilde\theta = \lambda\theta + (1 - \lambda)\theta'$ for some $\lambda \in [0, 1]$, the first inequality uses the fact that $\nabla \pi_{\tilde\theta}(a^m|s^m) = \pi_{\tilde\theta}(a^m|s^m) \nabla \log \pi_{\tilde\theta}(a^m|s^m)$, and the second inequality is due to Lemma 12 (i). Using the above property, we obtain

$$\left| p_\theta(s^{0:t}, a^{0:t}|s, a) - p_{\theta'}(s^{0:t}, a^{0:t}|s, a) \right|$$

$$= \left| \prod_{\tau=0}^{t-1} \pi_\theta(a^{\tau+1}|s^{\tau+1}) P(s^{\tau+1}|s^\tau, a^\tau) - \prod_{\tau=0}^{t-1} \pi_{\theta'}(a^{\tau+1}|s^{\tau+1}) P(s^{\tau+1}|s^\tau, a^\tau) \right|$$

$$\leq \prod_{\tau=0}^{t-1} P(s^{\tau+1}|s^\tau, a^\tau) \cdot \sqrt{2N} t \|\theta - \theta'\| \cdot \prod_{\tau=1}^t \pi_{\tilde\theta}(a^\tau|s^\tau)$$

$$= p_{\tilde\theta}(s^{0:t}, a^{0:t}|s, a) \cdot \sqrt{2N} t \|\theta - \theta'\|.$$

Substituting the above equation back into (56) yields

$$(56) \leq \sum_{t=0}^{\infty} \sum_{a^{0:t}} \sum_{s^{0:t}} \sqrt{2N} t \gamma^t p_{\tilde{\theta}}(s^{0:t}, a^{0:t}|s, a) \cdot \left\| \sum_{\tau=1}^{t} \nabla_{\theta_j} \log \pi_\theta(a^\tau|s^\tau) \right\| \cdot \|\theta - \theta'\|$$

$$\leq \sum_{t=0}^{\infty} \sum_{a^{0:t}} \sum_{s^{0:t}} 2\sqrt{N} t^2 \gamma^t p_{\tilde{\theta}}(s^{0:t}, a^{0:t}|s, a) \|\theta - \theta'\|,$$

where the second step uses (48) from Lemma 12 and the fact that $\pi_\theta$ is a product policy.

To upper bound (57), we apply Lemma 12 (ii) and obtain

$$(57) \leq \sum_{t=0}^{\infty} \sum_{a^{0:t}} \sum_{s^{0:t}} \gamma^t p_{\theta'}(s^{0:t}, a^{0:t}|s, a) \sum_{\tau=1}^{t} \left\| \nabla_{\theta_j} \log \pi_\theta(a^\tau|s^\tau) - \nabla_{\theta_j} \log \pi_{\theta'}(a^\tau|s^\tau) \right\|$$

$$\leq \sum_{t=0}^{\infty} \sum_{a^{0:t}} \sum_{s^{0:t}} 2t \gamma^t p_{\theta'}(s^{0:t}, a^{0:t}|s, a) \|\theta - \theta'\|.$$

Substituting the above upper bounds back into (56) and (57), we have

$$\left| \nabla_{\theta_j} Q_i^{s,a}(\theta) - \nabla_{\theta_j} Q_i^{s,a}(\theta') \right|$$

$$\leq \sum_{t=0}^{\infty} \sum_{a^{0:t}} \sum_{s^{0:t}} \gamma^t \left( 2\sqrt{N} t^2 p_{\tilde{\theta}}(s^{0:t}, a^{0:t}|s, a) + 2t p_{\theta'}(s^{0:t}, a^{0:t}|s, a) \right) \|\theta - \theta'\|$$

$$= \sum_{t=0}^{\infty} \gamma^t \left( 2\sqrt{N} t^2 + 2t \right) \|\theta - \theta'\|$$

$$\leq \frac{4\sqrt{N}\gamma(1+\gamma)}{(1-\gamma)^3} \|\theta - \theta'\|,$$

where the second step holds because $\sum_{a^{0:t}} \sum_{s^{0:t}} p_{\tilde{\theta}}(s^{0:t}, a^{0:t}|s, a) = 1$. The last step uses the facts that $2t \leq 2\sqrt{N}t^2$, and that

$$\sum_{t=1}^{\infty} \gamma^t \cdot t^2 = \frac{1}{1-\gamma} \sum_{t=0}^{\infty} (1-\gamma)\gamma^t \cdot t^2 = \frac{1}{1-\gamma} \cdot \mathbb{E}[T^2] = \frac{1}{1-\gamma} \cdot \frac{\gamma(1+\gamma)}{(1-\gamma)^2},$$

where $T$ is a random variable following a geometric distribution. We have hence derived that $\nabla_{\theta_j} Q_i^{s,a}(\theta)$ is Lipschitz continuous with constant $\frac{4\sqrt{N}\gamma(1+\gamma)}{(1-\gamma)^3}$.

Following the same reasoning as in step (i), we obtain that $\nabla_{\theta_i} \log \pi_\theta(a|s)$ is bounded by $\sqrt{2}$ and is 2-Lipschitz continuous. Similar to step (i), we can also use the Bellman equation and the policy gradient theorem to show that $\|\nabla_{\theta_j}^\top Q_i^{s,a}(\theta)\| \leq \frac{\sqrt{2}\gamma}{(1-\gamma)^2}$. Again, by applying Lemma 3, we can conclude that

$$\left\| h_{12}^{i,j}(\theta) \right\| \leq \frac{2\gamma}{(1-\gamma)^2} \text{ and } \left\| h_{12}^{i,j}(\theta) - h_{12}^{i,j}(\theta') \right\| \leq \frac{6\sqrt{2N}\gamma(1+\gamma)}{(1-\gamma)^3}. \tag{58}$$

(iv) Putting everything together: Let $h(\theta) := h_1^{i,j}(\theta) + h_2^{i,j}(\theta) + h_{12}^{i,j}(\theta) + (h_{12}^{i,j})^\top(\theta)$. Using the simple observation that the sum of two Lipschitz continuous functions is also Lipschitz continuous, we obtain from (54), (55), and (58) that

$$\|h(\theta)\| \leq \frac{4}{(1-\gamma)^2}, \text{ and } \|h(\theta) - h(\theta')\| \leq \frac{50\sqrt{N}}{(1-\gamma)^3} \|\theta - \theta'\|. \tag{59}$$

Recall from Lemma 15 that

$$\nabla_{i,j}^2 \Phi(\theta) = \frac{1}{1-\gamma} \sum_{s\in\mathcal{S}} \sum_{a\in\mathcal{A}} d_\rho^\theta(s,a) h(\theta).$$

By adding and subtracting the same value,

$$\left\| \nabla_{i,j}^2 \Phi(\theta) - \nabla_{i,j}^2 \Phi(\theta') \right\|$$

$$\leq \frac{1}{1-\gamma} \sum_{s\in\mathcal{S}} \sum_{a\in\mathcal{A}} \left\| d_\rho^\theta(s,a)h(\theta) - d_\rho^{\theta'}(s,a)h(\theta') \right\|$$

$$\leq \frac{1}{1-\gamma} \sum_{s\in\mathcal{S}} \sum_{a\in\mathcal{A}} \left( \left| d_\rho^\theta(s,a) - d_\rho^{\theta'}(s,a) \right| \|h(\theta)\| + d_\rho^{\theta'}(s,a) \|h(\theta) - h(\theta')\| \right)$$

$$\leq \frac{4}{(1-\gamma)^3} \sum_{s\in\mathcal{S}} \sum_{a\in\mathcal{A}} \left| d_\rho^\theta(s,a) - d_\rho^{\theta'}(s,a) \right| + \frac{50\sqrt{N}}{(1-\gamma)^4} \|\theta - \theta'\| \sum_{s\in\mathcal{S}} \sum_{a\in\mathcal{A}} d_\rho^{\theta'}(s,a)$$

$$\leq \frac{56\sqrt{N}}{(1-\gamma)^4} \|\theta - \theta'\|.$$

The third step uses the upper bounds from (59). The fourth step can be derived by using the following result from Equation (A.67) of [74]:

$$\sum_{s\in\mathcal{S}} \sum_{a\in\mathcal{A}} \left| d_\rho^\theta(s,a) - d_\rho^{\theta'}(s,a) \right| \leq \frac{\sqrt{2N}}{1-\gamma} \|\theta - \theta'\|.$$

This completes the proof of the Lipschitz continuity that $\left\| \nabla_{i,j}^2 \Phi(\theta) - \nabla_{i,j}^2 \Phi(\theta') \right\| \leq L_{ij} \|\theta - \theta'\|, \forall i,j \in \mathcal{N}$ for $L_{ij} = \frac{56\sqrt{N}}{(1-\gamma)^4}$. $\qquad\square$

### D.7 Proof of Lemma 14

*Proof.* Recall from (46) that the gradient of the meta-objective can be written as

$$\nabla F_T(\theta) = \mathbb{E}_{\mathbb{G}\sim\text{Unif}(\mathcal{G})} \left[ \left( \prod_{t=0}^{T-1} \left( I + \alpha\nabla^2\Phi(\theta^{(t)}(\mathbb{G}); \mathbb{G}) \right) \right) \nabla\Phi(\theta^{(T)}(\mathbb{G}); \mathbb{G}) \right],$$

where $\theta^{(0)}(\mathbb{G}) = \theta$ and $\theta^{(t+1)}(\mathbb{G}) = \Psi(\theta^{(t)}(\mathbb{G}); \mathbb{G})$. It suffices to show that for each individual game $\mathbb{G} \in \mathcal{G}$, the term

$$\left( \prod_{t=0}^{T-1} \left( I + \alpha\nabla^2\Phi(\theta^{(t)}(\mathbb{G}); \mathbb{G}) \right) \right) \nabla\Phi(\theta^{(T)}(\mathbb{G}); \mathbb{G}) \tag{60}$$

is Lipschitz continuous. In the following, we drop the dependence on $\mathbb{G}$ and simply write $\theta^{(t)}(\mathbb{G})$ and $\nabla\Phi(\theta^{(t)}(\mathbb{G}); \mathbb{G})$ as $\theta^{(t)}$ and $\nabla\Phi(\theta^{(t)})$, respectively.

We proceed by finding the upper bound and Lipschitz constant of each individual term in (60). First, from Lemma 13(ii), we know that $\left\| I + \alpha\nabla^2\Phi(\theta^{(t)}) \right\| \leq 1 + \alpha L_G, \forall 0 \leq t \leq T-1$. By using the chain rule, we also know that

$$\nabla_\theta \theta^{(t)} = \prod_{t'=0}^{t-1} (I + \alpha\nabla^2\Phi(\theta^{(t')})).$$

Hence, since $\left\| I + \alpha\nabla^2\Phi(\theta^{(t)}) \right\| \leq 1 + \alpha L_G, \forall 0 \leq t \leq T-1$, we know that $\theta^{(t)}$ is Lipschitz continuous with constant $(1 + \alpha L_G)^t$. Further, combining Lemma 13 (iii) with the fact that the Lipschitz constant of a composite function is equal to the product of the Lipschitz constants of the base functions, we conclude that $I + \alpha\nabla^2\Phi(\theta^{(t)})$ is Lipschitz (with respect to $\theta$) with constant $\alpha L_H(1 + \alpha L_G)^t$. For the case of $T \geq 2$, Lemma 3 thus implies that the $\prod_{t=0}^{T-1}(I + \alpha\nabla^2\Phi(\theta^{(t)}))$ factor from (60) is Lipschitz with constant $\alpha T L_H(1 + \alpha L_G)^{2T-1}$, while for $T = 1$, the Lipschitz constant is simply $\alpha L_H$.

For the $\nabla\Phi(\theta^{(T)})$ factor in (60), we know from Lemma 13(i) that it is bounded by $B_G$. Using Lemma 13(iii) and the Lipschitzness of a composite function, we also know that $\nabla\Phi(\theta^{(T)})$ is $L_G(1 + \alpha L_G)^T$-Lipschitz continuous. Finally, along with the results that the $\prod_{t=0}^{T-1}(I + \alpha\nabla^2\Phi(\theta^{(t)}))$ factor is bounded by $(1 + \alpha L_G)^T$ and Lipschitz with constant $\alpha T L_H(1 + \alpha L_G)^{2T-1}$, we again apply Lemma 3 to obtain that (60) is Lipschitz continuous with constant $\alpha T B_G L_H(1 + \alpha L_G)^{2T-1} + L_G(1 + \alpha L_G)^{2T}$. Using the fact that $\alpha \in (0, 1/L_G]$, we can conclude that the meta-objective $F_T(\theta)$ is $L_F$-smooth with $L_F = (\alpha T B_G L_H + L_G)2^{2T}$. $\qquad\square$

## D.8 Proof of Theorem 4

*Proof.* Based on the aforementioned series of lemmas, we are now ready to establish Theorem 4. The proof follows from standard analysis in non-convex optimization. Since the meta-objective function is $L_F$-smooth (Lemma 14), the smoothness property implies that

$$F_T(\theta^{k+1}) \geq F_T(\theta^k) + \nabla F_T(\theta^k)^\top (\theta^{k+1} - \theta^k) - \frac{L_F}{2} \left\| \theta^{k+1} - \theta^k \right\|^2.$$

Using the outer stage update rule (47) that

$$\theta^{k+1} = \theta^k + \eta \nabla F_T(\theta^k),$$

we obtain

$$F_T(\theta^{k+1}) \geq F_T(\theta^k) + \eta \left\| \nabla F_T(\theta^k) \right\|^2 - \frac{L_F \eta^2}{2} \left\| \nabla F_T(\theta^k) \right\|^2 \geq F_T(\theta^k) + \frac{1}{2L_F} \left\| \nabla F_T(\theta^k) \right\|^2,$$

where the last step uses $\eta = 1/L_F$. Summing the above inequality over $k$ and rearranging the terms lead to

$$\sum_{k=0}^{K-1} \left\| \nabla F_T(\theta^k) \right\|^2 \leq 2L_F \sum_{k=0}^{K-1} (F_T(\theta^{k+1}) - F_T(\theta^k)) = 2L_F(F_T(\theta^K) - F_T(\theta^0)) \leq \frac{4NL_F}{1-\gamma},$$

where the last step holds because $|\Phi(\theta; \mathbb{G}) - \Phi(\theta'; \mathbb{G})| \leq \Phi_{\max} \leq \frac{2N}{1-\gamma}, \forall \theta, \theta' \in \Theta, \mathbb{G} \in \mathcal{G}$. Therefore, for $K \geq \frac{4NL_F}{(1-\gamma)\varepsilon^2}$, we have

$$\min_{0 \leq k \leq K-1} \left\| \nabla F_T(\theta^k) \right\|^2 \leq \frac{1}{K} \sum_{k=0}^{K-1} \left\| \nabla F_T(\theta^k) \right\|^2 \leq \frac{4NL_F}{K(1-\gamma)} \leq \varepsilon^2.$$

This completes the proof of the theorem. $\qquad\square$

# E  Supplementary Material for Section 5

## E.1  Base Algorithm

In this appendix, we first describe our base algorithm for learning CCE in a general-sum Markov game, which was omitted in the main text due to space limitations. The optimistic online mirror descent algorithm for learning CCE in a general-sum Markov game is presented in Algorithm 2. Similar to Algorithm 1 for zero-sum Markov games, Algorithm 2 performs optimistic online mirror descent [55, 62] for policy updates in order to establish initialization-dependent convergence. Algorithm 2 also utilizes stage-based value updates to avoid the need for a complicated no-weighted-regret analysis. Different from Algorithm 1, the output policy $\bar{\pi}$ of Algorithm 2 is no longer a state-wise average policy but rather a correlated policy. The construction of $\bar{\pi}$, similar to the construction of the "certified policies" in the literature, is described in Algorithm 3.

We further introduce a few notations similar to the zero-sum game setting. For any $(\tau, h, s)$, we define the per-state regret for player $i \in \mathcal{N}$ as

$$\operatorname{reg}_{h,i}^\tau(s) := \max_{\pi_{h,i}^{\tau,\dagger}(\cdot|s) \in \Delta(\mathcal{A}_i)} \frac{1}{L_\tau} \sum_{j=t_\tau^{\text{start}}}^{t_\tau^{\text{end}}} \left\langle \pi_{h,i}^{\tau,\dagger} - \pi_{h,i}^j, Q_{h,i}^\tau \pi_{h,-i}^j \right\rangle(s). \tag{61}$$

We define the maximal regret (over the states and all the players) as

$$\operatorname{reg}_h^\tau := \max_{s \in \mathcal{S}} \max_{i \in \mathcal{N}} \{ \operatorname{reg}_{h,i}^\tau(s) \}.$$

Lemma 17 provides an upper bound of the per-state regret (61), which further leads us to the following initialization-dependent convergence guarantee of Algorithm 2. We finally define $\delta_{h,i}^t := \max_{s \in \mathcal{S}} (V_{h+1,i}^{\dagger, \bar{\pi}_h^t, -i} - V_{h,i}^{\bar{\pi}_h^t})(s)$, and let $\delta_h^t := \max_{i \in \mathcal{N}} \delta_{h,i}^t$.

---

**Algorithm 2:** Optimistic Online Mirror Descent for CCE in General-Sum Markov Game

---
1 **Input:** Initial policies $\tilde{\pi} : [\bar{\tau}] \times [H] \times \mathcal{S} \to \Delta(\mathcal{A}_{\text{all}})$;
2 Set stage index $\tau \leftarrow 1$, $t_\tau^{\text{start}} \leftarrow 1$, and $L_\tau \leftarrow H$;
3 **Initialize:** $\pi_h^0 = \hat{\pi}_h^0 \leftarrow \tilde{\pi}_h^1$, and $Q_h^\tau \leftarrow \mathbf{0}, \forall h \in [H]$;
4 **for** *iteration* $t \leftarrow 1$ *to* $T$ **do**
5     **Auxiliary policy update:** for each player $i \in \mathcal{N}$, step $h \in [H]$ and state $s \in \mathcal{S}$:

$$\hat{\pi}_{h,i}^t(\cdot|s) \leftarrow \operatorname*{argmax}_{\mu \in \Delta(\mathcal{A}_i)} \eta \left\langle \mu, [Q_{h,i}^\tau \pi_{h,-i}^{t-1}](s, \cdot) \right\rangle - D_R(\mu, \hat{\pi}_{h,i}^{t-1}(\cdot|s));$$

6     **Policy update:** for each player $i \in \mathcal{N}$, step $h \in [H]$ and state $s \in \mathcal{S}$:

$$\pi_{h,i}^t(\cdot|s) \leftarrow \operatorname*{argmax}_{\mu \in \Delta(\mathcal{A}_i)} \eta \left\langle \mu, [Q_{h,i}^\tau \pi_{h,-i}^{t-1}](s, \cdot) \right\rangle - D_R(\mu, \hat{\pi}_{h,i}^t(\cdot|s));$$

7     **if** $t - t_\tau^{\text{start}} + 1 \geq L_\tau$ **then**
8         $t_\tau^{\text{end}} \leftarrow t, t_{\tau+1}^{\text{start}} \leftarrow t + 1, L_{\tau+1} \leftarrow \lfloor (1 + 1/H)L_\tau \rfloor$;
9         **Value update:** for each $h \in [H], s \in \mathcal{S}, \boldsymbol{a} \in \mathcal{A}_{\text{all}}, i \in \mathcal{N}$:

$$Q_{h,i}^{\tau+1}(s, \boldsymbol{a}) \leftarrow \frac{1}{L_\tau} \sum_{t'=t_\tau^{\text{start}}}^{t_\tau^{\text{end}}} \left( r_{h,i} + P_h[Q_{h+1,i}^\tau \pi_{h+1}^{t'}] \right)(s, \boldsymbol{a});$$

10         $\tau \leftarrow \tau + 1; \pi_h^t = \hat{\pi}_h^t \leftarrow \tilde{\pi}_h^\tau, \forall h \in [H]$;
11 **Output policy:** Sample $t \sim \text{Unif}([T])$. Output $\bar{\pi} := \bar{\pi}_1^t$ as defined in Algorithm 3.

---

---

**Algorithm 3:** Construction of $\bar{\pi}_h^t$

---
1 **Input:** Policy trajectory $\{\pi_h^t\}_{h \in [H], t \in [T]}$ of Algorithm 2;
2 **for** *step* $h' \leftarrow h$ *to* $H$ **do**
3     Uniformly sample $j$ from $\{t_{\tau(t)-1}^{\text{start}}, t_{\tau(t)-1}^{\text{start}} + 1, \ldots, t_{\tau(t)-1}^{\text{end}}\}$;
4     Execute policy $\pi_h^j$ for step $h$;
5     Set $t \leftarrow j$;

---

## E.2   Proof of Theorem 5

*Proof.* From the construction of $\bar{\pi}$ (Algorithm 3) and the definition of CCE-gap, we have

$$\begin{aligned}
\text{CCE-gap}(\bar{\pi}) &= \max_{i \in \mathcal{N}} V_{1,i}^{\dagger, \bar{\pi}^{-i}}(s_1) - V_{1,i}^{\bar{\pi}}(s_1) \\
&\leq \frac{1}{T} \sum_{t=1}^T \max_{i \in \mathcal{N}} \max_{s \in \mathcal{S}} \left( V_{1,i}^{\dagger, \bar{\pi}_{1,-i}^t}(s) - V_{1,i}^{\bar{\pi}_1^t}(s) \right) \\
&\leq \frac{1}{T} \sum_{t=1}^T \delta_1^t.
\end{aligned}$$

Using Lemma 19, the above term can be further bounded by

$$\text{CCE-gap}(\bar{\pi}) \leq \frac{1}{T} \sum_{t=1}^{T} \delta_1^t$$

$$\leq \sum_{t=1}^{T} \frac{3}{\eta T L_{\tau(t)}} \sum_{h=1}^{H} \max_{i \in \mathcal{N}} \max_{s \in \mathcal{S}} D_R(\pi_{h,i}^{\tau(t)-h,\dagger}, \tilde{\pi}_{h',i}^{\tau(t)-h}(\cdot|s)) + 36(N-1)^2 \eta^2 H^4$$

$$= \frac{3}{\eta T} \sum_{\tau=1}^{\bar{\tau}} \sum_{h=1}^{H} \max_{i \in \mathcal{N}} \max_{s \in \mathcal{S}} D_R(\pi_{h,i}^{\tau-h,\dagger}, \tilde{\pi}_{h,i}^{\tau-h}(\cdot|s)) + 36(N-1)^2 \eta^2 H^4$$

$$\leq \frac{3}{\eta T} \sum_{\tau=1}^{\bar{\tau}} \sum_{h=1}^{H} \max_{i \in \mathcal{N}, s \in \mathcal{S}} D_R(\pi_{h,i}^{\tau,\dagger}, \tilde{\pi}_{h,i}^{\tau}(\cdot|s)) + 36(N-1)^2 \eta^2 H^4,$$

where the last step is simply by changing the counting method. This completes the proof for the first claim in the Theorem.

We now proceed to establish the second statement, which follows a similar argument as in the proof of Theorem 1 for the two-player zero-sum game setting. We repeat the proof below for completeness. Recall that we chose the negative entropy as the regularizer $R$. The Bregman divergence $D_R(\cdot, \cdot)$ reduces to the Kullback–Leibler divergence. Since $\pi_{h,i}^{\tau,\dagger}$ lies in the simplex, when we initialize $\tilde{\pi}_{h,i}^{\tau}(\cdot|s) = \mathbf{1}/A_i$ to be a uniform distribution, we naturally have $D_R(\pi_{h,i}^{\tau,\dagger}, \tilde{\pi}_{h,i}^{\tau}(\cdot|s)) \leq \log A_i, \forall i \in \mathcal{N}, s \in \mathcal{S}$, and $h \in [H]$.

It remains to upper bound the total number of stages $\bar{\tau}$. Recall that we have defined the lengths of the stages to increase exponentially with $L_{\tau+1} = \lfloor (1 + 1/H) L_\tau \rfloor$. Since the $\bar{\tau}$ stages sum up to $T$ iterations in total, by taking the sum of a geometric series, it suffices to find a value of $\bar{\tau}$ such that $(1 + 1/H)^{\bar{\tau}} \geq T/H$. Using the Taylor series expansion, one can show that $(1 + \frac{1}{H})^H \geq e - \frac{e}{2H}$. Hence, it reduces to finding a minimum $\bar{\tau}$ such that

$$\left(e - \frac{e}{2H}\right)^{\bar{\tau}/H} \geq \frac{T}{H}. \tag{62}$$

One can easily see that any $\bar{\tau} \geq \frac{H \log T}{\log(e/2)}$ satisfies the condition. Summarizing the above results, we can conclude that

$$\text{CCE-gap}(\bar{\pi}) \leq \frac{12 H^2 \log T}{\eta T} \log A_{\max} + 36(N-1)^2 \eta^2 H^4.$$

Choosing $\eta = H^{-2/3} T^{-1/3} (N-1)^{-2/3}$ yields the second claim in the Theorem. $\qquad\square$

### E.3 Supporting Lemmas for Section 5

**Lemma 17.** *For every stage $\tau \in \mathbb{N}_+$, every step $h \in [H]$ and every state $s \in \mathcal{S}$, the per-state average regret of player $i \in \mathcal{N}$ is bounded by:*

$$\text{reg}_{h,i}^{\tau}(s) \leq \frac{1}{\eta L_\tau} D_R(\pi_{h,i}^{\tau,\dagger}, \tilde{\pi}_{h,i}^{\tau}(\cdot|s)) + 36(N-1)^2 \eta^2 H^3. \tag{63}$$

*Proof.* Notice that the policy update steps in Algorithm 2 are exactly the same as the optimistic online mirror descent algorithm [55, 62], with the loss vector $g^t = [Q_{h,i}^{\tau} \pi_{h,-i}^t](s, \cdot)$ and the recency bias $M^t = [Q_{h,i}^{\tau} \pi_{h,-i}^{t-1}](s, \cdot)$. Since our stage-based value updates assign equal weights to each iteration, we end up with a classic no-(average-)regret learning problem instead of a no-(weighed-)regret learning problem as in [72, 77]. This allows us to directly apply the standard optimistic OMD results

(e.g., Lemma 1 in [55] and Proposition 5 in [62]) to obtain

$$
\mathrm{reg}_{h,i}^\tau(s) = \max_{\pi_{h,i}^{\tau,\dagger} \in \Delta(\mathcal{A}_i)} \frac{1}{L_\tau} \sum_{j=t_\tau^{\mathrm{start}}}^{t_\tau^{\mathrm{end}}} \left\langle \pi_{h,i}^{\tau,\dagger} - \pi_{h,i}^j, Q_{h,i}^\tau \pi_{h,-i}^j \right\rangle(s)
$$

$$
\leq \frac{1}{\eta L_\tau} D_R(\pi_{h,i}^{\tau,\dagger}, \tilde{\pi}_{h,i}^\tau(\cdot|s)) + \frac{\eta}{L_\tau} \sum_{j=t_\tau^{\mathrm{start}}}^{t_\tau^{\mathrm{end}}} \left\| [Q_{h,i}^\tau \pi_{h,-i}^j - Q_{h,i}^\tau \pi_{h,-i}^{j-1}](s,\cdot) \right\|_\infty^2
$$

$$
- \frac{1}{8\eta L_\tau} \sum_{j=t_\tau^{\mathrm{start}}}^{t_\tau^{\mathrm{end}}} \left\| \pi_{h,i}^j(\cdot \mid s) - \pi_{h,i}^{j-1}(\cdot \mid s) \right\|_1^2
$$

$$
\leq \frac{1}{\eta L_\tau} D_R(\pi_{h,i}^{\tau,\dagger}, \tilde{\pi}_{h,i}^\tau(\cdot|s)) + \frac{\eta}{L_\tau} \sum_{j=t_\tau^{\mathrm{start}}}^{t_\tau^{\mathrm{end}}} 2H^2 \left\| \pi_{h,-i}^j(\cdot \mid s) - \pi_{h,-i}^{j-1}(\cdot \mid s) \right\|_1^2, \qquad (64)
$$

where in the last step we used the Hölder's inequality and the fact that $\|Q_h^\tau(s,\cdot)\|_\infty \leq H$. To further upper bound (64), we apply Lemma 18 to obtain that for any $t \in [t_\tau^{\mathrm{start}}, t_\tau^{\mathrm{end}}]$,

$$
\left\| \pi_{h,-i}^t(\cdot \mid s) - \pi_{h,-i}^{t-1}(\cdot \mid s) \right\|_1^2 \leq 18(N-1)^2 \eta H. \qquad (65)
$$

We remark that the policy stability condition above has a slightly worse dependence on $\eta$ than those of the optimistic FTRL algorithms. In particular, Lemma G.4 of [77] has shown a $\left\| \pi_{h,-i}^t(\cdot \mid s) - \pi_{h,-i}^{t-1}(\cdot \mid s) \right\|_1^2 \leq 16(N-1)^2\eta^2 H^2$ condition for optimistic FTRL. This is because unlike optimistic FTRL, optimistic OMD lacks a smoothness condition that directly connects the stability of policies to the stability of utility functions (e.g., Lemma A.5 of [77]). Plugging (65) back into (64) leads to the desired result. $\qquad \square$

**Lemma 18.** *For a fixed $\tau$ and any $t \in [t_\tau^{start}, t_\tau^{end}], i \in \mathcal{N}, h \in [H], s \in \mathcal{S}$, the optimistic online mirror descent policy updates in Algorithm 2 satisfy:*

$$
\left\| \pi_{h,i}^t(\cdot \mid s) - \pi_{h,i}^{t-1}(\cdot \mid s) \right\|_1^2 \leq 18\eta H.
$$

*Consequently,*

$$
\left\| \pi_{h,-i}^t(\cdot \mid s) - \pi_{h,-i}^{t-1}(\cdot \mid s) \right\|_1^2 \leq 18(N-1)^2 \eta H.
$$

*Proof.* In this proof, since we focus on a fixed $(s,h) \to \mathcal{S} \times [H]$, we will drop the dependence on $(s,h)$ for notational convenience. To prove the first claim in the lemma, we first use the triangle inequality to obtain that

$$
\left\| \pi_i^t - \pi_i^{t-1} \right\|_1 \leq \left\| \pi_i^t - \hat{\pi}_i^t \right\|_1 + \left\| \hat{\pi}_i^t - \hat{\pi}_i^{t-1} \right\|_1 + \left\| \hat{\pi}_i^{t-1} - \pi_i^{t-1} \right\|_1. \qquad (66)
$$

In the following, we derive an upper bound for the first term on the RHS of the above inequality. The other two terms on the RHS can be bounded in a similar way.

We know from the Pinsker's inequality that

$$
\left\| \pi_i^t - \hat{\pi}_i^t \right\|_1 \leq \sqrt{2\,\mathrm{KL}\left(\pi_i^t \| \hat{\pi}_i^t\right)}. \qquad (67)
$$

In the following, it suffices to find an upper bound of $\mathrm{KL}\left(\hat{\pi}_i^t \| \pi_i^t\right)$. Recall that Algorithm 2 updates the policies as

$$
\pi_i^t = \operatorname*{argmax}_{\mu \in \Delta(\mathcal{A}_i)} \eta \left\langle \mu, [Q_i^\tau \pi_{-i}^{t-1}] \right\rangle - D_R(\mu, \hat{\pi}_i^t).
$$

Since we chose the negative entropy as the regularizer $R$, the policy update rule above is known (see Section 5.4.2 of [28]) to be equivalent to the following multiplicative weights update:

$$
\pi_i^t(a) = \frac{\hat{\pi}_i^t(a)\exp(\eta[Q_i^\tau \pi_{-i}^{t-1}](a))}{\sum_{a'} \hat{\pi}_i^t(a')\exp(\eta[Q_i^\tau \pi_{-i}^{t-1}](a'))}, \forall a \in \mathcal{A}_i.
$$

Hence, we have that

$$
\begin{aligned}
\mathrm{KL}\left(\pi_i^t \| \hat{\pi}_i^t\right) &= \sum_{a \in \mathcal{A}_i} \pi_i^t(a) \ln \frac{\pi_i^t(a)}{\hat{\pi}_i^t(a)} \\
&= \sum_{a \in \mathcal{A}_i} \pi_i^t(a) \ln \frac{\exp(\eta [Q_i^\tau \pi_{-i}^{t-1}](a))}{\sum_{a'} \hat{\pi}_i^t(a') \exp(\eta [Q_i^\tau \pi_{-i}^{t-1}](a'))} \\
&\leq \sum_{a \in \mathcal{A}_i} \pi_i^t(a) \ln \frac{\exp(\eta H)}{\sum_{a'} \hat{\pi}_i^t(a')} \\
&= \eta H,
\end{aligned}
$$

where the inequality uses the facts that $Q_i^\tau \geq 0$ and $\|Q_i^\tau\|_1 \leq H$. Substituting the above result back to (67) leads to

$$
\left\|\pi_i^t - \hat{\pi}_i^t\right\|_1 \leq \sqrt{2 \, \mathrm{KL}\left(\pi_i^t \| \hat{\pi}_i^t\right)} \leq \sqrt{2\eta H}.
$$

Similar results also hold for the other two terms on the RHS of (66). Therefore, we can conclude that $\left\|\pi_i^t - \pi_i^{t-1}\right\|_1 \leq 3\sqrt{2\eta H}$ and

$$
\left\|\pi_i^t - \pi_i^{t-1}\right\|_1^2 \leq 18\eta H.
$$

This proves the first claim in the lemma. To establish the second claim, we use the following simple fact for product distributions:

$$
\left\|\pi_{-i}^t - \pi_{-i}^{t-1}\right\|_1 \leq \sum_{j \neq i} \left\|\pi_j^t - \pi_j^{t-1}\right\|_1 .
$$

Applying Jensen's inequality yields

$$
\left\|\pi_{-i}^t - \pi_{-i}^{t-1}\right\|_1^2 \leq \left(\sum_{j \neq i} \left\|\pi_j^t - \pi_j^{t-1}\right\|_1\right)^2 \leq (N-1) \sum_{j \neq i} \left\|\pi_j^t - \pi_j^{t-1}\right\|_1^2 \leq 18(N-1)^2 \eta H.
$$

This proves the second claim in the lemma. $\qquad\square$

**Lemma 19.** *For any iteration $t \in [T]$ and any step $h \in [H]$, we have that*

$$
\delta_h^t \leq \frac{3}{\eta L_{\tau(t)}} \sum_{h'=h}^{H} \max_{i \in \mathcal{N}} \max_{s \in \mathcal{S}} D_R(\pi_{h',i}^{\tau(t)-h'+h-1,\dagger}, \tilde{\pi}_{h',i}^{\tau(t)-h'+h-1}(\cdot|s)) + 36(N-1)^2 \eta^2 H^4.
$$

*Proof.* In the following, when we consider a fixed iteration $t \in [T]$, we drop the notational dependence on $t$ and simply use $\tau$ (instead of $\tau(t)$) to denote the stage that iteration $t$ belongs to. For any $h \in [H-1]$, using a similar argument as in Lemma 10 for the zero-sum game setting, one can establish the following recursion for the value estimation error:

$$
\delta_h^t \leq \frac{1}{L_{\tau-1}} \sum_{j=t_{\tau-1}^{\mathrm{start}}}^{t_{\tau-1}^{\mathrm{end}}} \delta_{h+1}^j + \mathrm{reg}_h^{\tau-1}, \tag{68}
$$

where we recall that $\mathrm{reg}_h^\tau := \max_{s \in \mathcal{S}} \max_{i \in \mathcal{N}} \{\mathrm{reg}_{h,i}^\tau(s)\}$. Using Lemma 17, we can upper bound the regret by

$$
\mathrm{reg}_h^\tau \leq \max_{i \in \mathcal{N}} \max_{s \in \mathcal{S}} \frac{1}{\eta L_\tau} D_R(\pi_{h,i}^{\tau,\dagger}, \tilde{\pi}_{h,i}^\tau(\cdot|s)) + 36(N-1)^2 \eta^2 H^3.
$$

We substitute the regret bound above back into the recursion 68 to get that

$$
\delta_h^t \leq \max_{i \in \mathcal{N}} \max_{s \in \mathcal{S}} \frac{1}{\eta L_{\tau-1}} D_R(\pi_{h,i}^{\tau-1,\dagger}, \tilde{\pi}_{h,i}^{\tau-1}(\cdot|s)) + 36(N-1)^2 \eta^2 H^3 + \frac{1}{L_{\tau-1}} \sum_{j=t_{\tau-1}^{\mathrm{start}}}^{t_{\tau-1}^{\mathrm{end}}} \delta_{h+1}^j. \tag{69}
$$

Notice that according to the definition in Algorithm 3, the behavior of the policy $\bar{\pi}_h^t$ does not change with $t$ within the same stage $\tau$ as it always uniformly sample a time index from the previous stage

and execute the corresponding history policy. Consequently, the $\delta_{h+1}^j$ term is also unchanged within a stage. Hence, we have

$$\frac{1}{L_{\tau-1}} \sum_{j=t_{\tau-1}^{\text{start}}}^{t_{\tau-1}^{\text{end}}} \delta_{h+1}^j = \delta_{h+1}^{\tau-1}.$$

The recursion in (69) can hence be rewritten more succinctly as

$$\delta_h^t \leq \max_{i \in \mathcal{N}} \max_{s \in \mathcal{S}} \frac{1}{\eta L_{\tau-1}} D_R(\pi_{h,i}^{\tau-1,\dagger}, \tilde{\pi}_{h,i}^{\tau-1}(\cdot|s)) + 36(N-1)^2 \eta^2 H^3 + \delta_{h+1}^{\tau-1}.$$

Applying the above inequality recursively over $h$ leads to

$$\delta_h^t \leq \sum_{h'=h}^{H} \max_{i \in \mathcal{N}} \max_{s \in \mathcal{S}} \frac{1}{\eta L_{\tau-h'+h-1}} D_R(\pi_{h',i}^{\tau-h'+h-1,\dagger}, \tilde{\pi}_{h',i}^{\tau-h'+h-1}(\cdot|s)) + 36(N-1)^2 \eta^2 H^3 (H-h+1)$$

$$\leq \sum_{h'=h}^{H} \max_{i \in \mathcal{N}} \max_{s \in \mathcal{S}} \frac{1}{\eta L_\tau} \left(1 + \frac{1}{H}\right)^{h'-h+1} D_R(\pi_{h',i}^{\tau-h'+h-1,\dagger}, \tilde{\pi}_{h',i}^{\tau-h'+h-1}(\cdot|s)) + 36(N-1)^2 \eta^2 H^4$$

$$\leq \frac{3}{\eta L_\tau} \sum_{h'=h}^{H} \max_{i \in \mathcal{N}} \max_{s \in \mathcal{S}} D_R(\pi_{h',i}^{\tau-h'+h-1,\dagger}, \tilde{\pi}_{h',i}^{\tau-h'+h-1}(\cdot|s)) + 36(N-1)^2 \eta^2 H^4, \tag{70}$$

where the second step uses our choice of the stage lengths that $L_{\tau+1} = \lfloor (1+1/H)L_\tau \rfloor$, which further implies that

$$\frac{1}{L_{\tau-h'+h-1}} \leq \frac{1}{L_\tau} \left(1 + \frac{1}{H}\right)^{h'-h+1}.$$

The last step in (70) is due to the fact that $(1+1/H)^H \leq e \approx 2.71828$. $\qquad \square$

## E.4  Proof of Theorem 6

*Proof.* First, recall the definitions of $\tilde{\pi}^k$, $\bar{\pi}^k$ and $\pi_i^{k,\dagger}$. Since we use a negative entropy regularizer $R$, the Bregman divergence $D_R(\cdot,\cdot)$ reduces to the Kullback–Leibler divergence. Using these notations, our convergence results of learning CCE in an individual game $\mathbb{G}^k$ (Theorem 5) can be written more succinctly as

$$\text{CCE-gap}(\bar{\pi}^k) \leq \frac{3}{\eta T} \text{KL}\left(\pi^{k,\dagger} \| \tilde{\pi}^k\right) + 36 N^2 \eta^2 H^4.$$

where for ease of notations, we write

$$\text{KL}\left(\pi^{k,\dagger} \| \tilde{\pi}^k\right) := \sum_{h=1}^{H} \sum_{\tau=1}^{\bar{\tau}} \sum_{i=1}^{N} \max_{s \in \mathcal{S}} \text{KL}\left(\pi_{h,i}^{k,\tau,\dagger}(\cdot|s) \| \tilde{\pi}_{h,i}^k(\cdot|s)\right).$$

Here, $\pi_{h,i}^{k,\tau,\dagger}(\cdot|s)$ represents the value of $\pi_{h,i}^{\tau,\dagger}(\cdot|s)$ in game $\mathbb{G}^k$. By running Algorithm 2 on a sequence of $K$ games, we have that

$$\frac{1}{K} \sum_{k=1}^{K} \text{CCE-gap}(\bar{\pi}^k) \leq \frac{3}{\eta K T} \sum_{k=1}^{K} \text{KL}\left(\pi^{k,\dagger} \| \tilde{\pi}^k\right) + 36 N^2 \eta^2 H^4. \tag{71}$$

Recall the notation that $[\mathbf{x}]_\alpha = (1-\alpha)\mathbf{x} + \frac{\alpha}{d}\mathbf{1}$ for $\mathbf{x} \in \mathbb{R}^d$. By applying this notation entry-wise to each probability distribution in $\pi^{k,\dagger}$ and invoking Lemma 1, we obtain that

$$\frac{1}{K} \sum_{k=1}^{K} \text{KL}\left(\pi^{k,\dagger} \| \tilde{\pi}^k\right) \leq \frac{1}{K} \sum_{k=1}^{K} \text{KL}\left([\pi^{k,\dagger}]_\alpha \| \tilde{\pi}^k\right) + 4H\bar{\tau}\alpha \ln \frac{A_{\max}}{\alpha}. \tag{72}$$

Notice that the conditions of Lemma 1 are satisfied here because we select our initial policies to be $\tilde{\pi}_i^k = \frac{1}{k-1} \sum_{k'=1}^{k-1} [\pi_i^{k',\dagger}]_\alpha, \forall i \in \mathcal{N}$, which assigns a probability of at least $\alpha \mathbf{1}/A_i$ to each action.

Adding and subtracting the same term leads to

$$\sum_{k=1}^{K} \mathrm{KL}\left([\pi^{k,\dagger}]_{\alpha}\|\tilde{\pi}^{k}\right) = \min_{\pi}\sum_{k=1}^{K} \mathrm{KL}\left([\pi^{k,\dagger}]_{\alpha}\|\pi\right) + \min_{\pi}\sum_{k=1}^{K}\left(\mathrm{KL}\left([\pi^{k,\dagger}]_{\alpha}\|\tilde{\pi}^{k}\right) - \mathrm{KL}\left([\pi^{k,\dagger}]_{\alpha}\|\pi\right)\right)$$

$$\leq \min_{\pi}\sum_{k=1}^{K} \mathrm{KL}\left([\pi^{k,\dagger}]_{\alpha}\|\pi\right) + \frac{8A_{\max}(1+\ln K)}{\alpha}, \tag{73}$$

where the minimum $\pi$ is taken over all policies of the form of $\pi = (\pi_1, \ldots, \pi_N)$ such that $\pi_i : [\bar{\tau}] \times [H] \times \mathcal{S} \to \Delta(\mathcal{A}_i)$. We now turn to establish the second step in (73), which reduces to bounding the following regret where the loss functions are given by the Bregman divergences:

$$\mathrm{reg} = \min_{\pi}\sum_{k=1}^{K}\left(\mathrm{KL}\left([\pi^{k,\dagger}]_{\alpha}\|\tilde{\pi}^{k}\right) - \mathrm{KL}\left([\pi^{k,\dagger}]_{\alpha}\|\pi\right)\right).$$

It is known that the unique minimum of $\sum_{k'=1}^{k} \mathrm{KL}([\pi^{k',\dagger}]_{\alpha}\|\cdot)$ is attained at $\frac{1}{k}\sum_{k'=1}^{k}[\pi^{k',\dagger}]_{\alpha}$ (see Proposition 1 of [5] for a proof of this claim). Therefore, by letting $\tilde{\pi}_i^k = \frac{1}{k-1}\sum_{k'=1}^{k-1}[\pi_i^{k',\dagger}]_{\alpha}$, we are essentially running the follow the leader (FTL) algorithm (separately for each entry $(\tau, h, s) \in [\bar{\tau}] \times [H] \times \mathcal{S}$) on the sequence of losses defined by $\sum_{k=1}^{K} \mathrm{KL}([\pi^{k,\dagger}]_{\alpha}\|\cdot)$. We can then invoke the logarithmic regret guarantee of FTL with respect to Bregman divergences, which was established in [35] and is reproduced as Lemma 2 in Appendix A for completeness.

To show that Lemma 2 is applicable, we remark that the Kullback–Leibler divergence is not Lipschitz continuous near the boundary of the probability simplex, which breaks condition required by Lemma 2. However, by restricting to policies of the form $[\pi_i]_{\alpha} = (1-\alpha)\pi_i + \frac{\alpha}{A_i}\mathbf{1}$, which is at least $\frac{\alpha}{A_i}$-distance away from the simplex boundary, the Kullback–Leibler divergence is indeed Lipschitz continuous within this $\frac{\alpha}{A_i}$-restricted domain. One can show that the Lipschitz constant of each entry of $\mathrm{KL}([\pi_i^{k,\dagger}]_{\alpha}\|\cdot)$ is $\frac{2A_{\max}}{\alpha}$ within the $\frac{\alpha}{A_{\max}}$-restricted domain. This allows us to apply Lemma 2 to obtain the result in (73).

Moving forward from (73), we again apply the property that the unique minimum of $\sum_{k'=1}^{k} \mathrm{KL}([\pi^{k',\dagger}]_{\alpha}\|\cdot)$ is attained at $\frac{1}{k}\sum_{k'=1}^{k}[\pi^{k',\dagger}]_{\alpha}$, which leads to

$$\sum_{k=1}^{K} \mathrm{KL}\left([\pi^{k,\dagger}]_{\alpha}\|\tilde{\pi}^{k}\right) \leq \min_{\pi}\sum_{k=1}^{K} \mathrm{KL}\left([\pi^{k,\dagger}]_{\alpha}\|\pi\right) + \frac{8A_{\max}(1+\ln K)}{\alpha}$$

$$= \sum_{k=1}^{K} \mathrm{KL}\left([\pi^{k,\dagger}]_{\alpha}\|[\pi^{\star}]_{\alpha}\right) + \frac{8A_{\max}(1+\ln K)}{\alpha}$$

$$\leq (1-\alpha)\sum_{k=1}^{K} \mathrm{KL}\left(\pi^{k,\dagger}\|\pi^{\star}\right) + \frac{8A_{\max}(1+\ln K)}{\alpha}, \tag{74}$$

where the second step uses the definition that $\pi_i^{\star} = \frac{1}{K}\sum_{k=1}^{K}\pi_i^{k,\dagger}$, and the last step is by the (joint) convexity of the Kullback–Leibler divergence. Substituting (74) to (72) yields

$$\frac{1}{K}\sum_{k=1}^{K} \mathrm{KL}\left(\pi^{k,\dagger}\|\tilde{\pi}^{k}\right) \leq \frac{1}{K}\sum_{k=1}^{K} \mathrm{KL}\left(\pi^{k,\dagger}\|\pi^{\star}\right) + \frac{8A_{\max}(1+\ln K)}{K\alpha} + 4H\bar{\tau}\alpha\ln\frac{A_{\max}}{\alpha}.$$

Further substituting the above result back into (71) and using the definition

$$\Delta_{\pi} = \sum_{k=1}^{K}\sum_{i=1}^{N} \mathrm{KL}\left(\pi_i^{k,\dagger}\|\pi_i^{\star}\right),$$

we obtain that

$$\frac{1}{K}\sum_{k=1}^{K} \mathrm{CCE\text{-}gap}(\bar{\pi}^{k}) \leq \frac{3}{\eta KT}\left(\Delta_{\pi} + \frac{8A_{\max}(1+\ln K)}{\alpha} + 4KH\bar{\tau}\alpha\ln\frac{A_{\max}}{\alpha}\right) + 36N^2\eta^2 H^4.$$

Finally, using the conditions that $\alpha = 1/\sqrt{K}$, $\eta = K^{-1/6}H^{-2/3}T^{-1/3}N^{-2/3}$, and $\bar\tau \leq 4H \log T$ (see (62) for a proof) yields

$$\frac{1}{K}\sum_{k=1}^{K}\text{CCE-gap}(\bar\pi^k) \leq \left(\frac{HN}{T}\right)^{\frac{2}{3}}\left(\frac{\Delta_\pi}{K^{5/6}} + \frac{10A_{\max}\ln K}{K^{1/3}} + \frac{52H^2 \ln T \log(A_{\max}K)}{K^{1/3}}\right).$$

This completes the proof of the theorem. □

## F  Simulations

In this appendix, we provide detailed discussions of our simulation results. We first evaluate our algorithms on a sequence of handcrafted two-player zero-sum Markov games (Appendix F.1) and Markov potential games (Appendix F.2). Then, in Appendix F.3, we further demonstrate the scalability of our methods by considering larger-scale tasks, including a simplified version of the Poker endgame considered in [27] and a 1D linear-quadratic tracking task [37].

### F.1  Zero-Sum Markov Games

We first evaluate our meta-learning procedure presented in Section 3 on a sequence of $K = 10$ two-player zero-sum Markov games. We generate a sequence of $K$ similar games by first specifying a "base game" and then adding random perturbations to its reward function to get $K$ slightly different games. For our base game, we consider a simple zero-sum game with two states $\mathcal{S} = \{s_0, s_1\}$, where each player has two candidate actions $\mathcal{A} = \{a_0, a_1\}$ and $\mathcal{B} = \{b_0, b_1\}$, respectively. The reward matrices for the max-player at the two states are given in Table 1. We add independent $\mathcal{N}(0, 0.1)$ Gaussian perturbation to each entry of the reward matrix to generate $K = 10$ slightly different games.

| $s_0$ | $b_0$ | $b_1$ | | $s_1$ | $b_0$ | $b_1$ |
|---|---|---|---|---|---|---|
| $a_0$ | 0.5 | 0 | | $a_0$ | 0.5 | 0 |
| $a_1$ | -1 | 0.5 | | $a_1$ | 0.2 | 1 |

Table 1: Reward matrices for the max-player in the base game.

To better visualize the similarity level of these games, we plot the NE policies of the two perturbed matrix games in each of the $K = 10$ games. In particular, let $\mu^\star = (\mu_0^\star, \mu_1^\star) \in [0, 1]^2$ and $\nu^\star = (\nu_0^\star, \nu_1^\star) \in [0, 1]^2$ denote the NE policies of the two players in a certain game. Since $\mu_0^\star + \mu_1^\star = 1$ and $\nu_0^\star + \nu_1^\star = 1$, it suffices to simply use the two values $\mu_0^\star \in [0, 1]$ and $\nu_0^\star \in [0, 1]$ to characterize the NE policies. Figure 2 (c) plots the relative position of the $(\mu_0^\star, \nu_0^\star)$ pairs of the $K \times 2$ games in the space of $[0, 1] \times [0, 1]$ to illustrate their closeness, where the $[0, 1] \times [0, 1]$ space is large enough to cover all possible zero-sum games of the same form. We note that Figure 2 (c) only plots the NE pairs with respect to the perturbed matrix games as defined in Table 1. Due to the existence of the state transitions, the NE policies with respect to the stage Q-functions can be more diversified. In this sense, we can see that our similarity assumption of the games is not too stringent as it allows the games to have relatively diverse NE policies.

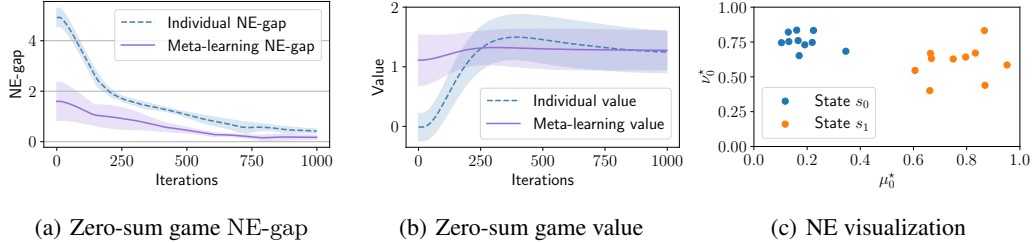

(a) Zero-sum game NE-gap          (b) Zero-sum game value          (c) NE visualization

Figure 2: Average (a) NE-gaps and (b) values of the policies output by individual learning and meta-learning in zero-sum Markov games. Shaded areas denote the standard deviations. (c) visualizes the NE policies of the $K$ games in the normalized space $[0, 1] \times [0, 1]$ to illustrate their closeness.

The state transition function is defined as follows: In both states $s_0$ and $s_1$, if the two players take matching actions (namely $(a_0, b_0)$ or $(a_1, b_1)$), the system stays at the current state with probability 0.9, and transitions to the other state with probability 0.1. On the other hand, if the two players take opposite actions (namely $(a_0, b_1)$ or $(a_1, b_0)$), the environment will stay at the current state with probability 0.1, and will transition to the other state with probability 0.9.

Each of the $K$ games lasts for $H = 10$ steps, and we run our algorithm for $T = 1000$ iterations on each game. We use a learning rate of $\eta = 0.02$ for Algorithm 1. We evaluate the convergences of the algorithms in terms of NE-gap$(\mu, \nu) := V_1^{\dagger, \nu}(s_1) - V_1^{\mu, \dagger}(s_1)$, which measures the distances from the output policies to each agent's best response policy. Figure 2 (a) compares the average NE-gap over the $K$ games between individual learning and meta-learning. Figure 2 (b) further compares the average values achieved by the two methods. All results are obtained on a laptop with an Intel Core i5-1240P CPU. We see that compared to learning each task individually, meta-learning can utilize knowledge from previous tasks to attain better policy initialization in a new task and converges to an approximate NE policy (and value) using much fewer iterations.

### F.2 Markov Potential Games

We now evaluate our meta-learning algorithm from Section 4 on a sequence of Markov potential games. We illustrate our algorithm in cooperative games, an important class of MPGs where the agents share the same rewards. We again generate a sequence of $K$ similar games by first specifying a base game and then adding random perturbations to its reward function to get $K$ slightly different games. Our base game has two states $\mathcal{S} = \{s_0, s_1\}$ and each player has two candidate actions $\mathcal{A} = \{a_0, a_1\}$ and $\mathcal{B} = \{b_0, b_1\}$. The shared reward matrices for both players at the two states are given in Table 2. We add independent $\mathcal{N}(0, 0.1)$ Gaussian perturbation to each entry of the reward matrix to generate $K = 10$ slightly different games.

| $s_0$ | $b_0$ | $b_1$ | | $s_1$ | $b_0$ | $b_1$ |
|---|---|---|---|---|---|---|
| $a_0$ | 0.1 | 0.5 | | $a_0$ | 0.8 | 0.2 |
| $a_1$ | 0.5 | 1 | | $a_1$ | 0.2 | 0.8 |

Table 2: Reward matrices for both players in the base game.

The state transition function is defined in the same way as in Appendix F.1: In both states $s_0$ and $s_1$, if the two players take matching actions (namely $(a_0, b_0)$ or $(a_1, b_1)$), the system stays at the current state with probability 0.9, and transitions to the other state with probability 0.1. On the other hand, if the two players take opposite actions (namely $(a_0, b_1)$ or $(a_1, b_0)$), the environment will stay at the current state with probability 0.1, and will transition to the other state with probability 0.9.

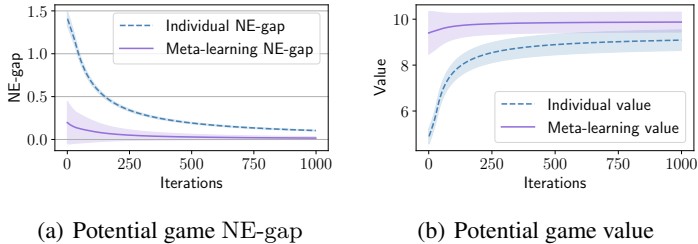

(a) Potential game NE-gap      (b) Potential game value

Figure 3: Average (a) NE-gaps and (b) values of the policies output by individual learning and meta-learning in Markov potential games. Shaded areas denote the standard deviations.

Each of the $K$ games lasts for $H = 10$ steps, and we run our algorithm for $T = 1000$ iterations on each game. We use a learning rate of $\alpha = 0.05$ for the independent projected Q-descent algorithm (7). We evaluate the convergences of the algorithms in terms of NE-gap$(\mu, \nu) :=$ $\frac{1}{2}(V_1^{\dagger, \nu}(s_1) + V_1^{\mu, \dagger}(s_1)) - V_1^{\mu, \nu}(s_1)$, which measures the distances from the algorithm's output policies to each agent's best response policy. Figure 3 (a) compares the average NE-gap over the $K$ games between individual learning and meta-learning. Figure 3 (b) further compares the average

values achieved by the two methods. Again, we see that meta-learning finds better policy initialization in a new task and converges to an approximate NE policy (and value) using much fewer iterations.

### F.3 Scalability

To demonstrate the scalability of our algorithms, we further provide simulation results on some larger-scale tasks including a Poker endgame and a 1D linear-quadratic tracking task.

The Poker endgame that we consider here is a simplified version of the one used in [27]. We use a public River endgame ("Endgame A" of [27]) that was released in the Brains vs AI competition [6]. This task is a zero-sum game with 2 players and roughly 1.7 million states. We simplify the game setup by restricting to 2 actions (namely calling and folding) for each player. Poker is a partially observable game, but we found that our algorithm still performs well if each agent simply uses its local observation as the state. We generate a sequence of $K = 10$ similar games by adding $\mathcal{N}(0, 0.5)$ perturbations to the normalized stack amounts of the players, which essentially perturbs the reward functions. The convergence of the average NE-gap over the $K$ games in Figure 4(a) shows that our method can handle such a large state space, and our meta-learning method can converge to an approximate NE policy faster than individual learning.

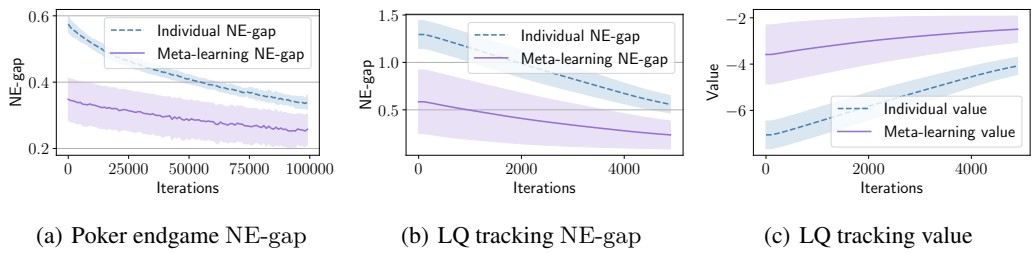

| (a) Poker endgame NE-gap | (b) LQ tracking NE-gap | (c) LQ tracking value |

Figure 4: Average NE-gaps and values of the policies output by individual learning and meta-learning in the Poker endgame and linear-quadratic tracking task. Shaded areas denote the standard deviations.

In the 1D linear-quadratic tracking problem, each agent tries to track the positions of the other agents and stay close to them. We adopt the discrete setting as has been utilized in a few recent works [52, 37, 46], which is an approximation of the classic continuous linear-quadratic formulations. This task has primarily been formulated as a mean-field game, but we consider a finite-agent variant of it in our simulations. Specifically, the task we consider can be modeled as a Markov potential game with 4 players, 625 states, and a joint action space of size 81. For each agent $i$, let $s_{t,i} \in \mathcal{S}_i$ and $a_{t,i} \in \mathcal{A}_i$, respectively, denote its local state (i.e., position) and local action at time step $t$, and we write $s_t = (s_{t,1}, \ldots, s_{t,4})$ and $a_t = (a_{t,1}, \ldots, a_{t,4})$. Each agent has 3 candidate actions $\mathcal{A}_i = \{-1, 0, 1\}$ and can stay at 5 different positions $\mathcal{S} = \{-2, -1, 0, 1, 2\}$. The state transition of agent $i$ is given by $s_{t+1,i} = s_{t,i} + a_{t,i}\Delta_t + \sigma \varepsilon_t \sqrt{\Delta_t}$, where $\Delta_t$ is the time duration, and $\varepsilon_t$ is the i.i.d. noise taking values from $\{-2, -1, 0, 1, 2\}$ following a normal distribution. Let $\mu_t$ denote the empirical mean of all the agents' positions at time $t$, i.e., $\mu_t = \frac{1}{4}\sum_{i=1}^{4} s_{t,i}$. The reward function for agent $i$ is specified as $r_i(s, a) = (-\frac{1}{2}a_{t,i}^2 - \frac{\kappa}{2}(\mu_t - s_{t,i})^2)\Delta_t$. Intuitively, this reward function incentivizes agents to track and stay close to the population (despite the random drift $\varepsilon_t$), but discourages agents from taking large-magnitude actions. We do not consider terminal costs in our simulations. The parameters are set as $\Delta_t = 1, \sigma = 1$, and $\kappa = 0.5$. We generate a sequence of similar games by adding $\mathcal{N}(0, 0.5)$ perturbations to the local state transition drift magnitudes. Figures 4(b) and 4(c) demonstrate that our meta-learning method achieves faster NE-gap and value convergences than individual learning in the linear-quadratic tracking task.