# OpenReview forum: "Multi-Agent Meta-Reinforcement Learning: Sharper Convergence Rates with Task Similarity"
_NeurIPS.cc/2023/Conference — NeurIPS 2023 poster_

### Official Review · Reviewer_Mmgh · 2023-07-04

**Soundness:** 4 excellent
**Presentation:** 4 excellent
**Contribution:** 3 good
**Rating:** 8
**Confidence:** 1

**Summary:**

This paper studies the interdependence between the convergence of MARL and the quality of policy initialization.

**Strengths:**

1. It proposes a new algorithm that has an initialization-dependent convergence guarantee.
2. It establishes several theoretical results that connect policy initialization and convergence of MARL in various types of games.

**Weaknesses:**

No

**Questions:**

No.

**Limitations:**

The authors have adequately addressed the limitations.

---

> ### Author Rebuttal · Authors · 2023-08-08
>
> We thank the reviewer for the appreciation of our work. We would be happy to discuss if the reviewer has any questions about the paper.

---

### Official Review · Reviewer_wdob · 2023-07-04

**Soundness:** 4 excellent
**Presentation:** 3 good
**Contribution:** 3 good
**Rating:** 8
**Confidence:** 4

**Summary:**

The paper introduces a meta-learning method to initialize the OOMD algorithm. Combined with the introduced initialization-dependent convergence guarantees, authors then can show faster convergence when the meta-learning initialization is close.

**Strengths:**

The paper is well written, and the prior work is well referenced.


The authors also provide extensive theoretical analysis of their algorithms. I think the most interesting result is Theorem 1 (other theorems often seem like an application of Theorem 1). I believe that the result is novel and interesting.


I also appreciate that authors analyze two-player zero-sum markov games as well as markov potential games


**Weaknesses:**

First, I agree that “closest to this work” is Harris, Keegan, et al. "Meta-learning in games." arXiv preprint arXiv:2209.14110 (2022). More recently, a closely related (also meta learning in games and regret minimization) paper presents an algorithm combining meta-learning and regret guarantees: Sychrovsky, David, et al. "Learning not to Regret." arXiv preprint arXiv:2303.01074 (2023). I believe that paper should be included in the related work.


My biggest issue though is the empirical analysis (Section 6 - Simulations).
There are no details in the main text and one can only find it in the appendix. It is fine to move details to the appendix, but the authors do not even mention the size of the games in the main text.
Looking into the appendix, I think the games are trivially small (2x2 matrix games). Furthermore, the games in the meta-learning sequence seem very similar (the epsilon noisy seems very small). I think this is rather confirmed by Figure 1c. It is then not a big surprise that the convergence is so fast, when the games are so small and the Nash Equilibria very similar/close. I think the authors should to evaluate substantially more interesting/diverse/larger games (e.g. some epsilon/noisy parametrization of small poker games).


If the authors add experiments in larger games than the trivially small currently included, I am happy to move my score to "Accept" as otherwise I think the paper is good.


----------------------------------------------------------------------------------------------------
Authors addressed my comments and included more experiments, increasing the score.


**Questions:**

Did you have a chance to run the algorithm on larger games than the small ones included in the current version of the submission?

**Limitations:**

Sufficient

---

> ### Author Rebuttal · Authors · 2023-08-08
>
> We thank the reviewer for the helpful feedback. Our detailed responses are as follows.
>
> 1. We thank the reviewer’s appreciation of our Theorem 1, which is indeed one of our most interesting results. While we agree that Theorem 2 can be considered as an application of Theorem 1 (as the reviewer mentioned), we would like to point out that our Theorem 4 (for potential games) and Theorems 5/6 (for general-sum games) are built upon quite different techniques and might be of interest to the reader as well.
>
> 2. We thank the reviewer for sharing the recent reference [Sychrovsky et al., 2023]. It focuses more on the empirical side of meta-learning in games, which is a bit different from our theoretical focus but is still a relevant and very interesting work. We have properly included [Sychrovsky et al., 2023] in our Related Work section. In addition, in this rebuttal, we have also added new simulations similar to the “kuhn_poker” example considered in [Sychrovsky et al., 2023]. The kuhn_poker task in [Sychrovsky et al., 2023] only considers 3 poker cards while our new simulation considers a full set of poker with 52 cards, which leads to a significantly larger state space. We would like to refer the reviewer to our “global” response for the details.
>
> 3. We appreciate the reviewer’s concern of our empirical analysis and for suggesting alternative experimental setups. We would like to remark that the focus of our work is mostly theoretical, and the simulations were primarily used as proof of concept. But given the reviewer’s interests in our empirical performances, we have added new and larger-scale simulations in this rebuttal to further evaluate the empirical performances of our algorithms. These new simulations include a task with more than 1.7 million states and a task with 4 agents, 625 states and 81 joint actions, both of which are significantly larger than our previous simulations. Please refer to our “global” response to all the reviewers for details of the new simulations.

---

> > ### Comment · Reviewer_wdob · 2023-08-12
> >
> > I appreciate the authors response, and I think the larger games experiments are a great addition.
> > Increasing my score to "Rating: 8: Strong Accept", nice paper!

---

### Official Review · Reviewer_CSow · 2023-07-05

**Soundness:** 2 fair
**Presentation:** 3 good
**Contribution:** 2 fair
**Rating:** 4
**Confidence:** 3

**Summary:**

The authors proposed a meta-learning approach based on MAML for multi-agent domains where tasks with similar NE policies, when learned sequentially, converges faster to desired equilibria solutions.

**Strengths:**

* Originality
The authors investigated theoretical convergence properties of multi-agent learning through the lens of meta-learning of a sequence of similar tasks. This is novel and the problem setting is relevant as non-stationarity is inherent to multi-agent learning that may benefit from the MAML family of methodologies.

* Quality
I find the theoretical results thorough but the work could benefit significantly with more substantial results on a few more standard benchmark domains.

* Clarity
The paper is generally well written and easy to follow.

**Weaknesses:**

* Task similarity metric: my main reservation with the proposed approach is related to the proposed task similarity metric (Sec 3.2) which translates to measuring similarity in different tasks' NE policy. Wouldn't selecting tasks in such way naturally promote faster convergence to NE policies when learning on "similar" tasks sequentially? I would appreciate further clarification on why this metric is used and if alternatives have been considered.

* Multi-agent learning inherently deals with non-stationarity during learning as the environment dynamics (from the perspective of any one player) is non-stationary. Quite often such non-stationarity does not translate to similar NE policy though (e.g. rock-paper-scissors, where NE polices would be quite different). Could you clarify if the proposed method should be applicable to such "similar tasks" of this nature?


**Questions:**

L82-83: " ... weaker solution concepts such as (C)CE ...", perhaps clarify that in general-sum games NE would not allow for coordination and is therefore restrictive in such games? By "weaker" perhaps you meant computationally tractable beyond two-player zero-sum games?

L49: "... convergence guarantees for MARL.", do you mean convergence to equilibria specifically? Perhaps clarify in writing that convergence bounds are wrt equilibria upfront?

Related Works: "meta-learning" is overloaded in the RL literature and another line of work in meta-learning follows from prior works such as  [1-3] where a policy is conditioned on prior belief over tasks and can infer Bayes-optimally through interaction at test time to adapt to different tasks. This idea is then extended to multi-agent setting [3-4], with a focus on faster convergence / transfer learning via shared representation learning. Would it make sense to include a brief discussion of this line of meta-learning works in related works?

[1] Meta-learning of Sequential Strategies: https://arxiv.org/abs/1905.03030
[2] Meta reinforcement learning as task inference: https://arxiv.org/abs/1905.06424
[3] Meta-Learning with Memory-Augmented Neural Networks: https://proceedings.mlr.press/v48/santoro16.pdf
[3] NeuPL: Neural Population Learning: https://arxiv.org/abs/2202.07415
[4] Simplex NeuPL: https://arxiv.org/abs/2205.15879

**Limitations:**

Please see my comments on the choice of task similarity metric which may be a limitation on when and where the proposed method could be effectively applied.

---

> ### Author Rebuttal · Authors · 2023-08-08
>
> We thank the reviewer for the insightful comments and the valuable suggestions on improving our work from multiple different perspectives. Our detailed responses are as follows.
>
> 1. We appreciate the reviewer’s concern of our task similarity metric. In Section 3.2, we choose the closeness of the NE policies of different games as the similarity metric because we believe it is most natural and comparably less restrictive. Such a metric also resonates existing works [Harris et al., 2022] for matrix games, which allows for direct comparisons. An important alternative metric that we have considered is the $L_1$ norm of the transition function together with the $L_{\infty}$ norm of the reward function; that is, two games are considered “similar” under this metric if their transition and reward functions are pointwise close. However, we believe that such a metric is more restrictive than the NE closeness metric we consider.  This is because, intuitively, in finite zero-sum games every saddle-point solution (NE) in mixed strategies is the solution to a linear program, and the optimal solutions of two LPs are very likely to be attained at neighboring points if these two LPs have similar constraints and objectives. In this sense, closeness of reward functions tends to imply closeness of NE, but not vice versa because if the optimal solutions of two LPs happen to be attained at the same point, this does not imply that their constraints or objectives are similar. Our NE closeness metric is hence less restrictive in this sense. We would be happy to learn if the reviewer can find any other natural and possibly less restrictive metric, which can be greatly helpful to improve our work.
>
> 2. For the “non-stationarity” part, if we understand it correctly, the reviewer’s concern is that there could be multiple NE in a game that are quite different, and there is no guarantee which one the learning algorithm will converge to due to the “non-stationary” learning dynamics. Our method should not be affected by such a case because our similarity metric only assumes one of the NE (not all of them) to be close to those of other games. In fact, the exact advantage of meta-learning is to be able to quickly find a particular NE that is close to the meta-initialization.
>
> 3. L82-83: By “a weaker solution concept”, we simply mean that every NE is automatically a (C)CE but not vice versa. As the reviewer has pointed out, NE is in general not computationally efficient beyond two-player zero-sum games, but (C)CE requires an additional public signal to correlate the players’ policies, which might not exist in practical scenarios. L49: Yes, we mean convergence to equilibria specifically. We thank the reviewer for calling attention to these points, and we have revised the paper to make these points clear.
>
> 4.  We thank the reviewer for sharing the references. These works tackle “meta-learning” from a slightly different perspective than ours, but we totally agree that they are very relevant and can help provide a more comprehensive viewpoint of the background to the readers. We have included a discussion of this line of research in our Related Work section.
>
> 5. We would also like to refer the reviewer to our “global” response section for new simulation results added during the rebuttal based on the feedback from the other reviewers. These new simulations include a task with more than 1.7 million states and a task with 4 agents, 625 states and 81 joint actions, both of which are significantly larger than our previous simulations.

---

> > ### Comment · Reviewer_CSow · 2023-08-11
> >
> > Thank you for your explanation.
> >
> > My two questions are related to each other. Let me clarify:
> >
> > In rock-paper-scissors, the BR may differ significantly, depending on the opponent strategy. However, the NE of (variations of, say by scaling all payoffs linearly) RPS game is the same, which according to your similarity metric the tasks would be similar.
> >
> > > Could you clarify if the proposed method should be applicable to such "similar tasks" of this nature?
> >
> > Quickly adapting to different opponent strategy in this case would be interesting and useful which is why I was wondering if your proposed method would be beneficial in such setting, though I understand this is a different problem setting than yours if I understood correctly. This is also why I referred to a different class of "meta-learning" algorithms in my comment which tackles this setting.
> >
> > My main reservation then is still if the proposed similarity metric doesn't make the sequential learning problem "easy" by definition. For instance, solving for NE in a sequence of re-scaled RPS game would be immediate and trivial. As such I would keep my initial rating.

---

> > > ### Author Response · Authors · 2023-08-15
> > >
> > > We thank the reviewer for the follow-up comments, which are helpful for us to better understand your questions. The reviewer is correct that our similarity metric is “behavior-dependent” in the sense that it depends on the best response policies of the players of each game rather than the exact NE: The former depends on the opponent strategy, but the latter only relies on the inherent properties of the game and is generally preferred. While our current solution only applies to the former case, we believe that extensions to the latter setting would be fairly straightforward. In particular, this is essentially the extension from Theorem 3.1 to Theorem 3.2 for meta-learning in normal-form games in [Harris et al., 2022], where their Theorem 3.1 assumes a similarity metric in terms of the optimum-in-hindsight strategies and Theorem 3.2 only depends on the similarity of any NE from each game. A similar extension can also be made possible in our setting for Markov games. We note that Theorem 3.2 of [Harris et al,. 2022] requires an additional assumption that after the termination of each game the players can obtain an exact NE of that game, which is also needed in our setting to make such extensions. When such an assumption does not hold, we can still use the approximate NE learned by the players at the end of each game, but our meta-learning convergence rate will suffer an additional term associated with the “inexactness” of the learned NE.
> > >
> > > We totally agree that “quickly adapting to different opponent strategies” is indeed a very interesting and useful setting and we thank the reviewer for sharing the references, but we believe that it is a different problem than what we considered. Using the second similarity metric as described in the paragraph above, the “different opponent behavior” does not affect the convergence behavior of our meta-learning methods, because our algorithms only depend on the properties of the game itself but not on the players’ behavior. One of our main contributions is to formally and quantitively show “why” and “how” the similarity metric makes the sequential learning problem “easier”.

---

### Official Review · Reviewer_kvor · 2023-07-07

**Soundness:** 2 fair
**Presentation:** 3 good
**Contribution:** 3 good
**Rating:** 6
**Confidence:** 3

**Summary:**

This paper establish theoretical results for meta-learning in a wide range of fundamental MARL settings, including learning Nash equilibria in two-player zero-sum Markov games and Markov potential games. Numerical results are shown to demonstrate the advantages of meta-learning.

**Strengths:**

This paper establish theoretical results for meta-learning in a wide range of fundamental MARL settings, including learning Nash equilibria in two-player zero-sum Markov games and Markov potential games. Numerical results are shown to demonstrate the advantages of meta-learning.

**Weaknesses:**

The simulation part is relatively simple since only toy examples are given. The advantages of meta-learning are not fully demonstrated.

**Questions:**

The simulation part is relatively simple since only toy examples are given. Is it possible to have much more sophisticated examples (at least more than two player) to demonstrate the theoretical results?

---

> ### Author Rebuttal · Authors · 2023-08-08
>
> We thank the reviewer for the valuable feedback. The focus of our work is mostly theoretical, and the simulations were primarily used as proof of concept. But given the reviewer’s interests in the empirical performances of our results, we have added new and larger-scale simulations in this rebuttal to further evaluate the empirical performances of our algorithms. These new simulations include a task with more than 1.7 million states and a task with 4 agents, 625 states and 81 joint actions, both of which are significantly larger than our previous simulations. Please refer to our “global” response to all the reviewers for details of the new simulations.

---

> > ### Comment · Reviewer_kvor · 2023-08-12
> >
> > Thanks the authros for taking the time to run additional experiments and answering the questions. I believe these additional experiments are very valuable validating the theoretical results.

---

### Official Review · Reviewer_8vaQ · 2023-07-11

**Soundness:** 3 good
**Presentation:** 3 good
**Contribution:** 3 good
**Rating:** 8
**Confidence:** 3

**Summary:**

The authors introduce theoretical results on Model-Agnostic Meta-Learning in a multi-agent reinforcement learning setting. In particular they show that meta-learning can achieve stronger convergence guarantees than an RL baseline when tasks are similar. The results hold for zero-sum, potential and general-sum games. In order to establish this, they provide theoretical results on convergence for online mirror descent which are dependent on initial conditions. Finally they conduct some small-scale experiments to validate the faster convergence empirically.

**Strengths:**

- Well motivated and timely article: meta-learning is becoming increasingly prevalent in the multi-agent setting, and theoretical results are lagging behind empirical ones, to the best of my knowledge.
- The notation is consistent and clear throughout.
- The theoretical results seem convincing, although I haven't checked the proofs in detail.
- There is an interesting and intuitive definition of game similarity, which may well find use in other algorithms in the future.

**Weaknesses:**

- The empirical results are quite small scale, and it would be useful to have a little more explanation about the environment in the main text.
- It would be nice to have at least one sketch proof in the main text. Perhaps there is a reorganisation of some other material to the Appendix that might permit this.
- There are a few places where more discussion or justification would be beneficial, for example: why is this "without loss of generality" on line 102; to what extent is it standard to neglect the logarithmic terms in line 201, and what is the reason that these terms show up.
- The limitations / future work section could have a little more detail. Would the authors be happy to comment on scaleability, and to discuss how similar results might be obtained for other meta-learning approaches outside MAML?

**Questions:**

On lines 94-95, the authors may want to cite another recent paper using meta-learning in the context of a distribution over multi-agent cooperative tasks: https://arxiv.org/abs/2301.07608.

**Limitations:**

See Weaknesses.

---

> ### Author Rebuttal · Authors · 2023-08-08
>
> We thank the reviewer for the appreciation of our work and the valuable feedback. Our detailed responses are as follows.
>
> 1. We thank the reviewer for the comments on our empirical results. The focus of our work is mostly theoretical, and the simulations were primarily used as proof of concept. But given the reviewer’s interests in the empirical performances of our results, we have added new and larger-scale simulations in this rebuttal to further evaluate the empirical performances of our algorithms. These new simulations include a task with more than 1.7 million states and a task with 4 agents, 625 states and 81 joint actions, both of which are significantly larger than our previous simulations. Please refer to our “global” response to all the reviewers for details of the new simulations.
>
> 2. We appreciate the reviewer’s advice on the organization of the paper. We will follow these guidelines to include more discussions of the simulation setup as well as a proof sketch in the main text when preparing for the camera-ready version (where one extra page is allowed).
>
> 3. In Line 102, our results readily extend to the setting where the initial state is sampled from an arbitrary distribution. Assuming a fixed initial state is without loss of generality because one can imagine that there exists an extra step $h=0$ where the agent starts from a fixed state $s_0$. The transition probabilities from $s_0$ to any other state $s_1$ can be made arbitrary, which equivalently allows us to start from an arbitrary state distribution at $s_1$. This is a very common assumption in existing works, e.g., [Jin et al., 2022; Song et al., 2022]. In Line 201, suppressing logarithmic terms is also common in the literature as the log terms grow much more slowly than the polynomial terms. In fact, most existing results do have this logarithmic term, e.g., [Zhang et al., 2022]. In our work, the direct reason of this log term is that we divide the $T$ iterations into $\bar{\tau}=O(\log T)$ stages. We have added more discussions in our paper to make these points clear to the reader.
>
> 4. For the limitations / future work section, we believe that our methods will not suffer much from scalability issues because our algorithms are essentially designed to be decentralized (in the same sense as discussed in [Mao et al., 2022]): The agents only use local information to update their policies and rarely need to exchange information with others. In addition, while our proofs do not directly generalize to other meta-learning approaches outside MAML, we believe that our analytical methodology (properly defining a similarity metric and then tracking how the policy trajectories on different tasks deviate according to the metric) could still be helpful.
>
> 5. We thank the reviewer for reminding us of the recent paper that applies meta-learning to cooperative tasks. We have properly included the reference in the Related Work section.

---

> > ### Comment · Reviewer_8vaQ · 2023-08-18
> > **Response to Authors**
> >
> > I thank the authors for their rebuttal. They have adequately addressed my points. I am particularly pleased to see the strong results of their algorithm on a larger game.
> >
> > Therefore I will increase my score to "Strong Accept".

---

### Author Rebuttal · Authors · 2023-08-08

We thank all the reviewers for the insightful feedback. In this “global” response, we would like to share some new and larger-scale simulations that we conducted in this rebuttal phase following some of the reviewers’ advice. We believe these new simulations can help address the reviewers’ questions on the empirical performances of our algorithms.

We added two sets of new simulations. The first task is a Poker endgame similar to the one considered in [Harris et al., 2022]. We use a public River endgame that was released by the authors [Brown and Sandholm, 2018] in the Brains vs AI competition. This task is a zero-sum game with 2 players, ~1.7 million states, and 2 actions (calling or folding) for each player. Poker is a partially observable game, but we found that our algorithm still performs well if each agent simply uses its local observation as the state. We generated similar games by adding $\mathcal{N}(0, 0.5)$ perturbations to the normalized stack amounts of the players, which essentially perturbates the reward function. Figure 1 in the attached PDF file shows that our method can handle such a large state space well, and our meta-learning method can converge to an approximate NE policy faster than individual learning.

In the second task, we consider a 1D linear-quadratic tracking problem where each agent tries to track and state close to the other agents. We adopt the discrete setting as has been utilized in a few recent works [Perrin et al., 2020; Laurière et al., 2022]. The tracking task we consider is a Markov potential game with 4 players, 625 states, and a joint action space of size 81. For each agent $i$, its location transition is given by $s_{t+1,i}=s_{t,i}+a_{t,i} \Delta_t +\epsilon_t \sqrt{\Delta_t}$, where $\Delta_t$ is the time duration, and $\epsilon_t$ is the i.i.d. noise taking values from $\{-2, -1, 0, 1, 2\}$ following a normal distribution. Let $\mu_t$ denote the empirical mean of all the agents’ locations at time $t$. The reward function for agent $i$ is specified as $(-\frac{1}{2}a_{t,i}^2-\frac{1}{4}(\mu_t – s_{t,i})^2)\Delta_t$. Intuitively, this reward function incentivizes agents to track and stay close to the population (despite the random drift $\epsilon_t$), but discourages agents from taking large-magnitude actions. We generated similar games by adding $\mathcal{N}(0, 0.5)$ perturbations to the location transition drift magnitude and the reward functions. Figures 2 and 3 in the attached PDF file demonstrate that our meta-learning method achieves faster NE-gap and value convergences.

References:

M. Laurière, S. Perrin, S. Girgin, P. Muller, A. Jain, T. Cabannes, G. Piliouras, J. Pérolat, R. Élie, O. Pietquin, et al. Scalable deep reinforcement learning algorithms for mean field games. arXiv preprint arXiv:2203.11973, 2022.

S. Perrin, J. Pérolat, M. Laurière, M. Geist, R. Elie, and O. Pietquin. Fictitious play for mean field games: Continuous time analysis and applications. Advances in Neural Information Processing Systems, 33:13199–13213, 2020.

---

### Decision · Program_Chairs · 2023-09-21

**Decision:**

Accept (poster)

**Comment:**

This paper studied meta-learning in several fundamental MARL settings (learning NE in zero-sum Markov games and Markov potential games, learning CCE in general-sum Markov games) via learning a good policy initialization to adapt to similar tasks. As the core contribution, the paper demonstrated sharper convergence results with meta-learning in theory and illustrated its numerical advantages in several examples. While the main analysis heavily build on existing convergence results for learning zero-sum games, MPGs, and general-sum games, most reviewers agree that the meta-learning results are interesting and novel. The additional large-scale experiments provided during the rebuttal phase also help strengthen the contribution. Therefore, I am recommending for acceptance and ask authors to provide implementation details and discussions on these additional experiments in the revision.